# GOBAI-O₂: temporally and spatially resolved fields of ocean interior dissolved oxygen over nearly two decades

Jonathan D. Sharp[1,2], Andrea J. Fassbender[2], Brendan R. Carter[1,2], Gregory C. Johnson[2], Cristina Schultz[3,4], John P. Dunne[3]

[1]Cooperative Institute for Climate, Ocean, and Ecosystem Studies, University of Washington, Seattle, WA, 98105, United States

[2]Pacific Marine Environmental Laboratory, National Oceanic and Atmospheric Administration, Seattle, WA, 98115, United States

[3]Geophysical Fluid Dynamics Laboratory, National Oceanic and Atmospheric Administration, Princeton, NJ, 08540, United States

[4]Princeton University, Princeton, NJ, 08540, United States

*Correspondence to*: Jonathan D. Sharp (jonathan.sharp@noaa.gov)

**Abstract.** For about two decades, oceanographers have been installing oxygen sensors on Argo profiling floats to be deployed throughout the world ocean, with the stated objective of better constraining trends and variability in the ocean's inventory of oxygen. Until now, measurements from these Argo-mounted oxygen sensors have been mainly used for localized process studies on air–sea oxygen exchange, upper ocean primary production, biological pump efficiency, and oxygen minimum zone dynamics. Here we present a new four-dimensional gridded product of ocean interior oxygen, derived via machine learning algorithms trained on dissolved oxygen observations from Argo-mounted sensors and discrete measurements from ship-based surveys, and applied to temperature and salinity fields constructed from the global Argo array. The data product is called GOBAI-O₂ for Gridded Ocean Biogeochemistry from Artificial Intelligence – Oxygen (Sharp et al., 2022; https://doi.org/10.25921/z72m-yz67; last access: 07 Jul. 2023); it covers 86% of the global ocean area on a 1° latitude by 1° longitude grid, spans the years 2004–2022 with monthly resolution, and extends from the ocean surface to two kilometers in depth on 58 levels. Two types of machine learning algorithms — random forest regressions and feed-forward neural networks — are used in the development of GOBAI-O₂, and the performance of those algorithms is assessed using real observations and simulated observations from Earth system model output. Machine learning represents a relatively new method for gap-filling ocean interior biogeochemical observations and should be explored along with statistical and interpolation-based techniques. GOBAI-O₂ is evaluated through comparisons to the oxygen climatology from the World Ocean Atlas, the mapped oxygen product from the Global Ocean Data Analysis Project, and direct observations from large-scale hydrographic research cruises. Finally, potential uses for GOBAI-O₂ are demonstrated by presenting average oxygen fields on isobaric and isopycnal surfaces, average oxygen fields across vertical–meridional sections, climatological seasonal cycles of oxygen averaged over different pressure layers, and globally integrated time series of oxygen. GOBAI-O₂ indicates that the oxygen inventory in the upper two kilometers of the global ocean has been declining by about $0.79 \pm 0.04$ % decade$^{-1}$ between 2004 and 2022.

# 1 Introduction

The inventory of dissolved oxygen in the global ocean has been declining over recent decades and is projected to continue declining through the current century (Keeling et al., 2010; Breitburg et al., 2018; Bindoff et al., 2019; Stramma and Schmidtko, 2019; Limburg et al., 2020), leading to detrimental consequences for aerobic marine organisms (Pörtner and Farrell, 2008; Sampaio et al., 2021) and changes to biogeochemical cycles, potentially triggering important climatological feedbacks (Gruber, 2004; Berman-Frank et al., 2008). Historical deoxygenation has been inferred from analyses of globally distributed observations (Helm et al., 2011; Schmidtko et al., 2017; Ito et al., 2017) and has been reproduced in Earth system model (ESM) reconstructions (Bopp et al., 2013; Frölicher et al., 2009; Kwiatkowski et al., 2020). Global observational studies have generally indicated a greater degree of deoxygenation than model studies over recent decades, indicating that ESMs may misrepresent the sensitivities of the physical and biological processes leading to deoxygenation, which has implications for the reliability of future projections (Oschlies et al., 2017; 2018; Stramma and Schmidtko, 2021). Model studies, however, are based on gridded output that is continuously resolved in space and time, whereas observational studies rely on interpolation of measurements from discrete bottle samples and/or profiling sensors. These observational datasets have significant spatiotemporal gaps and may not robustly represent global deoxygenation trends.

Discrete measurements of dissolved oxygen concentration ($[O_2]$) are typically made using Winkler titrations (Winkler, 1888; Carpenter, 1965; Langdon, 2010), which are also used to calibrate measurements from electrode (or more recently sometimes optical) dissolved oxygen sensors mounted on Conductivity-Temperature-Depth (CTD) profilers. Globally distributed $[O_2]$ observations from discrete measurements and CTD profilers have been provided by hydrographic programs like the World Ocean Circulation Experiment (WOCE), the Climate and Ocean: Variability, Predictability and Change (CLIVAR) program, and the Global Ocean Ship-Based Hydrographic Investigations Program (GO-SHIP). Data from these programs are publicly available and are conveniently compiled into databases such as the World Ocean Database (WOD; Boyer et al., 2018) and the Global Ocean Data Analysis Project (GLODAP; Lauvset et al., 2022). Though unprecedented spatial coverage is provided by global hydrographic programs, the decadal-scale temporal resolution of WOCE, CLIVAR, and GO-SHIP data precludes robust analyses of year-to-year and/or seasonal variability in $[O_2]$.

Since the mid-2000s, approximately 1,800 Argo floats equipped with oxygen sensors have been deployed. Argo floats profile the upper ~2000 meters of the water column every ~10 days. Many oxygen-equipped Argo floats have been deployed as parts of regional arrays such as the Southern Ocean Carbon and Climate Observations and Modeling (SOCCOM; soccom.princeton.edu) project and the North Atlantic Aerosols and Marine Ecosystems Study (NAAMES; science.larc.nasa.gov/NAAMES/). More recently, the push for a global biogeochemical Argo array has spurred the deployment of oxygen-equipped Argo floats into more sparsely sampled ocean regions (Johnson and Claustre, 2016; Claustre et al., 2020). As more floats have been deployed, improvements have been made to sensor calibration, data adjustments, and quality control.

Notably, pre-deployment drift corrections (D'Asaro and McNeil, 2013; Johnson et al., 2015; Bittig and Körtzinger, 2015; Bushinsky et al., 2016; Drucker and Riser, 2016; Nicholson and Feen, 2017), climatology-based calibrations (Takeshita et al., 2013), calibrations via in-air oxygen measurements (Körtzinger et al., 2005; Fiedler et al., 2013; Bittig and Körtzinger, 2015; Johnson et al., 2015; Bushinsky et al., 2016), post-deployment drift corrections (Johnson et al., 2017; Bittig et al., 2018a), and established procedures for delayed-mode quality control (Maurer et al., 2021) have substantially reduced the uncertainty and increased the reproducibility of optode-based $[O_2]$ measurements on Argo floats.

From the time it began, the Argo-Oxygen program (now oxygen is a measured variable under the Biogeochemical Argo program) intended to document ocean deoxygenation, predict and assess anoxic and hypoxic events, and determine seasonal to interannual changes in export production (Gruber et al., 2010). Until now, these goals have been achieved primarily on a regional scale. For example, $[O_2]$ measurements from biogeochemical Argo floats have been used to examine ventilation and air–sea exchange of oxygen in the Southern Ocean (Bushinsky et al., 2017) and during Deep Water formation in the Subpolar North Atlantic (Körtzinger et al., 2004; Piron et al., 2016; 2017; Wolf et al., 2018); denitrification and the spatial extent of the oxygen minimum zone in the Bay of Bengal (Sarma and Udaya Bhaskar, 2018; Johnson et al., 2019; Udaya Bhaskar et al., 2021); and carbon production and export in the Pacific Ocean (Bushinsky and Emerson, 2015, 2018; Yang et al., 2017), Southern Ocean (Stukel and Ducklow, 2017; Arteaga et al., 2019), and North Atlantic Ocean (Alkire at al., 2012; Estapa et al., 2019). Recently, in an early global-scale analysis of $[O_2]$ from the Argo array, Johnson and Bif (2021) used the diel cycle of oxygen measured by the ocean-wide array of biogeochemical Argo floats to constrain net primary production in the surface ocean.

With the work presented here, we seek to capitalize on the collective efforts of global hydrographic programs, Biogeochemical Argo, and Core Argo (Johnson et al., 2022) to create a first-of-its-kind data product: a four-dimensional monthly record of dissolved oxygen in the global ocean. We combine autonomous observations of $[O_2]$ from BGC Argo floats with discrete observations of $[O_2]$ from hydrographic cruises in the GLODAP database to create a dataset with extensive spatial and temporal resolution. With this dataset, we train machine learning algorithms on ocean interior predictor variables co-located with $[O_2]$ observations; evaluate those algorithms using real and simulated data; and apply the algorithms to gridded ocean interior predictor variables mapped from Core Argo to produce a gridded $[O_2]$ data product at a monthly resolution from 2004–2022, on 58 pressure levels in the upper two kilometers of the ocean, and on a near-global 1° latitude by 1° longitude grid.

In this paper, we present the four-dimensional gridded $[O_2]$ product, which we call GOBAI-$O_2$: Gridded Ocean Biogeochemistry from Artificial Intelligence – Oxygen (Sharp et al., 2022; https://doi.org/10.25921/z72m-yz67; last access: 07 Jul. 2023). Artificial intelligence (AI) is a broad term describing computerized systems that are able to recognize patterns, make decisions, and otherwise perform tasks previously reserved only for humans; GOBAI-$O_2$ is built using machine learning (ML), which is a subfield of AI that is focused on training, understanding, and applying algorithms that leverage data to

artificially learn and reproduce patterns. We introduce GOBAI- $O_2$ by analyzing spatial patterns, seasonal cycles, and decadal variability. We also describe the process for creating GOBAI-$O_2$, show the results of evaluation exercises, assess uncertainty in the gridded [$O_2$] fields, and compare the data product to other gridded datasets and discrete measurements. GOBAI-$O_2$ represents the first step in leveraging the emerging global array of BGC Argo floats to produce spatially-resolved, time-varying snapshots of global ocean biogeochemical distributions in near real-time. It also emphasizes regions that are good candidates for new observational assets; particularly where gaps in observational coverage coincide with high background variability in [$O_2$]. Critically, GOBAI-$O_2$ can be used to address the goals of the Argo-Oxygen program set by Gruber et al. (2010) over a decade ago, providing regional and global insight into ocean deoxygenation and hypoxia on timescales ranging from a few months to multiple years.

## 2 Methods

### 2.1 Data sources and processing

Hydrographic cruise data were obtained from the GLODAP version 2022 data product (GLODAPv2.2022; Key et al., 2015; Olsen et al., 2016; Lauvset et al., 2022). GLODAPv2.2022 provides quality-controlled data from throughout the entire water column obtained via discrete analyses of more than 1.4 million water samples collected on 1,085 research cruises. Discrete Winkler titration data were chosen rather than CTD oxygen profiles due to the issues with the quality of calibration of a subset of CTD oxygen measurements and the relatively coarse vertical resolution of the final GOBAI-$O_2$ product, which would not benefit from the high vertical resolution of CTD profiles. Data from GLODAP were chosen rather than data from the WOD or any other database due to the high degree of quality control applied to GLODAP data. Dissolved oxygen is the most represented biogeochemical variable in GLODAPv2.2022, with more than 1.2 million data points from 991 research cruises. Data from GLODAPv2.2022 were filtered to retain only samples collected after 1 Jan. 2004, from 0 to 2500 decibars (dbars), and with a quality-control flag of 1 (meaning the data were manually inspected) and quality flags of 2 (good) for both salinity and [$O_2$]. Temperature is not assigned either flag and is assumed to be of sufficient quality if it is reported (Lauvset et al., 2022). This filtering left 450,032 data points from 21,513 unique profiles from 393 total cruises (red points in Fig. 1).

Float data were obtained from synthetic profile ("Sprof") files (Bittig et al., 2022) stored in the Argo Global Data Assembly Centres (GDACs) via the OneArgo-Mat toolbox (Frenzel et al., 2022) for MATLAB (MathWorks). At the time data were obtained (13 Jun. 2023), the Argo GDACs contained data from about 1800 floats equipped with [$O_2$] sensors. Float data were filtered to retain only delayed-mode-adjusted data with quality flags of 1 (good), 2 (probably good), or 8 (interpolated/extrapolated) for pressure, temperature, salinity, and [$O_2$]. This filtering step ensured float data had been manually reviewed by a data manager and assigned an appropriate quality flag. This filtering left 27,832,192 data points from 138,180 unique profiles from 1022 total floats (blue points in Fig. 1). Of the float profiles, 51.4% were quality-controlled by comparisons to climatologies (WOA or CARS), 30.3% using in-air oxygen measurements, 7.0% by comparisons to subsurface

measurements (World Ocean Database, OMZ assumed to have zero oxygen, or CTD profile upon deployment), 5.3% using the in situ optode calibration of Drucker and Riser (2016), 3.3% via another method, 1.9% were uncategorized, and the remaining 0.9% were not adjusted.

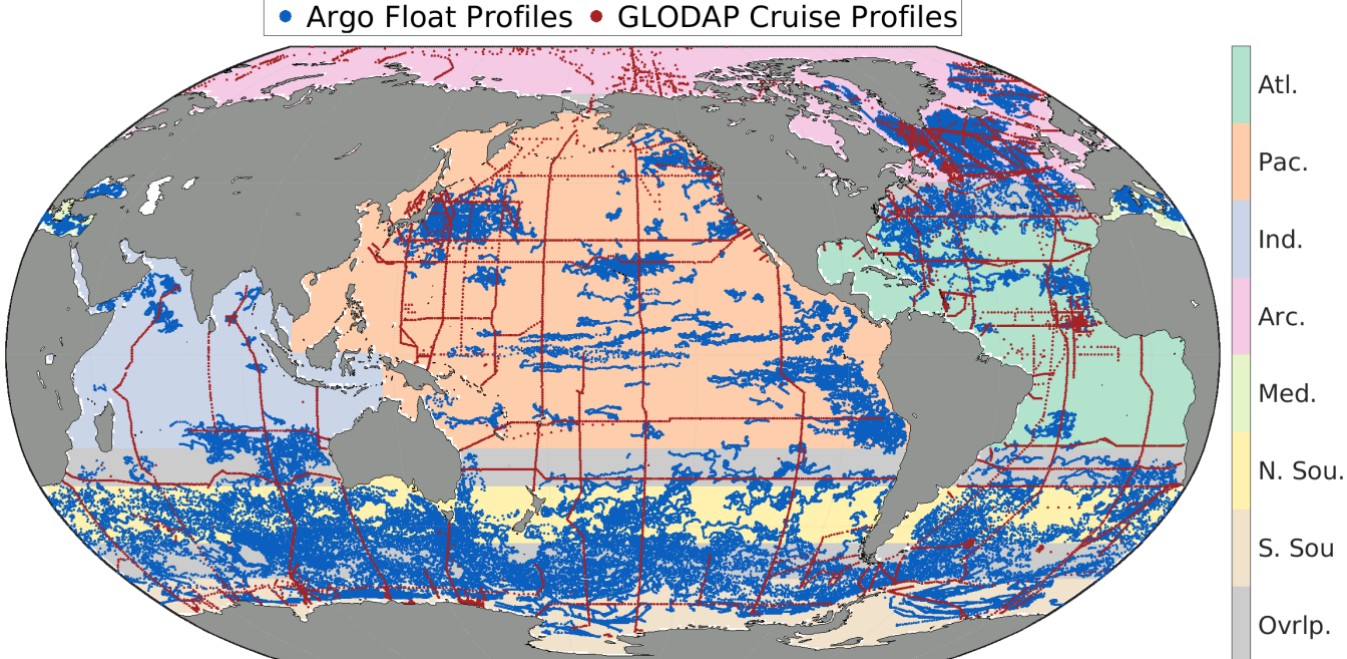

**Figure 1:** Discrete profile locations from oxygen-equipped Argo floats (blue) and GLODAPv2.2022 cruises (red) from 1 Jan. 2004 to 13 Jun. 2023. Data from these profiles were binned and used to train ML algorithms to estimate $[O_2]$ in each of seven regions: the Atlantic Ocean (Atl.), Pacific Ocean (Pac.), Indian Ocean (Ind.), Arctic Ocean (Arc.), Mediterranean Sea (Med.), northern section of the Southern Ocean (N. Sou.), and southern section of the Southern Ocean (S. Sou.). Overlapping areas between regions are shown in grey (Ovrlp.), where $[O_2]$ estimates are made by taking distance-weighted averages of outputs from two regional ML algorithms. The regional boundaries are presented in numerical form in Table B1.

The discrete temperature, salinity, and $[O_2]$ data obtained from GLODAPv2.2022 and the Argo GDACs are archived online (Appendix C; https://doi.org/10.5281/zenodo.7747237). To ensure the trained machine learning algorithms were not biased toward BGC Argo float data, which in their native format have higher vertical resolution than GLODAP data, each profile was interpolated to, at most, 58 standard pressure levels (the same pressure levels on which the final GOBAI-O$_2$ data product is provided). Interpolated temperature, salinity, and $[O_2]$ data from each source are also archived online (Appendix C; https://doi.org/10.5281/zenodo.7747237). After interpolation, the total number of GLODAP data points used for algorithm training increased to 1,096,324 and the total number of Argo float data points used for algorithm training decreased to 6,635,749. Co-located, interpolated GLODAP and BGC Argo profiles that fell within the same 1° × 1° monthly, depth-dependent grid cells were compared for internal consistency. The float $[O_2]$ values were adjusted according to the procedure

in Appendix D to remove the small global discrepancy between co-located ship and float measurements to ensure internal consistency between the two datasets.

BGC Argo float and GLODAP cruise data were combined into a single dataset after this bias adjustment, which will be referred to as the "combined dataset" from here on. The combined dataset was grouped into seven overlapping regions (Fig. 1, Table B1). This grouping was intended to account implicitly for similar physical–biogeochemical relationships within large ocean regions and to reduce the computational burden of the machine learning (ML) algorithm fits described below. The regions were initially chosen to imitate the biomes presented by Fay and McKinley (2014), and were then expanded to relatively large regions bound either by land masses or by overlapping boundaries along constant lines of latitude. The number of profiles made within each $1° × 1°$ box by either a discrete ship cast or Argo float (supplementary material, Fig. A1) provides a measure of the temporal resolution of the combined dataset in addition to the spatial distribution shown in Fig. 1.

Gridded temperature and salinity data to which the trained algorithms were applied were obtained from the latest version of the Roemmich and Gilson (2009) (RG09) Argo Climatology (https://sio-argo.ucsd.edu/RG_Climatology.html; last access: 12 Jan. 2023). The RG09 climatology is an upper ocean (0–2000 dbar) gridded temperature and salinity product constructed exclusively from Argo observations. Long-term (2004–2018) mean fields of temperature and salinity are provided on 58 pressure levels, along with monthly anomaly fields on each of those pressure levels from 2004 to the present day. The most recent major update of the RG09 climatology was made in 2019, and new monthly anomaly fields are provided in near-real-time between major updates. Monthly gridded temperature and salinity were calculated from the RG09 long-term mean and monthly anomaly fields (Fig. A2), then used for the creation of the gridded $[O_2]$ product discussed below.

Output from the NOAA Geophysical Fluid Dynamics Laboratory's Earth System Model Version 4 (GFDL-ESM4; Dunne et al., 2020) was used to assess algorithm performance. Model output was downloaded from the World Climate Research Programme database (https://esgf-node.llnl.gov/projects/cmip6/; last access: 8 Apr. 2022), which hosts data from models participating in the sixth phase of the Coupled Model Intercomparison Project (CMIP6). Potential temperature, practical salinity, and $[O_2]$ were downloaded to coincide with available ocean interior observations (Fig. A3). Historical outputs (2004–2014) and projected outputs under SSP2-4.5 (2015–2022) were combined to cover the time period over which observations were available. A spatial mask was applied to retain only GFDL-ESM4 grid cells with corresponding temperature and salinity values in the RG09 climatology, because that is the final grid on which GOBAI-$O_2$ is produced.

## 2.2 Algorithm training

The combined dataset was used to train ML algorithms for each region to estimate $[O_2]$ from absolute salinity, conservative temperature, potential density anomaly, hydrostatic pressure, bottom depth, and additional spatiotemporal information to allow for geographic, seasonal, and interannual variation (see Table 1). Though biology is not explicitly accounted for in the ML

algorithms, Giglio et al. (2018) demonstrate that, with an appropriately distributed dataset, the inclusion of spatiotemporal variables in algorithm training can implicitly accommodate biological processes.

Absolute salinity ($S_A$) was calculated from practical salinity ($S_P$), hydrostatic pressure ($P$), latitude, and longitude. Conservative temperature ($\theta$) was calculated from in situ temperature ($T$), $S_A$, and $P$. Potential density anomaly ($\sigma_\theta$) was calculated from $S_A$ and $\theta$. These calculations were made using the Gibbs-SeaWater (GSW) Oceanographic Toolbox for MATLAB (McDougall and Barker, 2011). As was done by Carter et al. (2021), longitude was transformed into two separate predictors: cos(Longitude − 20° E) and cos(Longitude − 110° E). Cosine functions were applied to maintain the cyclical nature of longitude as a predictor, and offsets of 20° E and 110° E were intended to shift regions where the cosine function has minimum explanatory power over landmasses. Bottom depth was determined by matching each observational location with the corresponding bathymetry from the ETOPO2v2 global relief model (NOAA National Geophysical Data Center, 2006).

**Table 1.** Predictor variables used to train random forest regressions and feed-forward neural networks to predict [$O_2$].

| Predictor Variable | Abbreviation | Unit | Range (approx.) |
|---|---|---|---|
| Conservative Temperature | $\theta$ | °C | −2 to 32 |
| Absolute Salinity | $S_A$ | N/A | 14 to 40 |
| Potential Density Anomaly | $\sigma_\theta$ | kg m$^{-3}$ | 9.3 to 29.3 |
| Hydrostatic Pressure | $P$ | dbar | 0 to 2000 |
| Latitude | $lat$ | ° | −78 to 90 |
| cos(Longitude – 20°) | $lon_{cos20}$ | N/A | −1 to 1 |
| cos(Longitude – 110°) | $lon_{cos110}$ | N/A | −1 to 1 |
| Bottom depth | $bot$ | meters | 0 to 10,000 |
| Year | $yr$ | years | 2004 to 2023 |
| sin(2π · Day of Year / 365.25) | $doy_{sin}$ | N/A | −1 to 1 |
| cos(2π · Day of Year / 365.25) | $doy_{cos}$ | N/A | −1 to 1 |

Two types of ML algorithms were trained: feed-forward neural networks (FNNs; Demuth et al., 2008) and random forest regressions (RFRs; Breiman, 2001), each of which were trained on the input variables given in Table 1 to produce estimates of [$O_2$] (Fig. A4). Three separate FNNs were trained for each of the seven basins shown in Fig. 1, with an average of the three taken to obtain one equally weighted FNN result. The FNNs were constructed using the "feedforwardnet" function and trained using the "train" function, both from Version 14.4 of the Deep Learning Toolbox for MATLAB (R2022a). Each FNN was trained using a Levenberg-Marquardt algorithm, with 15% of the data reserved for testing the network during training steps. Each FNN had two hidden layers, with the following combinations of neurons in the first and second layer, respectively: 20 and 10, 15 and 15, and 10 and 20. One RFR was trained for each of the seven basins shown in Fig. 1. RFRs are ensembles of decision trees, each created with a bootstrapped version of the full dataset chosen randomly with replacement. Each RFR consisted of 600 trees, a minimum leaf size of 10, and six of the eleven predictors used for each decision split. These parameters

were chosen after some trial and error to strike a balance between computational efficiency and algorithm performance. The MATLAB "treebagger" function was used to train RFRs.

In areas where two regions overlap (see Fig. 1), weighted averages of $[O_2]$ estimates were calculated in overlapping grid cells from each regional algorithm. These averages were weighted by distance from the center latitude line of the overlapping area (e.g., a point at 33 °S in the overlapping area between the N. Sou. region (whose northern border extends to 25 °S) and the Atl. region (whose southern border extends to 35 °S) would be calculated as $[O_2] = 0.8[O_2]_{N.Sou.} + 0.2[O_2]_{Atl.}$). Overlapping areas

were used to mitigate discontinuities at the boundaries between regions in the final gridded product.

The average of FNN and RFR estimates (ENS, for ensemble average) was used as the $[O_2]$ estimate for a given set of input data. This ensemble averaging procedure was implemented due to insights from previous work showing that averaging the outputs of multiple ML algorithms or linear regression models often outperforms the output from just one approach on its own

(Gregor et al., 2017; 2019; Bittig et al., 2018b; Carter et al., 2021; Djeutchouang et al., 2022), likely due to complementary strengths and weaknesses of each approach. For this work, any especially erroneous result from either the FNN or RFR should be mitigated by better results from the other algorithm.

## 2.3 Algorithm evaluation

We performed two exercises to evaluate the effectiveness of the ML algorithms used to estimate $[O_2]$. The first exercise

involved training separate evaluation algorithms ($RFR_{Data-Eval}$ and $FNN_{Data-Eval}$ algorithms) as described in section 2.2 using a subset of the observational dataset for training while reserving the remaining subset for assessment. For this exercise, data were split randomly into training (80%) and assessment (20%) groups; this split was made according to measurement platform (cruise or float; see Fig. A5) to ensure that inherent correlations among the data points from a single cruise or float did not contribute to the apparent effectiveness of each ML algorithm. Then $[O_2]$ values from the subset of reserved assessment data

were compared to estimates of $[O_2]$ from $RFR_{Data-Eval}$, $FNN_{Data-Eval}$, and the ensemble average of the two ($ENS_{Data-Eval}$). This exercise was intended to evaluate the ability of the ML algorithms to reproduce measured data that was not involved in algorithm training (section 3.1.1).

The second exercise involved training evaluation algorithms ($RFR_{ESM4-Eval}$ and $FNN_{ESM4-Eval}$ algorithms) using synthetic

"profiles" extracted from gridded GFDL-ESM4 output at the times and locations where observational data were available, then assessing the evaluation algorithms using spatially and temporally continuous monthly GFDL-ESM4 output from 2004 through 2022. For this exercise, synthetic profiles for algorithm training were defined by matching the latitude, longitude, month, and year of each available grid cell from the binned observational dataset with the corresponding GFDL-ESM4 output. This resulted in 75,879 synthetic profiles for algorithm training. $RFR_{ESM4-Eval}$ and $FNN_{ESM4-Eval}$ algorithms were trained as

described in section 2.2 with the synthetic training data, then used to produce $[O_2]$ estimates for the complete model output.

These [O$_2$] estimates from RFR$_{ESM4-Eval}$, FNN$_{ESM4-Eval}$, and an ensemble average of the two (ENS$_{ESM4-Eval}$) were compared to [O$_2$] values from the full GFDL-ESM4 output fields at the grid-cell level. This exercise was intended to evaluate the ability of the ML algorithms to estimate [O$_2$] in a spatiotemporally resolved Earth system model environment when limited to training data representative of the available collection of ocean oxygen observations (section 3.1.2). The four-dimensional field of [O$_2$] from ENS$_{ESM4-Eval}$ that represents a reconstruction of the GFDL-ESM4 environment, which we refer to as GOBAI-O$_2$-ESM4, can also be used as an analogue for how well GOBAI-O$_2$ (trained on real observational data, section 2.4) might represent [O$_2$] variability in the real-world environment. For this reason, the four-dimensional field of differences between GOBAI-O$_2$-ESM4 and GFDL-ESM4 output were used to inform the evaluation of GOBAI-O$_2$ uncertainty (sections 2.5 and 3.2.4). Additionally, we quantified global means, seasonal cycle amplitudes, long-term trends, and interannual variabilities in [O$_2$] across different pressure layers of GOBAI-O$_2$-ESM4. To evaluate the performance of GOBAI-O$_2$-ESM4 on a global scale, these metrics are compared to the same metrics for the spatiotemporally resolved GFDL-ESM4 output and subsampled grid cells in GFDL-ESM4 corresponding to observational data coverage (section 3.1.2). Comparisons of global means from GOBAI-O$_2$-ESM4 to GFDL-ESM4 are also used to approximate uncertainty in oxygen inventories for the assessment of trends (section 3.2.3).

## 2.4 Creation of GOBAI-O$_2$

FNNs and RFRs for each of the seven regions shown in Fig. 1 were trained with the full combined dataset, using the predictor variables shown in Table 1, with [O$_2$] as a target variable. Then, the FNNs and RFRs were applied to $S_A$, $\theta$, and $\sigma_\theta$ calculated from RG09 temperature and salinity fields, along with spatiotemporal information from RG09 grid cells. Weighted averages were calculated where regions overlapped, and ensemble averages (ENS) were calculated from the FNN and RFR estimates. One exception occurred in the subsurface of the north Arabian Sea (above 21° N, between 900 and 1412.5 dbars), where erroneously high ENS [O$_2$] values caused by a clearly non-physical feature in RFR [O$_2$] values were replaced by NN [O$_2$] values only. This produced a monthly gridded [O$_2$] product in the upper two kilometers of the ocean on a global grid from January 2004 to December 2022, i.e. GOBAI-O$_2$ (Sharp et al., 2022; https://doi.org/10.25921/z72m-yz67; last access: 07 Jul. 2023; sections 3.2.1–3.2.3). GOBAI-O$_2$ was compared to gridded climatological oxygen fields from the 2018 World Ocean Atlas (WOA18; Garcia et al., 2019; section 3.2.5), the GLODAP mapped data product (Lauvset et al., 2016; section 3.2.5), and discrete measurements of oxygen from select cruises between 2004 and 2022 (section 3.2.6).

## 2.5 Uncertainty estimation

Similar to previous studies that have estimated uncertainty in observation-based biogeochemical data products (e.g., Landschützer et al., 2014; Gregor and Gruber, 2021; Keppler et al., 2020; 2023), we combine uncertainty from three separate sources — measurement, gridding, and algorithm — to estimate uncertainty in GOBAI-O$_2$ (section 3.2.4).

Measurement uncertainty ($u([O_2])_{meas.}$) is attributable to the [O$_2$] observations themselves. For this quantity, gridded [O$_2$] from GOBAI-O$_2$ is multiplied by 3%, which is an estimate based on a few factors: the consistency of the GLODAPv2.2022

dataset, the accuracy of BGC Argo observations, the relative proportion of each data source, and the recognized issue of float oxygen sensor response time. The nominal value for the consistency of the GLODAPv2.2022 cruise dataset is stated to be 1% (Lauvset et al., 2022). The approximate accuracy of BGC Argo float observations is estimated as about 3% of surface $[O_2]$ which is about 4% to 5% of average ocean $[O_2]$, for floats quality-controlled by climatological comparisons (Takeshita et al., 2013) and about 3 μmol kg$^{-1}$ (Johnson et al., 2017; Maurer et al., 2021), which is about 2% of average ocean $[O_2]$, for floats quality-controlled by in-air measurements. We choose a value for measurement uncertainty closer to the float accuracy estimates because float observations outweigh ship observations by about six to one in the dataset used to construct GOBAI-$O_2$. The estimates for the accuracy of float $[O_2]$ sensors may be somewhat optimistic, especially when the floats are crossing large vertical oxygen gradients (i.e. steep oxyclines). This is because the response time of oxygen optodes is not instantaneous (Bittig et al., 2014; 2018a; Bittig and Körtzinger, 2017; Körtzinger, 2017). Without response-time-corrections, which are not widely applied to the BGC Argo oxygen dataset, there is potential for systematic biases where float profiles traverse steep oxyclines, such as in the eastern tropical Pacific Ocean and North Indian Ocean.

Gridding uncertainty ($u([O_2])_{grid.}$) is attributable to using a single $[O_2]$ value to represent a four-dimensional box that is coarser in time and space than the resolution of many processes that influence $[O_2]$. We estimate gridding uncertainty by (1) binning the combined GLODAP and Argo observational dataset to grid cells equal in size to the RG09 grid cells; (2) calculating the standard deviation among the observations in cells with more than ten observations (Fig. A6); (3) fitting a multivariate polynomial regression relating those standard deviations to pressure, potential density anomaly, and bottom depth; and (4) applying that regression to the RG09 grid to compute an estimated standard deviations (i.e., gridding uncertainty) in each grid cell.

Algorithm uncertainty ($u([O_2])_{alg.}$) is attributable to the ML algorithms that estimate $[O_2]$ on the RG09 grid. We estimate algorithm uncertainty using the four-dimensional field of absolute differences between $[O_2]$ from GFDL-ESM4 model output versus GOBAI-$O_2$-ESM4, determined from the GFDL-ESM4 algorithm evaluation exercise described in section 2.3.

The three uncertainty sources were combined in quadrature (assuming independence) to calculate a combined uncertainty estimate for each gridded $[O_2]$ value in GOBAI-$O_2$ ($u([O_2])_{tot.}$):

$$u([O_2])_{tot.} = \sqrt{u([O_2])_{meas.}^2 + u([O_2])_{grid.}^2 + u([O_2])_{alg.}^2} \tag{1}$$

## 3 Results and Discussion

### 3.1 Algorithm evaluation

The evaluation exercises indicated that the ML algorithms trained on the combined GLODAP and Argo observational dataset were effective in their ability to estimate $[O_2]$ and reconstruct seasonal to decadal variability in the global oxygen inventory. Mean offsets ($\Delta[O_2] = [O_2]_{obs/mod} - [O_2]_{est}$) and root mean squared differences (RMSDs) between $[O_2]$ from direct measurements ($[O_2]_{obs}$) or GFDL-ESM4 output ($[O_2]_{mod}$) and $[O_2]$ estimated from ML algorithms ($[O_2]_{est}$) were determined as an assessment of the ability of the algorithms to estimate $[O_2]$ at a grid-cell level (Table 2; Fig. 2; Tables B2– B4). Mean $\Delta[O_2]$ and RMSD determined using $[O_2]_{est}$ from the ESPER-Mixed model (Carter et al., 2021) — an average of predictions from a neural network and moving window multiple linear regression trained on GLODAPv2.2020 data — were also determined as a point of comparison for the observational data-based validation test (Table 2; Fig. A7; Table B4). In the case of the GFDL-ESM4-based validation test, metrics to summarize means, amplitudes, trends, and variability in integrated mean $[O_2]$ values were determined to demonstrate the ability of the GOBAI-$O_2$ method to capture seasonal to decadal scale variability in oxygen at the global scale (Table 3; Fig. 3). The results of each evaluation exercise are discussed in more detail in the following sections.

**Table 2.** Regional and global error statistics (mean $\Delta[O_2]$ and RMSD) for evaluation exercises using the ensemble average (ENS$_{Data-Eval}$) of FNN$_{Data-Eval}$ and RFR$_{Data-Eval}$ algorithms trained on a subset of data from the combined GLODAP and Argo observational dataset and tested with a separate subset of withheld data, or the ensemble average (ENS$_{ESM4-Eval}$) of FNN$_{ESM4-Eval}$ and RFR$_{ESM4-Eval}$ algorithms trained on a subset of output from GFDL-ESM4 (corresponding to locations of available Argo and GLODAP data) and tested using the full field of GDFL-ESM4 output. Error statistics calculated using the ESPER-Mixed model are also shown for comparison to the data-based test. The numbers of data points used in the training and assessment of each algorithm are shown.

| Basin | Evaluation Exercise with Observational Data | | | | | | Evaluation Exercise with GFDL-ESM4 Output | | | |
| | | | ENS$_{Data-Eval}$ | | ESPER-Mixed | | | | ENS$_{ESM4-Eval}$ | |
| | Training Data Points | Assessment Data Points | Mean $\Delta[O_2]$ ($\mu$mol kg$^{-1}$) | RMSD ($\mu$mol kg$^{-1}$) | Mean $\Delta[O_2]$ ($\mu$mol kg$^{-1}$) | RMSD ($\mu$mol kg$^{-1}$) | Training Data Points | Assessment Data Points | Mean $\Delta[O_2]$ ($\mu$mol kg$^{-1}$) | RMSD ($\mu$mol kg$^{-1}$) |
|---|---|---|---|---|---|---|---|---|---|---|
| Atl. | 592,099 | 109,134 | -1.8 | 9.2 | -3.9 | 10.7 | 184,418 | 28,235,064 | -1.3 | 9.6 |
| Pac. | 1,816,367 | 466,788 | 0.6 | 9.9 | -3.3 | 15.6 | 533,208 | 69,369,456 | 0.0 | 7.4 |
| Ind. | 335,768 | 82,491 | 0.9 | 7.1 | -3.0 | 11.6 | 86,060 | 20,736,144 | 0.2 | 7.2 |
| Arc. | 800,328 | 263,873 | -1.4 | 9.1 | -2.4 | 12.1 | 293,540 | 11,547,744 | 0.0 | 4.1 |
| Med. | 214,540 | 33,899 | 3.2 | 11.0 | 2.3 | 21.6 | 32,110 | 1,096,680 | 1.0 | 5.5 |
| N. Sou. | 2,236,153 | 480,846 | 0.2 | 7.2 | -1.4 | 10.0 | 756,444 | 67,626,624 | -0.1 | 4.4 |
| S. Sou. | 1,430,492 | 364,133 | -0.7 | 8.2 | -2.3 | 11.7 | 519,610 | 31,412,472 | 0.0 | 3.3 |
| Global | 7,425,747 | 1,801,164 | -0.2 | 8.8 | -2.6 | 13.1 | 2,405,390 | 230,024,184 | -0.2 | 6.7 |

### 3.1.1 Test with withheld observational data

Estimates of $[O_2]$ using ENS$_{Data-Eval}$ algorithms tracked closely with $[O_2]_{obs}$ and showed no strong systematic biases with $[O_2]_{est}$ or depth (Fig. 2a and 2b), though variability in $\Delta[O_2]$ was greatest from just below the surface to about 500 dbars. Mean offsets were between −1.8 and 3.2 $\mu$mol kg$^{-1}$ for the seven regions, with a global average of −0.2 $\mu$mol kg$^{-1}$; RMSDs were between

7.1 and 11.0 µmol kg$^{-1}$ for the seven regions, with a global average of 8.8 µmol kg$^{-1}$ (Table 2). The slightly negative global average offset suggests somewhat higher estimated than measured [O$_2$] values, and some of the lowest RMSDs from the ENS$_{Data-Eval}$ algorithms were found in the Southern Ocean regions (Table 2 and Fig. 2c), likely because these regions have significant amounts of available training data (Figure 1). However, this evaluation exercise is influenced by the incomplete subset of data (20%) used to test the ENS$_{Data-Eval}$ algorithms. A cross-fold validation (e.g., repeating this exercise with five separate 20% chunks of data withheld from algorithm training) was prohibitively computationally expensive. Therefore, the associated Δ[O$_2$] and RMSD values alone are not as instructive as a comparison to the Δ[O$_2$] and RMSD values obtained from the ESPER-Mixed model (Table 2).

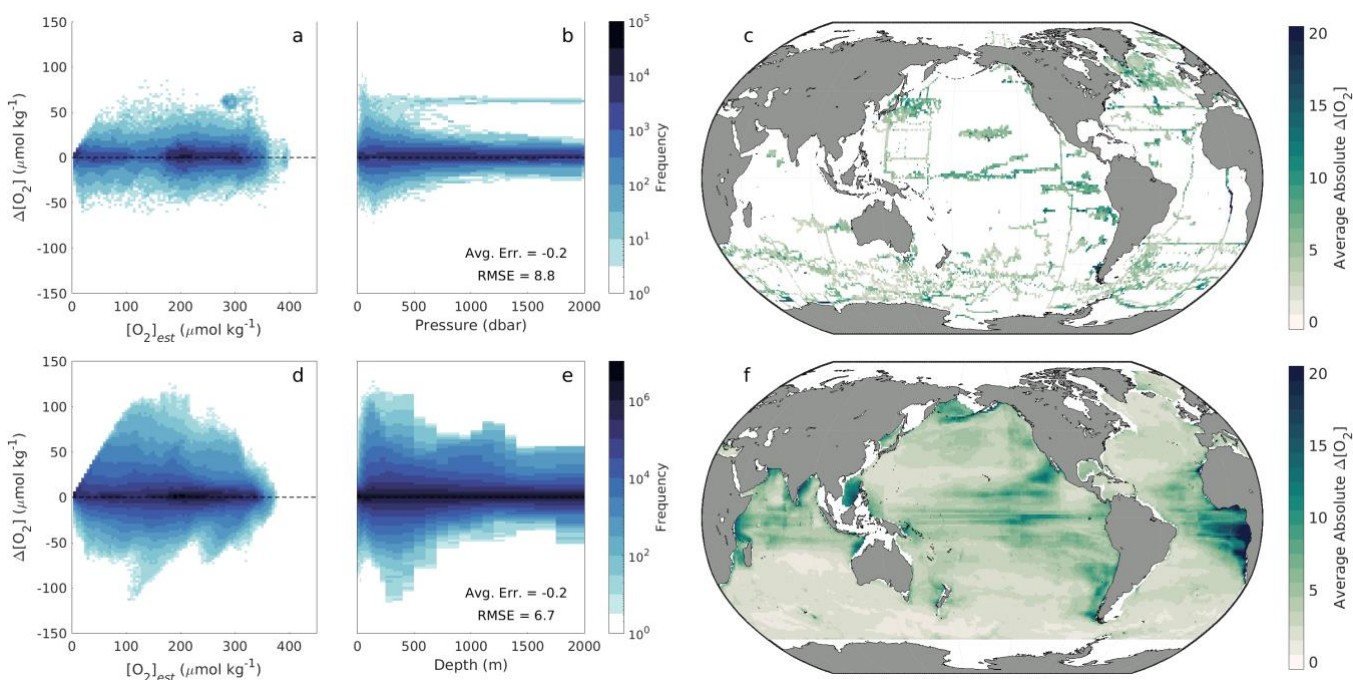

**Figure 2.** (a,b,d,e) Two-dimensional histograms showing offsets between measured versus estimated oxygen (Δ[O$_2$] = [O$_2$]$_{obs}$ − [O$_2$]$_{est}$) for (a,b) withheld observational data and (d,e) modeled versus estimated oxygen (Δ[O$_2$] = [O$_2$]$_{mod}$ − [O$_2$]$_{est}$) for GFDL-ESM4 model output as a function of (a,d) [O$_2$]$_{est}$ and (b,e) depth in the water column. Offsets are binned into cells that are 2.5 µmol kg$^{-1}$ tall in terms of Δ[O$_2$] and (a,d) 5 µmol kg$^{-1}$ wide in terms of [O$_2$]$_{est}$ or equivalent in width to (b) the interpolated pressure levels of the data or (e) the vertical resolution of GFDL-ESM4 grid cells. The frequency of offsets that fall into a given bin is shown on a logarithmic scale, de-emphasizing the significant clustering around Δ[O$_2$] = 0 in favor of showing the few outliers. (c,f) Absolute Δ[O$_2$] values averaged over depth and time for 1° latitude by 1° longitude grid cells in the global ocean for (c) withheld observational data and (f) GFDL-ESM4 model output.

Estimates of [O$_2$] using ESPER-Mixed (Fig. A7) showed average offsets between −3.9 and −1.4 µmol kg$^{-1}$ for the seven regions (with a global average of −2.6 µmol kg$^{-1}$) and RMSDs between 10.0 and 21.6 µmol kg$^{-1}$ for the seven basins (with a global average of 13.1 µmol kg$^{-1}$) (Table 2). Again, the negative global average offset suggests higher estimated than measured

[O$_2$] values. Compared to ESPER-Mixed (Carter et al., 2021), the ENS$_{Data-Eval}$ algorithms performed better, both in terms of Δ[O$_2$] and RMSD in each individual region and overall. This result is likely a reflection of the fact that ENS$_{Data-Eval}$ algorithms were trained with more varied data than the ESPER-Mixed model (Argo and GLODAP compared to just GLODAP), and that the withheld data for which estimates were made also comprised more varied data (both Argo and GLODAP as well).

Importantly, when estimates were made for just the GLODAP dataset, the ENS$_{Data-Eval}$ algorithms still performed better than ESPER-Mixed (Table B4), suggesting that the seasonally-resolved float data supply important information to the relationships established during algorithm training.

### 3.1.2 Test with GFDL-ESM4 output

As introduced in section 2.3, we refer to the four-dimensional field of [O$_2$]$_{est}$ values calculated by applying ENS$_{ESM4-Eval}$
algorithms to GFDL-ESM4 output as GOBAI-O$_2$-ESM4. [O$_2$]$_{est}$ values from GOBAI-O$_2$-ESM4 tracked closely with [O$_2$]$_{mod}$ and showed no significant systematic biases with [O$_2$] or depth (Fig. 2d and 2e). Similar to the data-based test, variability in Δ[O$_2$] was greatest from just below the surface to about 500 meters. Average offsets were between −1.3 and 1.0 μmol kg$^{-1}$ for the seven regions (with a global average of −0.2 μmol kg$^{-1}$) and RMSDs were between 3.3 and 9.6 μmol kg$^{-1}$ for the seven basins (with a global average of 6.7 μmol kg$^{-1}$) (Table 2). The near-zero global average offset suggests that [O$_2$]$_{est}$ values from
GOBAI-O$_2$-ESM4 matched well with values from GFDL-ESM4 output. The lowest RMSDs were found in the Southern Ocean and Arctic regions (Table 2; Fig. 2f), again due to the high density of training data in these regions.

In addition to direct comparisons of [O$_2$] values, GOBAI-O$_2$-ESM4 effectively captured local decadal scale and seasonal variability in [O$_2$] in the GFDL-ESM4 model environment (Fig. 3; Fig. A8–A10; Table 3). The average Pearson's correlation
coefficient between gridded monthly mean [O$_2$] integrated from 0 to 200 dbars from GFDL-ESM4 output versus GOBAI-O$_2$-ESM4 was 0.92 ± 0.17 (Fig. 3b), and the seasonal amplitudes differed in magnitude (GFDL-ESM4 minus GOBAI-O$_2$-ESM4) by 1.8 ± 4.0 μmol kg$^{-1}$ (Fig. 3c). The average Pearson's correlation coefficient between gridded annual mean [O$_2$] integrated from 200 to 1000 dbars from GFDL-ESM4 output versus GOBAI-O$_2$-ESM4 was 0.66 ± 0.37 (Fig. 3e), and the trends differed in magnitude (GFDL-ESM4 minus GOBAI-O$_2$-ESM4) by −0.3 ± 2.1 μmol kg$^{-1}$ decade$^{-1}$ (Fig. 3f).

When considered on the global scale, mean values, seasonal cycle amplitudes, long-term trends, and interannual variabilities in [O$_2$] matched well between GFDL-ESM4 output and GOBAI-O$_2$-ESM4 (Table 3). In almost every case, agreement was far better than it was when simply considering GFDL-ESM4 grid cells for which observations are available over this time period, with no spatiotemporal interpolation. For example, the trend in monthly mean [O$_2$] integrated from 0 to 2000 dbars was −0.38
380 μmol kg$^{-1}$ decade$^{-1}$ for GFDL-ESM4 output versus −0.18 μmol kg$^{-1}$ decade$^{-1}$ for GOBAI-O$_2$-ESM4 (Fig. A11). On the other hand, grid cells where observations are available actually indicated an increase in monthly mean [O$_2$] integrated from 0 to 2000 dbars of 7.3 μmol kg$^{-1}$ decade$^{-1}$ over this time period when no spatiotemporal interpolation is applied.

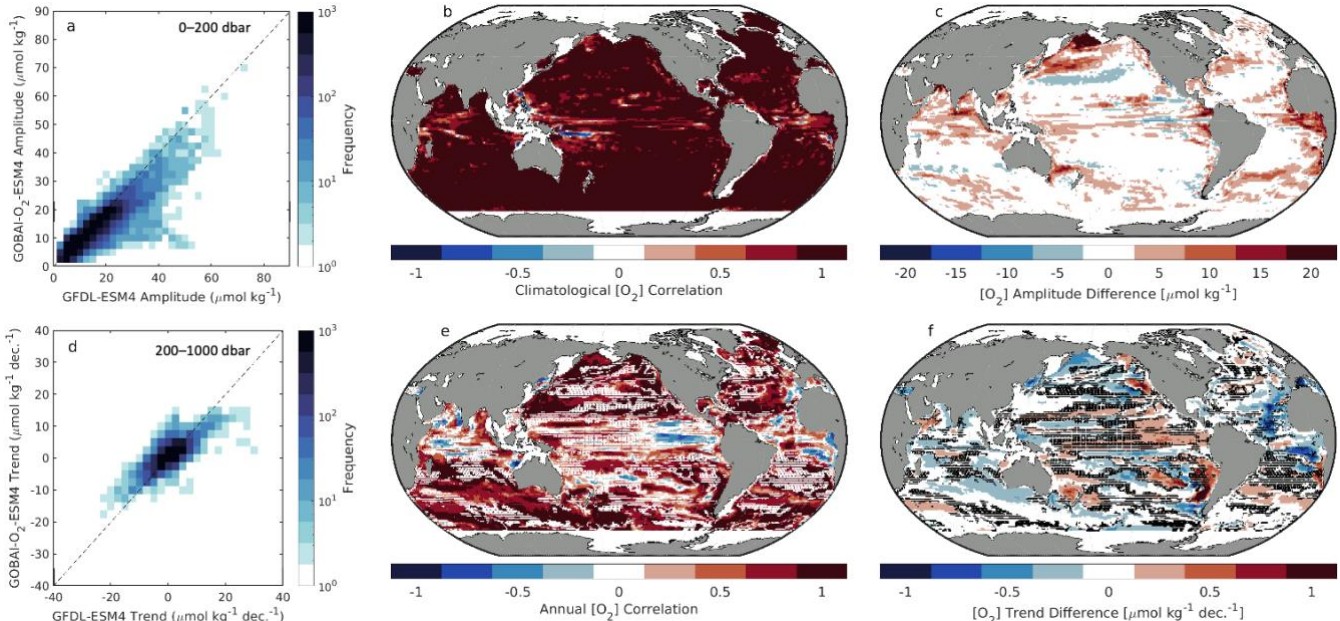

**Figure 3.** (a,d) Two-dimensional histograms showing grid cell level (a) climatological seasonal amplitudes in monthly mean [O$_2$] (weighted means according to the size of each pressure layer) from 0 to 200 dbars and (d) trends in annual mean [O$_2$] from 200 to 1000 dbars between GFDL-ESM4 and GOBAI-O$_2$-ESM4. (b,e) Pearson's correlation coefficients between GFDL-ESM4 and GOBAI-O$_2$-ESM4 for (b) monthly mean [O$_2$] from 0 to 200 dbars, showing coherence between the surface seasonal cycles, and (e) annual mean [O$_2$] from 200 to 1000 dbars, showing coherence between the subsurface trends. (c,f) Absolute difference between GFDL-ESM4 and GOBAI-O$_2$-ESM4 for (c) climatological seasonal amplitudes in monthly mean [O$_2$] from 0 to 200 dbars and (f) trends in annual mean [O$_2$] from 200 to 1000 dbars. (e,f) Stippling indicates grid cells in which the GFDL-ESM4 trend is not significantly different from zero.

Whether the internal variability in GFDL-ESM4 is truly representative of the ocean or is biased in one or more dimensions, the success of GOBAI-O$_2$-ESM4 in this evaluation exercise demonstrates an ability for the ML algorithms employed here to capture that variability with the current distribution of available [O$_2$] observations as training data. This bodes well for the ability of GOBAI-O$_2$, which is trained on actual observational data, to represent decadal scale and seasonal variability in global ocean oxygen in the real world. However, the GFDL-ESM4 output has undergone substantial spatial and temporal averaging and has no observational uncertainties, and thus the assessed skill can be thought of as an upper limit of the reconstruction skill achievable with the currently available observations.

The results of the exercise with GFDL-ESM4 model output are critical for evaluating the uncertainty of gridded oxygen values in GOBAI-O$_2$ (section 3.2.4). Further, the spatial distribution of Δ[O$_2$] (Fig. 2f) and the comparisons of reconstructed to modeled decadal trends and seasonal variability (Fig. 3b, 3c, 3e, 3f; Fig. A8–A10) can help inform our observing efforts (e.g., future cruise planning and BGC Argo float deployments). For example, large Δ[O$_2$] values in the eastern tropical Pacific and eastern tropical Atlantic, coupled with some negative correlations in annual mean [O$_2$] and differences in annual trends and

seasonal amplitudes, suggest more observations will be required for GOBAI-O$_2$ (or likely any observation-based gap-filled [O$_2$] data product) to fully capture variability in that region.

**Table 3.** Statistics representing the mean values, seasonal cycle amplitudes, long-term trends, and interannual variabilities of [O$_2$] from the GFDL-ESM4 model, a reconstruction of [O$_2$] fields from GFDL-ESM4 using the approach of GOBAI-O$_2$ (GOBAI-O$_2$-ESM4), and subsampled grid cells from GFDL-ESM4 where and when real observations are available. Global weighted means (μ) of grid-cell level values are shown, along with differences (Δ) between the fully resolved GFDL-ESM4 means versus GOBAI-O$_2$-ESM4 and versus the subsampled GFDL-ESM4 grid cells.

| Metric | Pressure Layer (dbar) | GFDL-ESM4 μ | GOBAI-O$_2$-ESM4 μ | GOBAI-O$_2$-ESM4 Δ | Subsampled GFDL-ESM4 μ | Subsampled GFDL-ESM4 Δ |
|---|---|---|---|---|---|---|
| Mean [O$_2$] (μmol kg$^{-1}$) | 0−200 | 214.02 | 214.31 | −0.29 | 230.12 | −16.11 |
| | 200−1000 | 154.83 | 155.19 | −0.36 | 173.54 | −18.70 |
| | 0−2000 | 155.59 | 155.82 | −0.23 | 169.63 | −14.03 |
| Seasonal Cycle Amplitude (μmol kg$^{-1}$) | 0−200 | 12.04 | 10.22 | 1.82 | 12.20 | −0.16 |
| | 200−1000 | 3.37 | 2.06 | 1.31 | 5.99 | −2.62 |
| | 0−2000 | 2.60 | 1.84 | 0.75 | 3.94 | −1.34 |
| Long−term Trend (μmol kg$^{-1}$ dec.$^{-1}$) | 0−200 | −0.30 | −0.13 | −0.17 | 7.10 | −7.40 |
| | 200−1000 | −0.48 | −0.23 | −0.26 | 5.22 | −5.70 |
| | 0−2000 | −0.38 | −0.18 | −0.20 | 7.28 | −7.66 |
| Interannual Variability (μmol kg$^{-1}$) | 0−200 | 0.22 | 0.20 | 0.03 | 8.95 | −8.73 |
| | 200−1000 | 0.29 | 0.18 | 0.12 | 10.43 | −10.13 |
| | 0−2000 | 0.22 | 0.12 | 0.10 | 10.29 | −10.07 |

## 3.2 GOBAI-O$_2$ product

### 3.2.1 Spatial oxygen distribution

The full GOBAI-O$_2$ product is available at https://doi.org/10.25921/z72m-yz67 (Sharp et al., 2022; last access: 07 Jul. 2023). Vertical–meridional sections of oxygen (Figs. 4 and 5) show that surface oxygen concentrations are generally high, as these waters tend to be near equilibrium with the atmosphere. This is particularly true at high latitudes where cold, dense waters have a high capacity for dissolved oxygen. Southern Ocean surface waters, however, are generally undersaturated with respect to oxygen (Fig. A12), consistent with observations from previous studies that suggest this undersaturation is the result of O$_2$-depleted thermocline water upwelling into the mixed layer (Chierici et al., 2004; Reuer et al., 2007; Jonsson et al., 2013) making the Southern Ocean on average an oxygen sink (Gruber et al., 2001; Bushinsky et al., 2017). This phenomenon can also be seen in the equatorial Pacific (Fig. A12). Undersaturation in high-latitude regions that are ice-covered during parts of the year can also be the result of limited air sea gas exchange when sea ice is present.

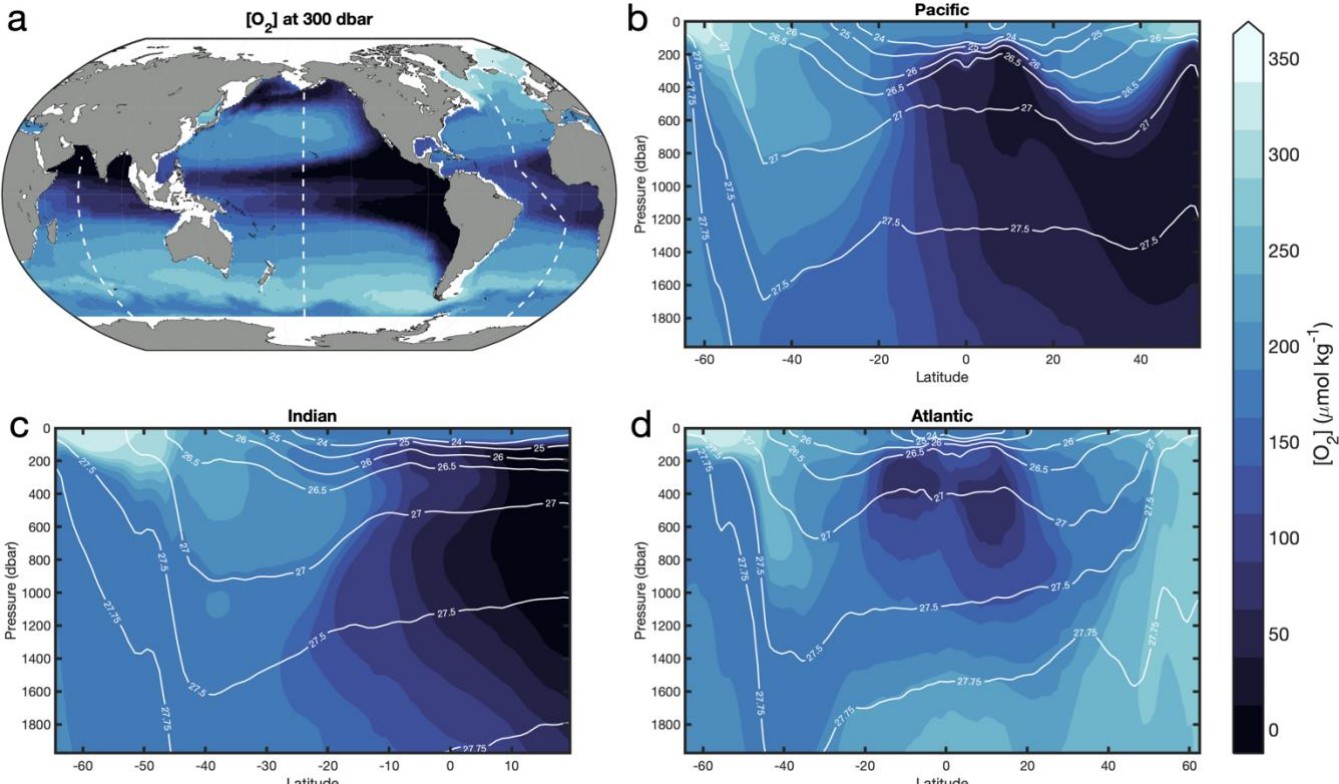

**Figure 4.** Long-term mean [O₂] from GOBAI-O₂ at (a) 300 dbars and from the surface to 2000 dbars in the (b) Pacific, (c) Indian, and (d) Atlantic Oceans. White dashed lines in panel a show the locations of the sections in panels b–d. White contour lines in panels b–d are potential isopycnals (kg m$^{-3}$).

Isobaric maps, isopycnal maps, and vertical–meridional sections with pressure and density vertical coordinates (Figs. 4 and 5) also reveal the [O₂] signatures of distinct subsurface water masses. In each basin, well-ventilated subtropical mode waters can be identified by relatively high [O₂] at mid-latitudes on the 300 dbar surface (Fig. 4a) and along dips in isopycnals plotted against pressure and latitude (Fig. 4b–d) or along sloping isobars plotted against density and latitude (Fig. 5b–d) within the upper ~500 dbars. Beneath the southern mode waters in each basin, Antarctic Intermediate Water that originates in the Southern Ocean with a relatively high [O₂] signal is prevalent. Beneath northern mode waters in the Pacific and Indian basins, respectively, relatively old and oxygen-poor North Pacific Intermediate Water (NPIW) and Red Sea Overflow Water (RSOW) can be observed (Talley et al., 2011). Beneath northern mode waters in the Atlantic, intermediate waters are younger and more highly oxygenated. Near the equator, subsurface oxygen minima are visible in each basin; this is a result of organic matter export from high production in the surface ocean that fuels strong subsurface respiration and relatively poor ventilation (old waters) in this region. Finally, the signatures of higher oxygen deep or bottom waters can be observed near the bottom or at high latitudes in each vertical–meridional section.

Oxygen concentrations at 300 dbars (Fig. 4a) are highest in the North Atlantic and Southern Oceans — where highly oxygenated, newly formed deep and intermediate waters are formed — and lowest in the North and Equatorial Pacific Ocean and the North Indian Ocean — where the oxygen content of subsurface waters has been greatly reduced by heterotrophic respiration over time. The same can be said for [$O_2$] on the 27.0 kg m$^{-3}$ $\sigma_\theta$ surface (Fig. 5a). Oxygen concentrations are extremely low in the deep North Pacific Ocean (Figs. 4b and 5b) and North Indian Ocean (Figs. 4c and 5c) due to the accumulated effects of oxygen-depleting respiration over the long lifespans of those water masses (i.e., long time since gas exchange with the atmosphere).

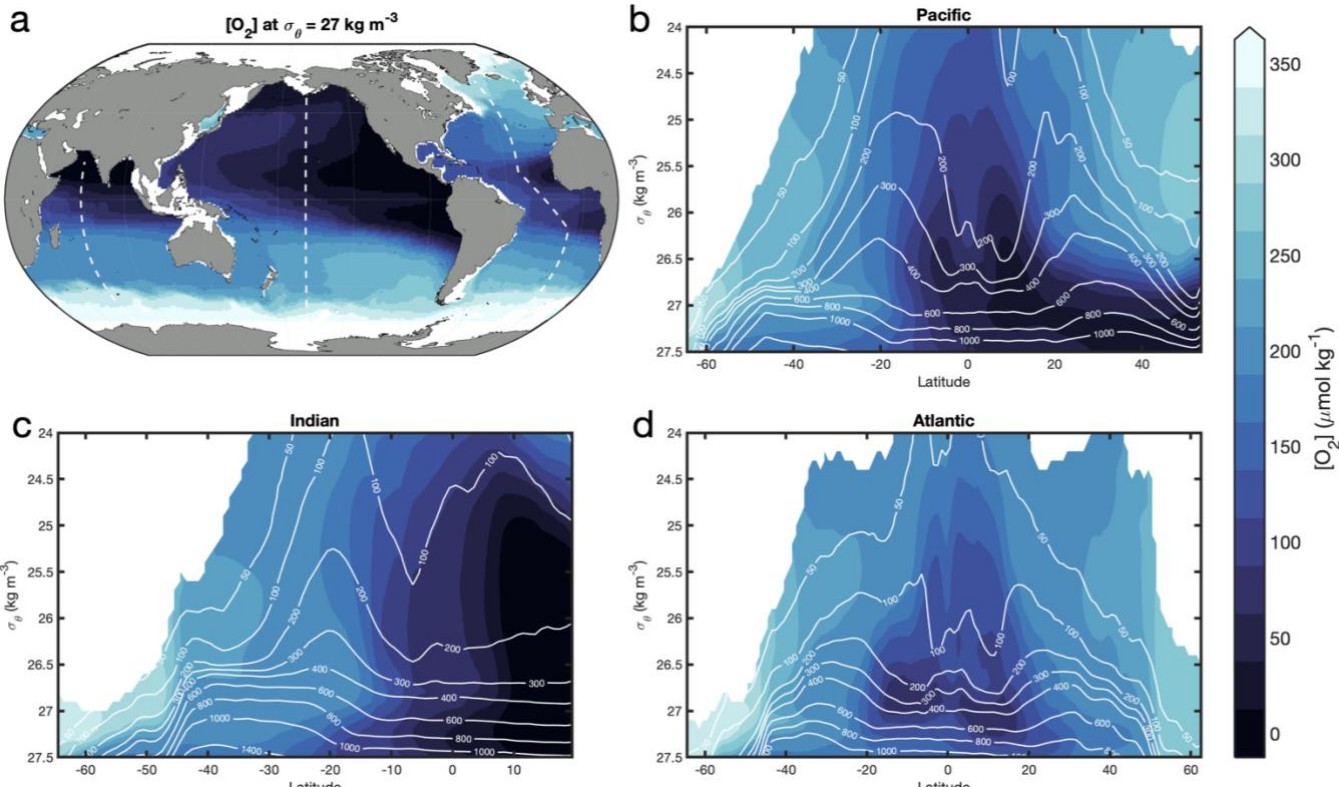

**Figure 5.** Long-term mean [$O_2$] from GOBAI-$O_2$ at (a) $\sigma_\theta$ = 27 kg m$^{-3}$ and from $\sigma_\theta$ = 24 to 27.5 kg m$^{-3}$ in the (b) Pacific, (c) Indian, and (d) Atlantic Oceans. White dashed lines in panel a show the locations of the sections in panels b–d. White contour lines in panels b–d are constant isobars (dbars).

### 3.2.2 Climatological seasonal oxygen cycles

Seasonal cycles in [$O_2$] reflect a balance among physical and biological processes (Wang et al., 2022). Climatological hemispheric mean [$O_2$] integrated over three pressure layers from GOBAI-$O_2$ (Fig. 6) reveals that the magnitude of the [$O_2$] seasonal cycle is greatest near the surface and decreases with depth. The amplitude of the [$O_2$] seasonal cycle in a near-surface

layer (0–100 dbars) is about 10.7 µmol kg$^{-1}$ in the Northern Hemisphere and 8.9 µmol kg$^{-1}$ in the Southern Hemisphere. Maximum [O$_2$] in this pressure layer (April/May in the Northern Hemisphere and October/November in the Southern Hemisphere) lags about two months behind the temperature minimum, suggesting an interaction between a thermally driven increase in oxygen solubility and biologically driven oxygen production. Minimum [O$_2$] in the near-surface layer (October in the Northern Hemisphere and March/April in the Southern Hemisphere) is more coincident with the temperature maximum, indicating primary control by a thermally driven decrease in oxygen solubility. The amplitude of the [O$_2$] seasonal cycle is about 2.4 µmol kg$^{-1}$ in the Northern Hemisphere and 2.6 µmol kg$^{-1}$ in the Southern Hemisphere in the intermediate layer (100–600 dbars), and about 0.2 µmol kg$^{-1}$ in the Northern Hemisphere and 0.1 µmol kg$^{-1}$ in the Southern Hemisphere in the deep layer (600–2000 dbars). The timing of maximum [O$_2$] values is similar between the near-surface layer and intermediate layer in both hemispheres, indicating the well-mixed nature of the ocean in winter and early spring when [O$_2$] is high. On the other hand, minimum [O$_2$] in the intermediate layer lags behind that in the near-surface layer in both hemispheres, possibly reflecting higher stratification in the upper ocean when temperatures are warmer and/or the remineralization of sinking organic matter after summer production. Further analysis of climatological [O$_2$] cycles from GOBAI-O$_2$ can provide insight into the physical and biological factors that control surface and subsurface oxygen on regional and global scales.

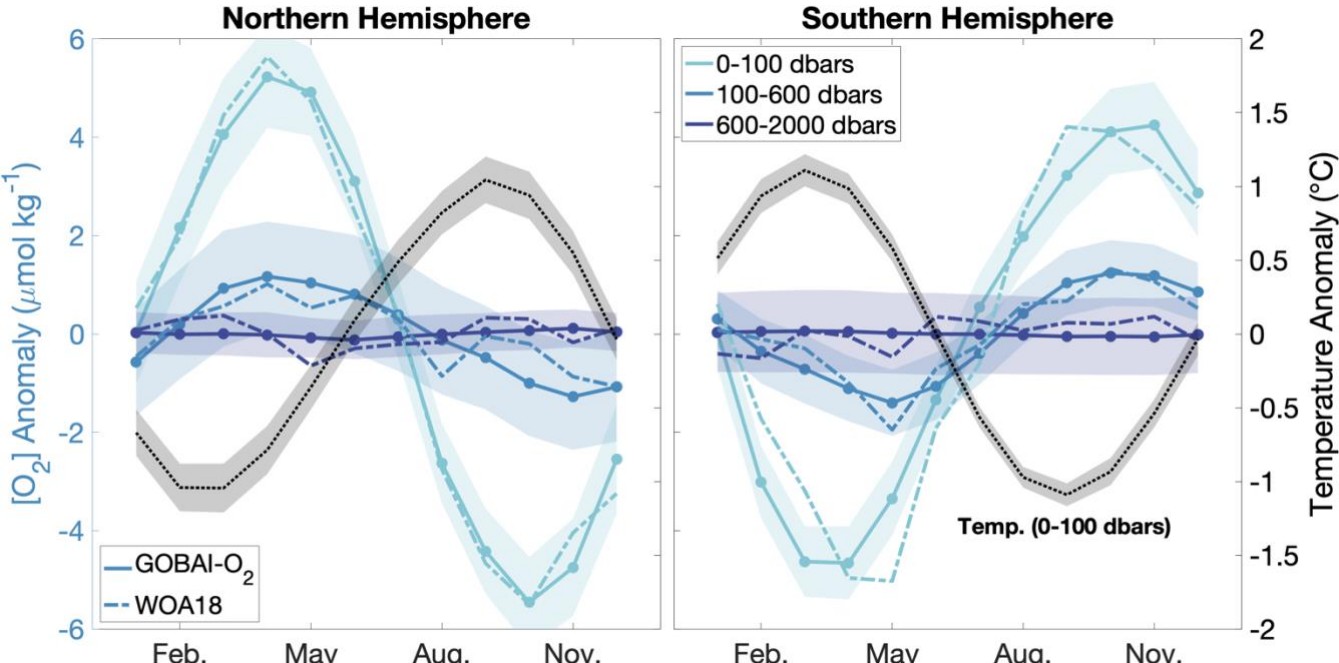

**Figure 6.** Climatological seasonal cycles of [O$_2$] anomalies (monthly [O$_2$] minus long-term mean [O$_2$]) integrated globally over three pressure layers: 0–100, 100–600, and 600–2000 dbars. The black dotted line shows climatological temperature anomaly integrated globally over the 0–100 dbar layer. Shading indicates the standard deviation of the climatological seasonal cycle from 2004 to 2022. The dashed lines show climatological seasonal cycles of [O$_2$] anomalies from WOA18 over similar depth layers to GOBAI-O$_2$: 0–100, 100–600, and 600–1500 meters.

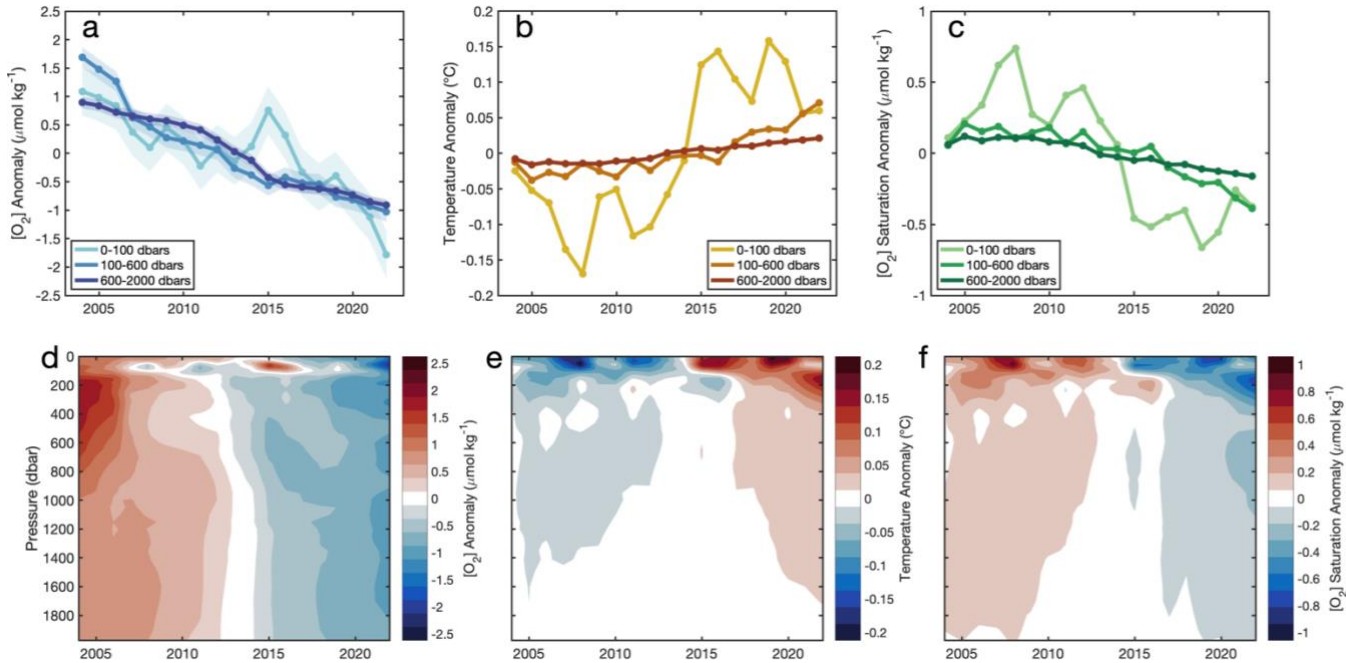

**Figure 7.** (a,b,c) Annual mean (a) $[O_2]$ anomalies from GOBAI-$O_2$, (b) temperature anomalies from RG09, and (c) $[O_2]_{sat.}$ anomalies calculated from RG09 temperature and salinity fields, each integrated globally over three pressure layers: 0–100, 100–600, and 600–2000 dbars. (a) Shading represents uncertainty determined as the average difference between mean $[O_2]$ from GOBAI-$O_2$-ESM4 versus GFDL-ESM4 in each layer. (d,e,f) Hovmöller diagrams showing annual mean (d) $[O_2]$

anomalies from GOBAI-$O_2$, (e) temperature anomalies from RG09, and (f) $[O_2]_{sat.}$ anomalies calculated from RG09 temperature and salinity fields, each versus pressure in decibars and time from 2004 to 2022. Anomalies in each parameter are calculated as annual mean values minus the long-term mean either (a–c) integrated over a pressure layer or (d–f) on a given pressure level.

**3.2.3 Interannual oxygen trends and variability**

Deoxygenation is evident in GOBAI-$O_2$ over the past two decades, coincident with ocean warming (Fig. 7; Table B5). The spatially weighted rate of deoxygenation in the upper two kilometers globally (along with a 95% confidence interval) is $-1.19 \pm 0.05$ μmol kg$^{-1}$ decade$^{-1}$ ($-0.79 \pm 0.04$ % decade$^{-1}$). The rate of deoxygenation in GOBAI-$O_2$ varies over depth, with a near-surface pressure layer (0–100 dbars) displaying a trend in $[O_2]$ of $-1.10 \pm 0.60$ μmol kg$^{-1}$ decade$^{-1}$ ($-0.49 \pm 0.26$ % decade$^{-1}$),

an intermediate layer (100–600 dbars) $-1.38 \pm 0.42$ μmol kg$^{-1}$ decade$^{-1}$ ($-0.86 \pm 0.26$ % decade$^{-1}$), and a deep layer (600–2000 dbars) $-1.12 \pm 0.54$ μmol kg$^{-1}$ decade$^{-1}$ ($-0.79 \pm 0.38$ % decade$^{-1}$). Interannual variability is greatest in the near-surface layer: when the multi-year trends and seasonal cycles are removed, the standard deviation of annual global mean $[O_2]$ anomalies is 0.52 μmol kg$^{-1}$ in the near-surface layer compared to 0.26 μmol kg$^{-1}$ in the intermediate layer and 0.12 μmol kg$^{-1}$ in the deep layer. Trends and uncertainties were determined by fitting linear least squares models to spatially weighted

monthly mean [$O_2$] and monthly oxygen inventories integrated over the specified pressure layers, with uncertainties in monthly values determined by comparing GOBAI-$O_2$-ESM4 to GFDL-ESM4; more information on this is provided in Appendix E.

Ocean warming has a direct effect on oxygen concentrations by lowering the solubility of $O_2$ in ocean water (Garcia and Gordon, 1992). Solubility changes explain about 56% of deoxygenation in the near-surface ocean layer (0–100 dbars), 20% in the intermediate ocean layer (100–600 dbars), and 15% in the deep ocean layer (600–2000 dbars) (Fig. 7c and 7f). The remaining deoxygenation must then be caused by indirect consequences of ocean warming (such as increased ocean stratification hence decreased subsurface ventilation) or other processes, including changes to oxygen utilization and ocean ventilation variability (Oschlies et al., 2018), the magnitudes of which this analysis does not attempt to deconvolve. The RG09 temperature and salinity fields are constructed such that they relax toward the climatological means during periods of low data density. For this reason, , toward the beginning of the time series when fewer observations are available, temperature is biased somewhat high (Figs. 7b and 7e) and therefore $O_2$ solubility biased somewhat low (Figs. 7c and 7f). This artifact may influence GOBAI-$O_2$ (Figs. 7a and 7d) since it was constructed using the RG09 temperature and salinity fields; however, its influence is partially mitigated because temporal information included in the training and application of the GOBAI-$O_2$ algorithms allows for the trend inherent to the underlying oxygen data to be retained.

GOBAI-$O_2$ trends can be viewed in the context of other recent analyses that explore long term changes in ocean oxygen. From the surface to 1000 dbars, the GOBAI-$O_2$ trend of $-0.82 \pm 0.11$ % decade$^{-1}$ from 2004–2022 is larger than that assessed by Bindoff et al. (2019) of $-0.48 \pm 0.35$ % decade$^{-1}$ from 1970–2010 (surface to 1000 meters), which takes into account estimates from Helm et al. (2011) ($-0.44 \pm 0.14$ % decade$^{-1}$), Schmidtko et al. (2017) ($-0.34 \pm 0.35$ % decade$^{-1}$), and Ito et al. (2017) ($-0.68 \pm 0.33$ % decade$^{-1}$). In the surface layer (0–100 dbars), the GOBAI-$O_2$ trend of $-0.49 \pm 0.26$ % decade$^{-1}$ can be compared to the Bindoff et al. (2019) assessment of $-0.28 \pm 0.24$ % decade$^{-1}$; in the intermediate layer (100–600 dbars), the GOBAI-$O_2$ trend of $-0.86 \pm 0.26$ % decade$^{-1}$ can be compared to the Bindoff et al. (2019) assessment of $-0.52 \pm 0.36$ % decade$^{-1}$. Considering that these comparisons represent different periods of time such that one should not expect perfect agreement, we find the results encouraging. The somewhat more negative GOBAI-$O_2$ trends compared to previous estimates suggest a possible acceleration of ocean deoxygenation over the last decade or so, which would be consistent with expectations (Kwiatkowski et al., 2020). Further, agreement between GOBAI-$O_2$ and other observation-based studies provides additional support for the notion that current ESMs, which exhibit weaker deoxygenation trends (see section 3.1.2), may not fully capture the sensitivities of physical and biological processes leading to deoxygenation (Oschlies et al., 2017; 2018; Stramma and Schmidtko, 2021). This comparison not only places the GOBAI-$O_2$ trends in a longer term context but suggests that the enhanced observations and analysis result in a reduced trend uncertainty despite the comparatively-shorter 19-year record ($\pm0.04$ % decade$^{-1}$) versus the longer but more sparse 40-year records assessed by Bindoff et al. ($\pm0.14$ to $\pm0.35$ % decade$^{-1}$; 2019).

The trends presented here represent both natural and potentially anthropogenic variability over the interval between 2004 and 2022, as well as uncertainties in the algorithm predictions (see section 3.2.4). As such, these trends should not be interpreted to be driven exclusively by ocean warming and other associated impacts of anthropogenic climate change; the period of time examined is relatively short and the domain is not inclusive of the entire global ocean. Accordingly, decadal-scale variability in ocean ventilation, interior circulation, and biological oxygen utilization may exert significant influence over these trends. This is especially true of the regional trends. Finally, a sensitivity test indicated that including versus withholding temporal predictors did not significantly impact the global [$O_2$] trend. However, the shift from a relative dominance by shipboard observations during the early portion of the GOBAI-$O_2$ timespan to a relative dominance by float observations during the later portion cannot be ignored a potential contributor to a deoxygenation trend, especially considering the potential for systematic biases in the float [$O_2$] dataset (see section 2.5). This kind of shift in measurement platforms has precedent for producing spurious trends in oceanographic observational datasets (Rykaczewski and Dunne, 2011).

### 3.2.4 Uncertainty

GOBAI-$O_2$ uncertainty fields, which were estimated as described in section 2.5, can be used to assess confidence in multi-year trends and seasonal cycles of [$O_2$], both on a global and regional scale. Time-averaged uncertainty fields at 150 dbar (Fig. 8) suggest that the largest driver of geographic variability in uncertainty is the algorithm uncertainty. Averaged globally over space and time, $u([O_2])_{meas.}$ was equal to 4.5 µmol kg$^{-1}$ (5.6 µmol kg$^{-1}$ on the 150 dbar level), $u([O_2])_{grid.}$ was equal to 3.1 µmol kg$^{-1}$ (5.3 µmol kg$^{-1}$ on the 150 dbar level), and $u([O_2])_{alg.}$ was equal to 3.8 µmol kg$^{-1}$ (6.3 µmol kg$^{-1}$ on the 150 dbar level). Combined, $u([O_2])_{tot.}$ (Eq. 1) was equal to 7.6 µmol kg$^{-1}$ (11.2 µmol kg$^{-1}$ on the 150 dbar level), which can be compared to the global average RMSD of 8.8 µmol kg$^{-1}$ determined independently by withholding data from algorithm training (Table 2, Fig 2a-2c).

Measurement uncertainty provides an estimate of confidence in an [$O_2$] value assigned to a water sample by direct measurement; gridding uncertainty provides an estimate of confidence that the [$O_2$] value provided for a four-dimensional grid cell might represent [$O_2$] at any point in time and space within that grid cell; and algorithm uncertainty provides an estimate of confidence that the predicted [$O_2$] value for a given grid cell is appropriate as the average value for that grid cell. Algorithm uncertainty in particular depends upon the distribution of data available to train the ML algorithms and the ability of the trained algorithms to represent underlying variability in the system. On the isobar shown in Fig 8. (150 dbar), the underlying variability is relatively high in [$O_2$] minimum zones (e.g., near the equator and on the eastern boundaries of ocean basins), hence the elevated algorithm (and total) uncertainties in those regions. Here, algorithm uncertainty was assessed via the exercise with synthetic data from GFDL-ESM4 (see sections 2.3 and 3.1.2).

Algorithm uncertainty should in general decrease as the spatiotemporal coverage of available training data increases. Regionally, algorithm uncertainty depends upon the degree to which the underlying variability of the system is captured by

the available training observations and the ability of the ML algorithms to reconstruct that variability from concurrent measurements of other seawater properties. Comparing the $\Delta[O_2]$ map in Fig. 2f or the algorithm uncertainty map in Fig. 8c to the data distribution map in Fig. 1 or Fig. A1 suggests that sparse sampling is primarily to blame for limitations related to algorithm uncertainty. Detailed analysis of GFDL-ESM4 water mass characteristics in the California Current System has also revealed that high uncertainties occur where water masses with similar physical characteristics but different oxygen signatures mix, underscoring that the measurement of additional biogeochemical parameters can supplement the physical/spatiotemporal-based $[O_2]$ estimates presented here. Overall, the significant influence of algorithm uncertainty is consistent with uncertainty analyses conducted for gap-filling methods applied to other ocean biogeochemical variables (e.g., Landschützer et al., 2014; Gregor and Gruber, 2020). For this reason, continued expansion of oxygen observations in undersampled regions will be critical to reduce uncertainty in our gap filling, and ultimately our understanding, of global subsurface oxygen distributions and variability. Similarly, the significant influence of measurement uncertainty underscores the importance of continued development of oxygen sensor calibration (Bittig et al., 2018a) and data quality control (Maurer et al., 2021) from the evolving BGC Argo fleet.

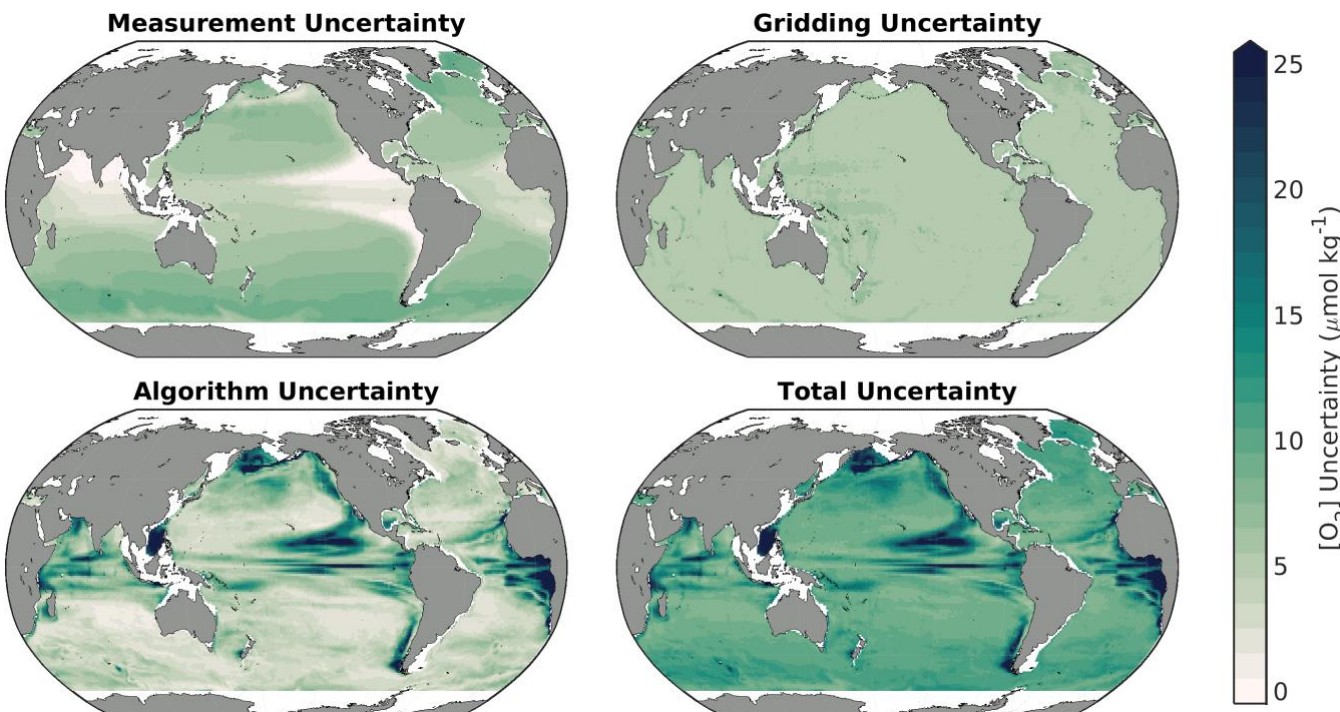

**Figure 8.** Long-term means of the uncertainty contributors to GOBAI-O$_2$ at 150 dbar, including (a) measurement uncertainty, (b) gridding uncertainty, (c) algorithm uncertainty, and (d) total uncertainty.

Global mean depth profiles of uncertainty contributors (Fig. A14) emphasize the general attenuation of uncertainty away from the surface, with subsurface maxima of algorithm uncertainty at 200 dbars and total uncertainty at 100 dbars. The algorithm uncertainty maximum correspond to pressures at which vertical gradients in [$O_2$] are relatively high (see Fig. 4). Here, small variations in the depths of density surfaces can influence [$O_2$] on a given pressure level; this variability is challenging to capture, even with potential density as a predictor variable in the ML models (see Table 1). The total uncertainty maximum

represents this vertical gradient effect balanced against relatively high measurement uncertainty closer to the surface, associated with higher [$O_2$].

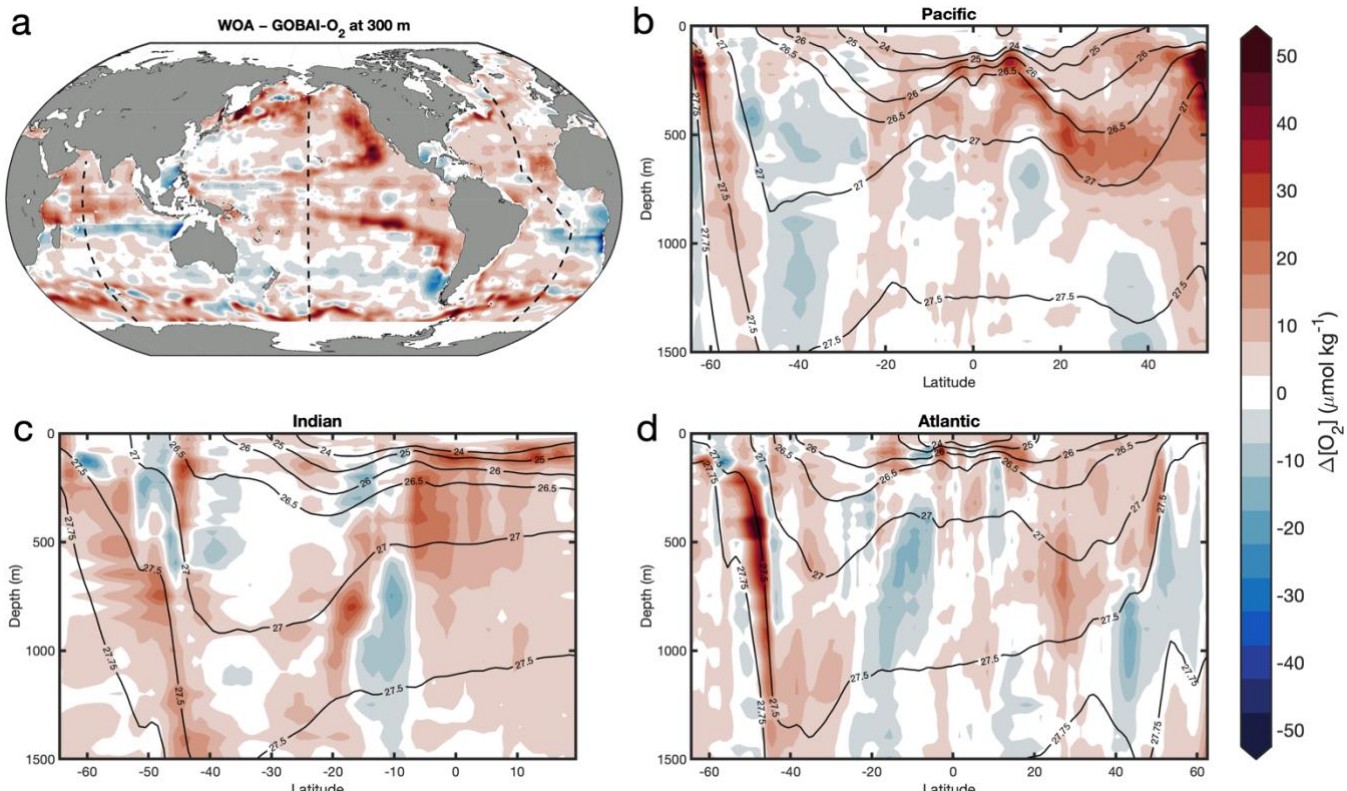

**Figure 9.** The difference between climatological mean [$O_2$] from WOA18 and long-term mean [$O_2$] from GOBAI-$O_2$ ($\Delta$[$O_2$]

= [$O_2$]$_{WOA}$ − [$O_2$]$_{GOBAI}$) at (a) 300 meters and from the surface to 1500 dbars in the (b) Pacific, (c) Indian, and (d) Atlantic Oceans.

### 3.2.5 Comparison to other gridded products

The long-term mean field of [$O_2$] from GOBAI-$O_2$ was compared to the corresponding mean field of [$O_2$] from the WOA18

monthly climatology (Fig. 9) and climatological field of [$O_2$] from the GLODAPv2.2016 mapped product (Fig. 10). On average, GOBAI-$O_2$ oxygen concentration is 1.4 μmol kg$^{-1}$ lower than GLODAP and 9.6 μmol kg$^{-1}$ lower than WOA18. This

can be partly explained by the fact that GOBAI-$O_2$ is centered on the year 2012, whereas observations in GLODAPv2.2016 are centered around 2002 (Lauvset et al., 2016), WOA18 takes into account [$O_2$] observations dating back to 1965, and global deoxygenation has occurred in recent decades (Bindoff et al., 2019). Spatially, the largest differences occur within and especially near the boundaries of oxygen minimum zones (eastern tropical Pacific, eastern Atlantic coastal zones, and northern Indian), along $\sigma \approx 27.5$ kg m$^{-3}$ in the Southern Ocean, and along $\sigma \approx 26.75$ kg m$^{-3}$ in the North Pacific. It is difficult to determine whether these differences are functions of data availability (ship data for WOA18 and GLODAP versus ship and float data for GOBAI-$O_2$), representative time period, or mapping method (objective interpolation for WOA18 and GLODAP versus machine learning algorithms for GOBAI-$O_2$). A future intercomparison exercise between mapping methods using an identical starting dataset could be helpful in diagnosing these differences among gridded products.

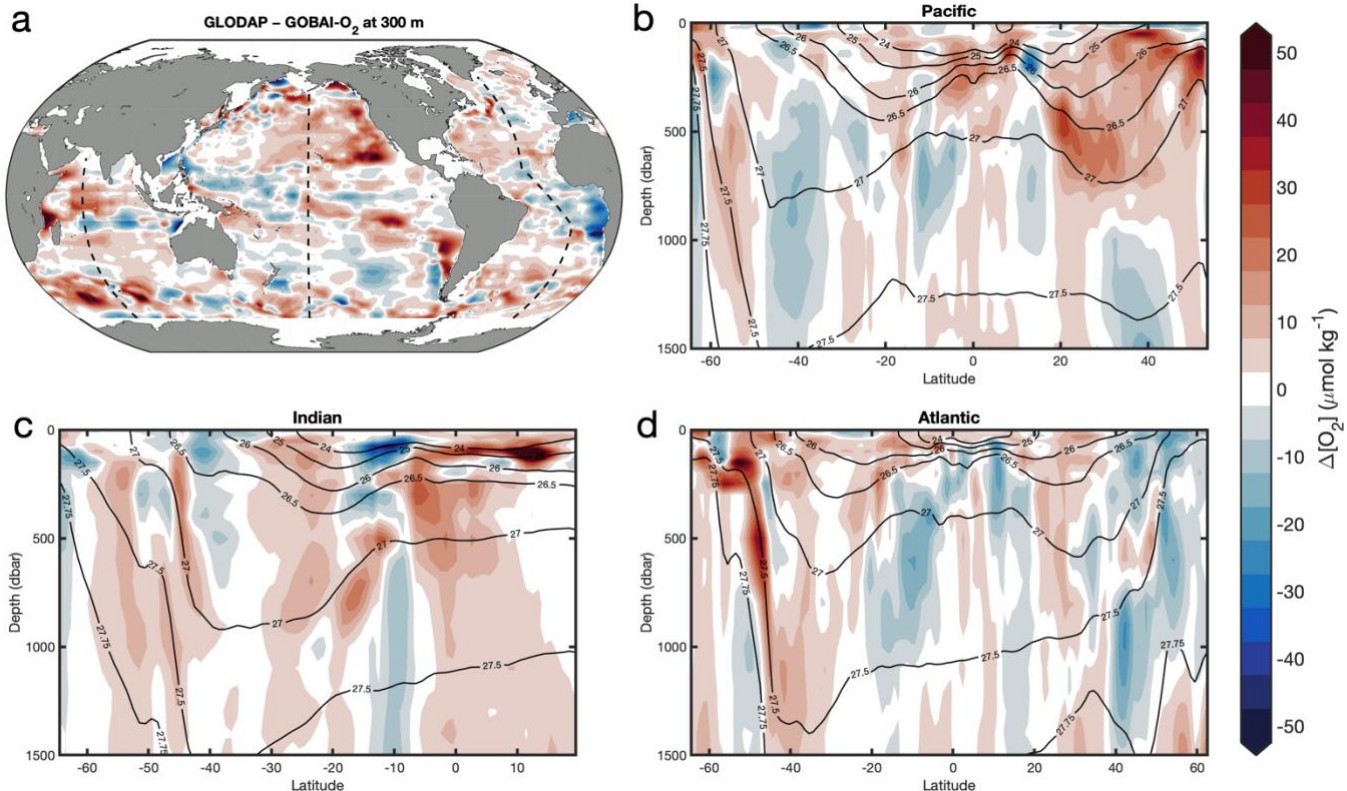

**Figure 10.** The difference between 2002-centered mean [$O_2$] from the GLODAPv2.2016 mapped product and long-term mean [$O_2$] from GOBAI-$O_2$ ($\Delta$[$O_2$] = [$O_2$]$_{GLODAP}$ − [$O_2$]$_{GOBAI}$) at (a) 300 meters and from the surface to 1500 dbars in the (b) Pacific, (c) Indian, and (d) Atlantic Oceans.

### 3.2.6 Comparison to synoptic in situ measurements

GOBAI-O$_2$ was compared to direct observations from repeat hydrography cruises, including meridional transects across the Atlantic (A16 in 2013 and A20 in 2021), Pacific (P16 in 2005), and Indian (I08 and I09 in 2016) Oceans, as well as a zonal transect across the Pacific Ocean (P02 in 2012). This exercise assessed how well monthly [O$_2$] estimates from GOBAI-O$_2$ were able to represent high-quality [O$_2$] measurements at distinct points in time and space. Due to fundamental differences between gridded estimates and point observations, we don't expect every matchup to be perfect. However, we would hope to see general coherence in mean values across large-scale ocean sections and to see a pattern of differences that make sense given our *a priori* expectations.

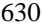

**Figure 11.** Section plots displaying comparisons between discrete observations of [O$_2$] from repeat hydrography cruises and [O$_2$] extracted from corresponding grid cells in GOBAI-O$_2$. Thick lines in each panel represent mixed layer depth calculated as the depth at which potential density anomaly increased to 0.03 kg m$^{-3}$ greater than potential density anomaly at 10 dbars. Thin lines are contours representing increments of 50 μmol kg$^{-1}$ in [O$_2$].

For the cruise datasets examined, GOBAI-O$_2$ estimates matched fairly well with discrete measurements in the mixed layer and below ~1000 dbars (Fig. 11). In intermediate depths, however, large differences occasionally occur. These large differences tended to cluster around areas with strong vertical gradients in [O$_2$] (thin contours in Fig. 11 represent increments of 50 μmol in [O$_2$]). Comparison of Fig. 11 to Fig. A15 gives confidence to our uncertainty evaluation: larger differences between discrete

measurements and GOBAI-$O_2$ occur where $u([O_2])_{tot.}$ is large. Median biases, mean biases, and RMSDs between direct observations and GOBAI-$O_2$ are given in Table B6.

## 4 Conclusions

GOBAI-$O_2$ is a major step toward the fulfilment of the primary goal set out by Gruber et al. (2010): "to determine, on a global-scale, seasonal to decadal time-scale variations in dissolved oxygen concentrations throughout the upper ocean". Quantifying these variations is important for documenting ocean deoxygenation, determining global net primary productivity and carbon export, and facilitating studies of the oceanic uptake of anthropogenic $CO_2$. In addition, insights into ocean biogeochemical dynamics, when observations are unavailable, often come from ocean models, and GOBAI-$O_2$ can bring value to modelling studies by providing fields of $[O_2]$ to be used for boundary conditions and model initialization. GOBAI-$O_2$ can also be useful as a dynamic reference check for new, sensor-based $[O_2]$ measurements that would otherwise be compared to a static monthly climatology like WOA18. Still, users should carefully consider the associated spatial uncertainty fields, especially when conducting regional analyses. Spatial and temporal errors and discontinuities may be significant when GOBAI-$O_2$ is analyzed over small areas, but are mitigated when looking at broader scales.

The uncertainty analysis conducted here confirms that GOBAI-$O_2$ remains limited, largely by sparse sampling and inadequate representation of $[O_2]$ across strong gradients. The most consequential actions to improve GOBAI-$O_2$ fields over the next decade will be the continued deployment of Argo floats with oxygen optodes — emphasizing the importance of bolstering the biogeochemical Argo array and expanding the international OneArgo network into high latitudes, the deep ocean, and marginal seas (Roemmich et al., 2019; 2021; Schofield et al., 2022) — continued work toward ensuring reliable measurements from those optodes, and the continued collection of discrete dissolved oxygen observations — primarily through the international GO-SHIP program — both for use in $[O_2]$ mapping and for calibration/validation of the Argo oxygen data.

Besides these actions, additional steps can be taken to improve GOBAI-$O_2$ fields. For one, more predictor variables and ML algorithms can be tested. Different processes dominate $[O_2]$ variability in different regions (Keeling et al., 2010; Oschlies et al., 2018; Garcia-Soto et al., 2021), and certain predictor variables will be better suited for capturing these processes. Also, ML algorithms adapt to data sparseness and modes of variability in different ways (Ritter et al., 2017; Gregor et al., 2019), so estimates in a given region that are worse using one algorithm may be better using another. Therefore, regionally-tuned predictors and more diverse ensembles of ML algorithms should lead to increased confidence in estimates of ocean interior $[O_2]$. Another action that could result in improved fidelity of GOBAI-$O_2$ fields is the use of predictor variable fields with higher spatial and temporal resolution across sharp biogeochemical gradients. Ocean profiles of temperature and salinity tend to be relatively smooth, so a depth resolution on the order of tens of meters in the upper ocean increasing to hundreds of meters at depth is sufficient for gridded products. Biogeochemical parameters like oxygen, on the other hand, tend to be characterized

by profiles with sharp gradients and with distinct minima and maxima in the water column (Sarmiento and Gruber, 2006). These minima and maxima can occur very near the surface or hundreds of meters below it. For this reason, comparisons of GOBAI-O$_2$ to direct measurements of [O$_2$] can be uniquely problematic in the ~100–1000 dbar range when sharp gradients are present (Fig. 11). A complicating factor is the lack of response-time-corrections applied to float sensor data (section 2.5), which contributes to uncertainty in observation-based [O$_2$] products like GOBAI-O$_2$. Biogeochemical gradients over horizontal space and time can also be sharp, especially in highly dynamic coastal zones and in the surface ocean where the residence time of oxygen is often less than a month (Luz and Barkan, 2000). Recent work from Lyman and Johnson (2023) uses Argo observations coupled with machine learning to provide well-resolved (7-day $\times$ ¼° grid) ocean heat content maps, and continued development toward maps of temperature and salinity could be helpful for overcoming the issue of resolving sharp biogeochemical gradients. Alternatively, [O$_2$] estimates could be made using temperature and salinity observations at their original resolution, then mapped onto four-dimensional grids that are uniquely suited in their spatial resolution for biogeochemical parameters. A necessary consideration of the latter option would be computing resources: applying complex ML algorithms to temperature and salinity measurements from Argo floats at their original resolution may prove to be impractical. Finally, observations from additional platforms could be incorporated into approaches like this one to map [O$_2$] in the global ocean. Ocean gliders and moored profilers have long been equipped with oxygen optodes. These platforms collect data at unique spatiotemporal scales and could add predictive information for [O$_2$] that is not provided by Argo float observations or discrete shipboard measurements. To facilitate the incorporation of new data streams into the development of gridded data products, accessible databases should be created and maintained (Testor et al., 2019; Grégoire et al., 2021).

The method used to develop GOBAI-O$_2$ can be applied in a similar way to other ocean chemical parameters. In addition to dissolved oxygen, the BGC Argo program has deployed floats with sensors for measuring dissolved nitrate, pH, chlorophyll-a, particle backscatter, and downwelling irradiance. Machine learning methods have been used to develop four-dimensional fields of optical properties, i.e. chlorophyll-a and particle backscatter (Sauzède et al., 2015; 2016), and continued refinement of those fields is ongoing (Sauzède et al., 2021). Chemical properties, i.e. nitrate and pH, that exhibit distributions more similar to [O$_2$] are good candidates for adoption into the GOBAI mapping approach. Together with property estimation algorithms for TA (Bittig et al., 2018b; Carter et al., 2021), a mapped ocean interior pH product could be used to resolve the entire ocean carbonate system in four dimensions in near real time.

Ultimately, global changes to the amount of dissolved oxygen in ocean waters will have profound effects on the metabolism of marine organisms (Pörtner and Farrell, 2008; Sampaio et al., 2021) and the cycling of biogeochemically important elements (Gruber, 2004; Berman-Frank et al., 2008). Whereas ocean models agree that the ocean's oxygen inventory has been declining and will continue to decline, disagreement remains as to regional patterns of this deoxygenation. Direct observations are critical for the confirmation or contradiction of model trends. With this work we have turned to autonomous and discrete observations, with the assistance of machine learning algorithms, to bridge the model–observational gap. We produce and analyze a multi-

year gridded product of ocean dissolved oxygen called GOBAI-$O_2$, independently confirming a phenomenon that has been demonstrated previously: the ocean is losing dissolved oxygen at a rapid rate ($0.79 \pm 0.04$ % decade$^{-1}$ in the upper two kilometers according to GOBAI-$O_2$). In addition, we provide this valuable observation-based product for community use. GOBAI-$O_2$ can be turned to as a reference for [$O_2$] observations and model boundary conditions, compared to new and existing observational and model-based reconstructions of ocean deoxygenation, and used for critical analyses of seasonal to decadal and regional to global oxygen variability.

# 5 Appendices

## Appendix A. Supplemental Figures

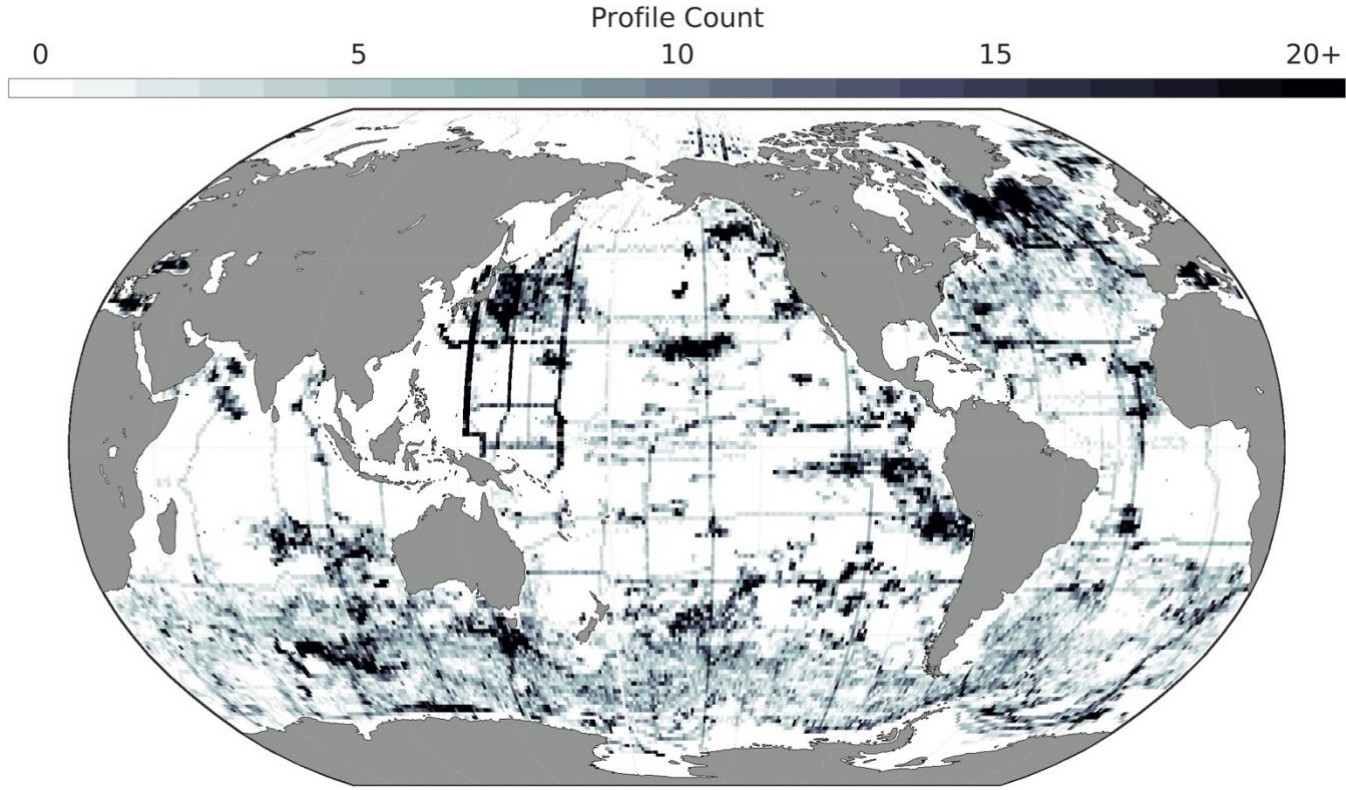

**Figure A1.** The number of profiles (either ship-based or Argo float-based) from the combined dataset used to train machine learning algorithms to produce GOBAI-O$_2$ that are contained within each $1° \times 1°$ box in the global ocean.

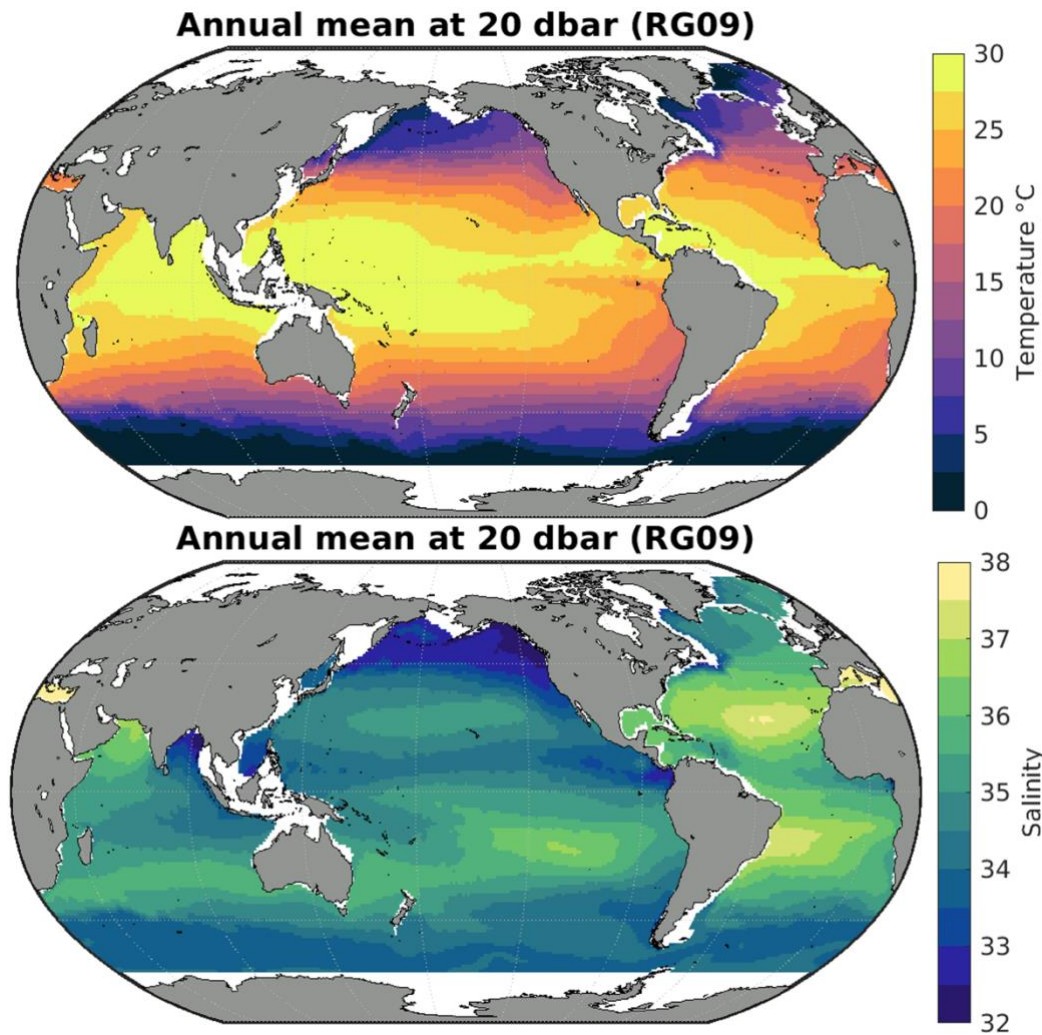

**Figure A2.** Annual mean in situ temperature (top) and salinity (bottom) from RG09 (2004–2022) at 20 dbars.

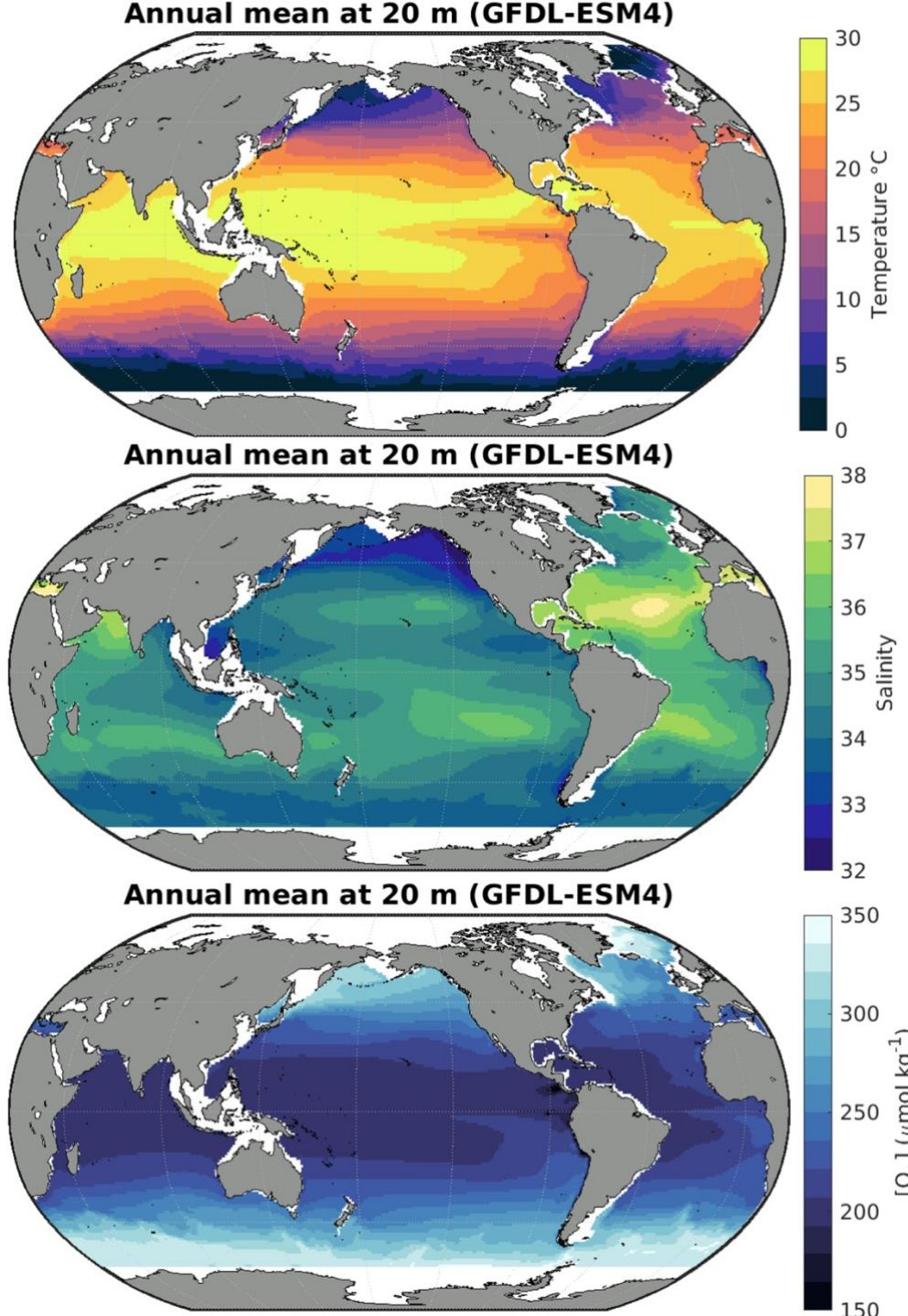

**Figure A3.** Annual mean in situ temperature (top), salinity (middle), and dissolved oxygen concentration (bottom) from GFDL-ESM4 (2004–2021) at 20 meters.

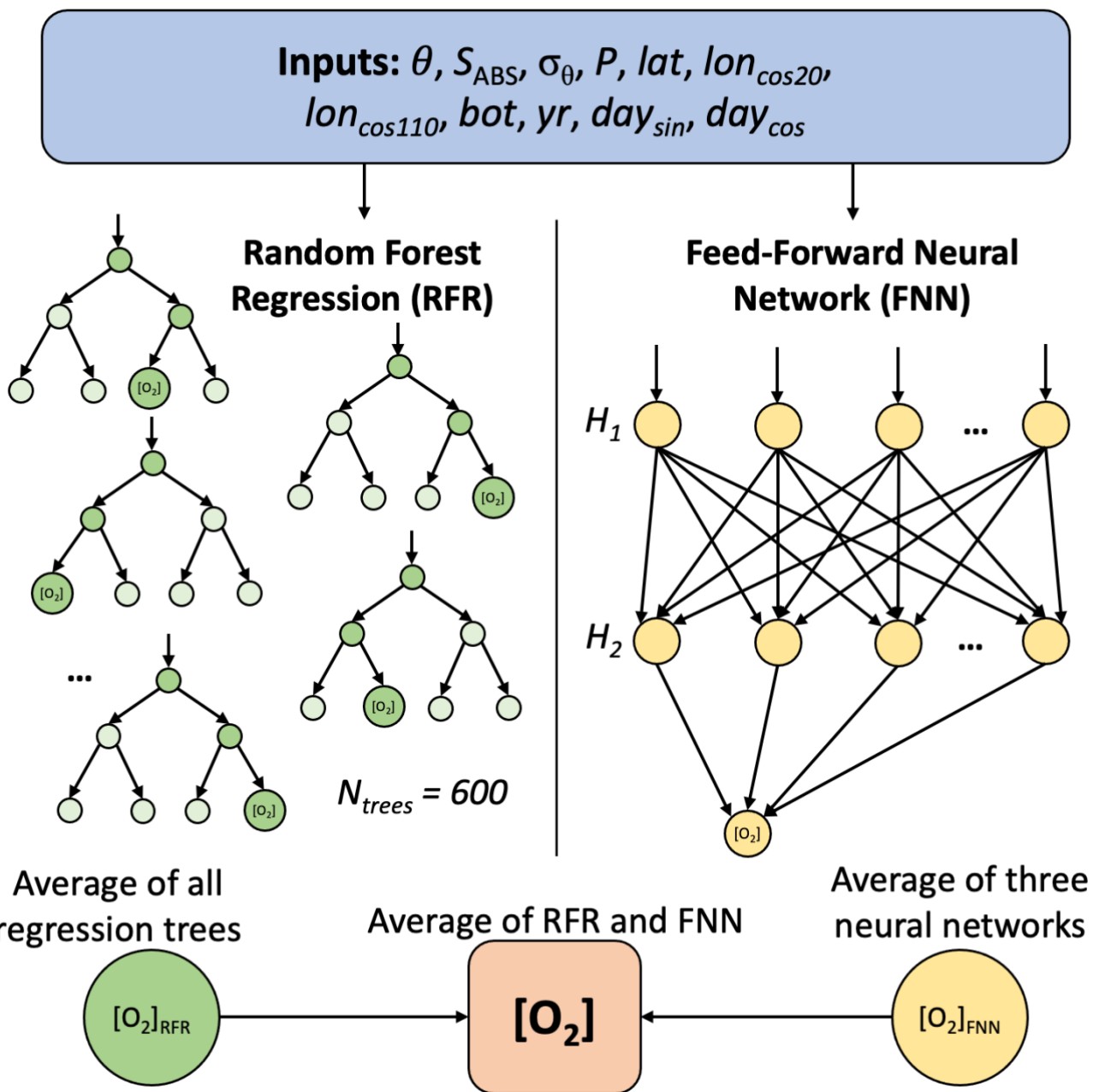

**Figure A4.** A schematic for the random forest regressions (RFRs) and feed-forward neural networks (FNNs). A random subset of the predictors is used for each tree in the RFR, and a randomly chosen predictor is used for each node split. The two hidden layers ($H_1$ and $H_2$) in each of the three FNNs have 10 and 20, 15 and 15, and 20 and 10 nodes. Each machine learning algorithm is trained with input data and [$O_2$] observations, then used to predict [$O_2$] from new input data.

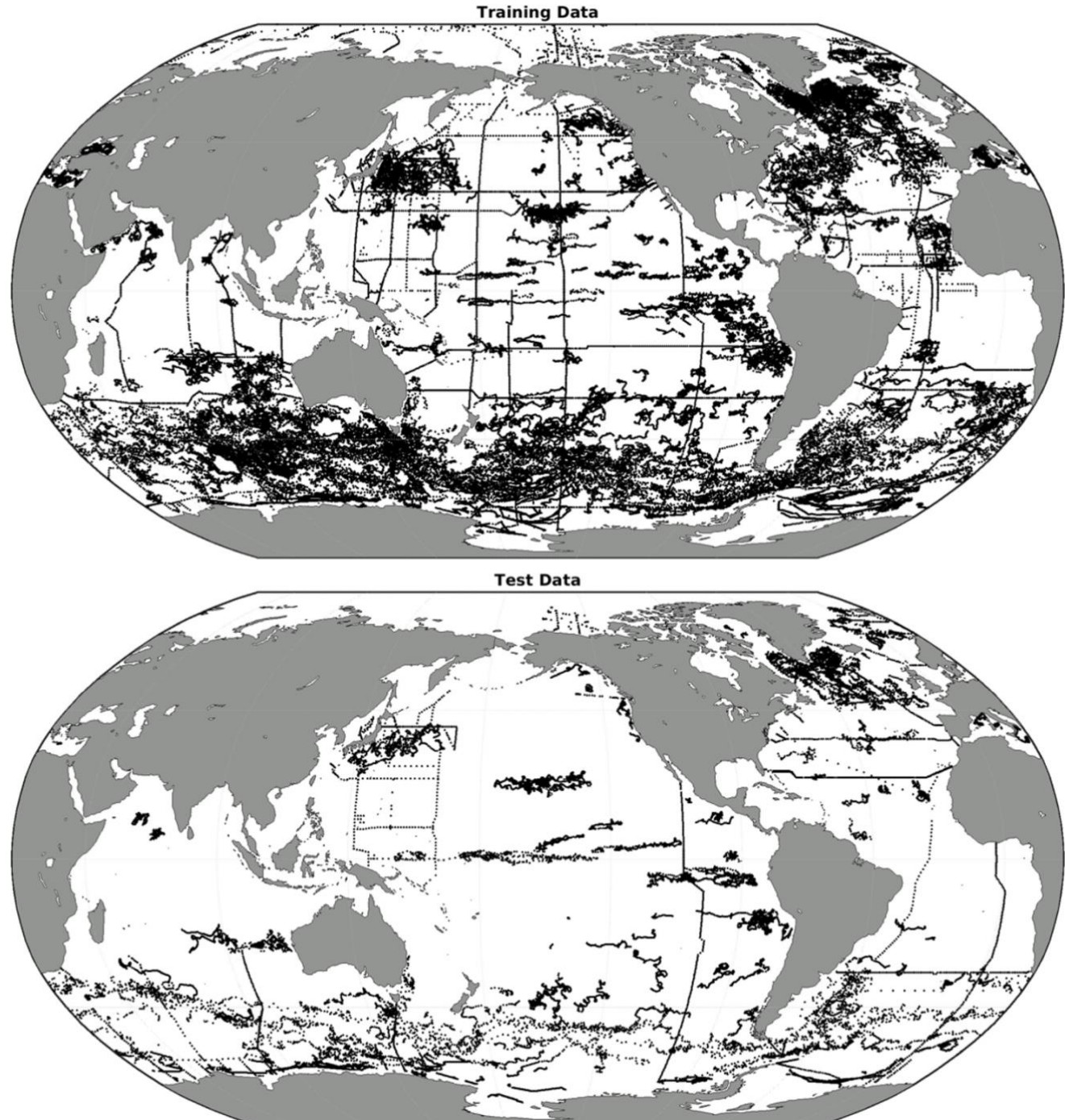

**Figure A5.** The spatial distribution of profile data used to (a) train and (b) test RFR$_{Data-Eval}$ and FNN$_{Data-Eval}$ algorithms.

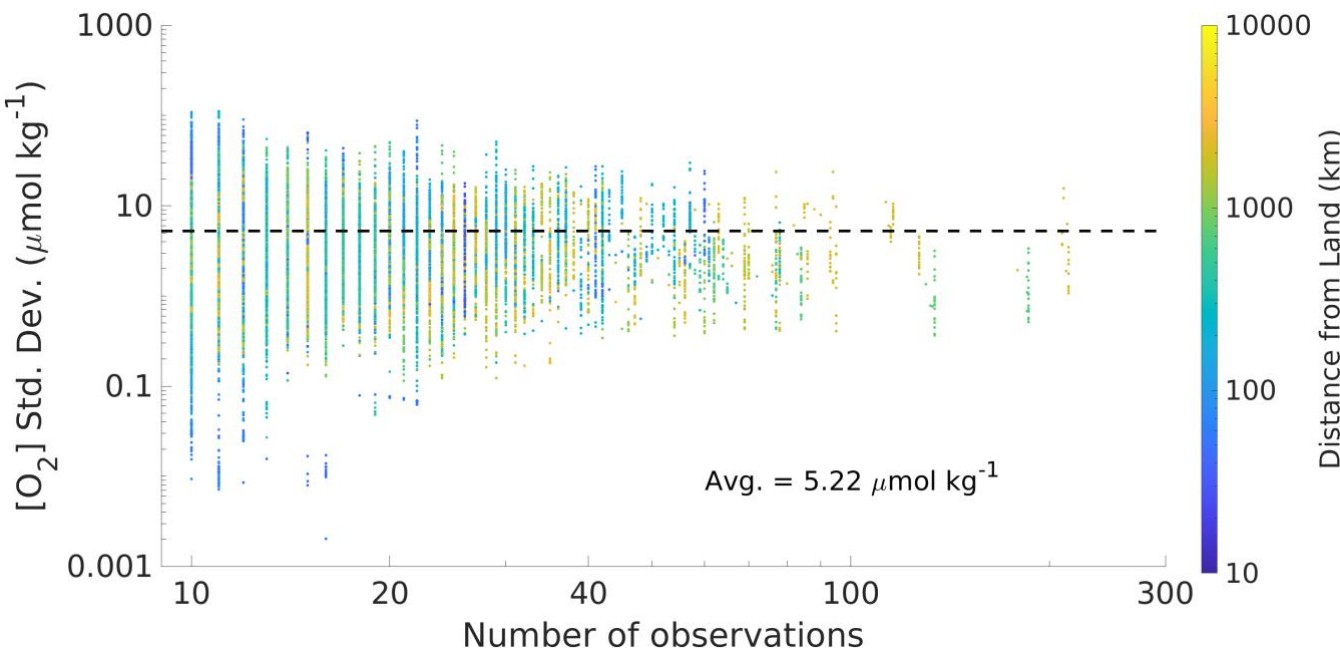

**Figure A6.** A comparison between the number of observations binned within a four-dimensional grid cell and the standard deviation in [O$_2$] among those observations. The horizontal black line shows the mean standard deviation (5.21 μmol kg$^{-1}$).

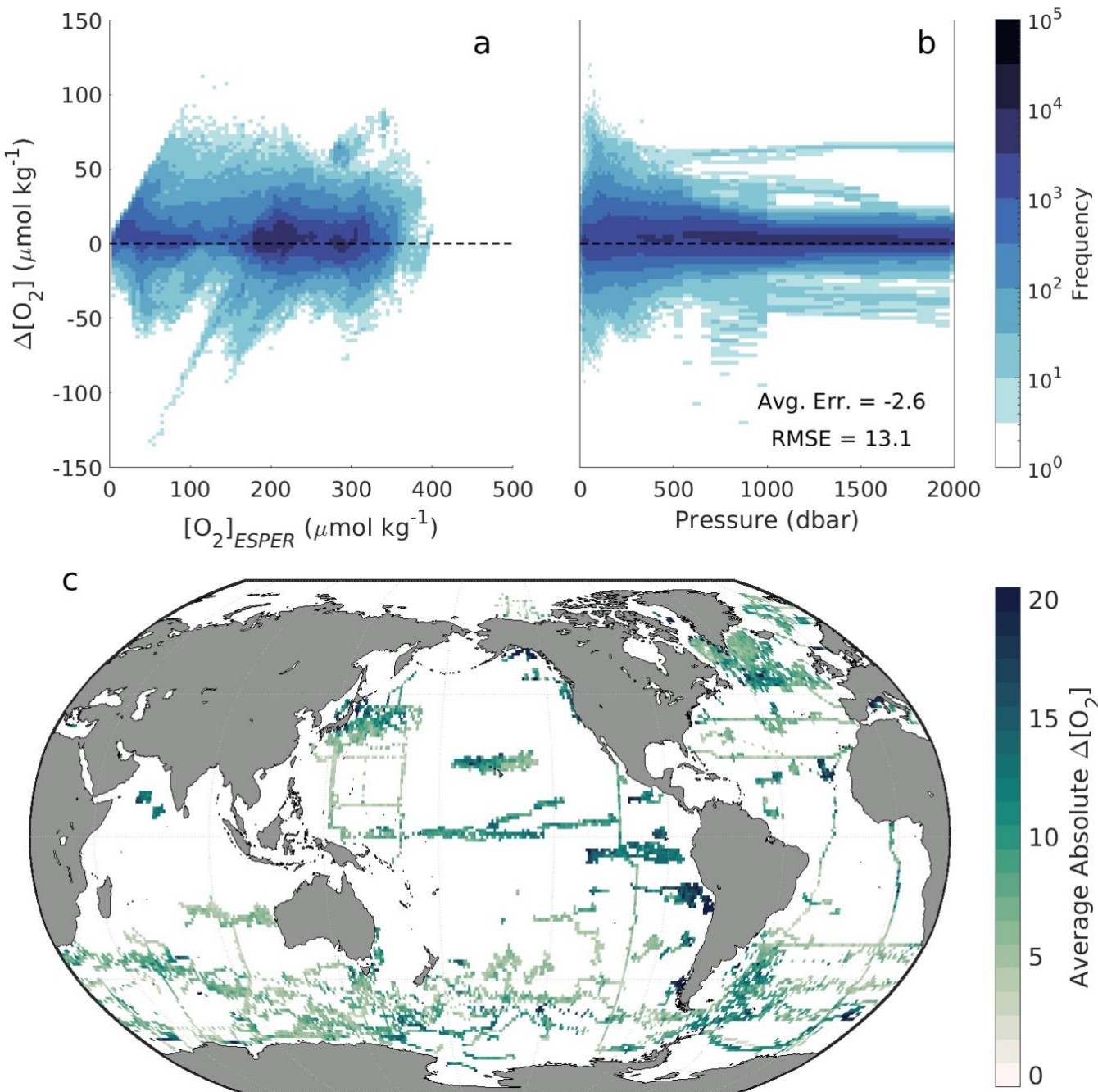

**Figure A7.** For withheld Argo and GLODAP data, two-dimensional histograms showing offsets between measured and ESPER-Mixed-calculated oxygen ($\Delta[O_2] = [O_2]_{meas} - [O_2]_{ESPER}$) as a function of (a) $[O_2]_{ESPER}$ and (b) pressure in the water column. Offsets are binned into cells that are 2.5 µmol kg$^{-1}$ tall in terms of $\Delta[O_2]$ and 5 µmol kg$^{-1}$ wide in terms of (a) $[O_2]_{ESPER}$ or (b) equivalent in width to the interpolated pressure levels of the data. (c) Absolute $\Delta[O_2]$ values averaged over depth and time for 1° latitude by 1° longitude grid cells in the global ocean.

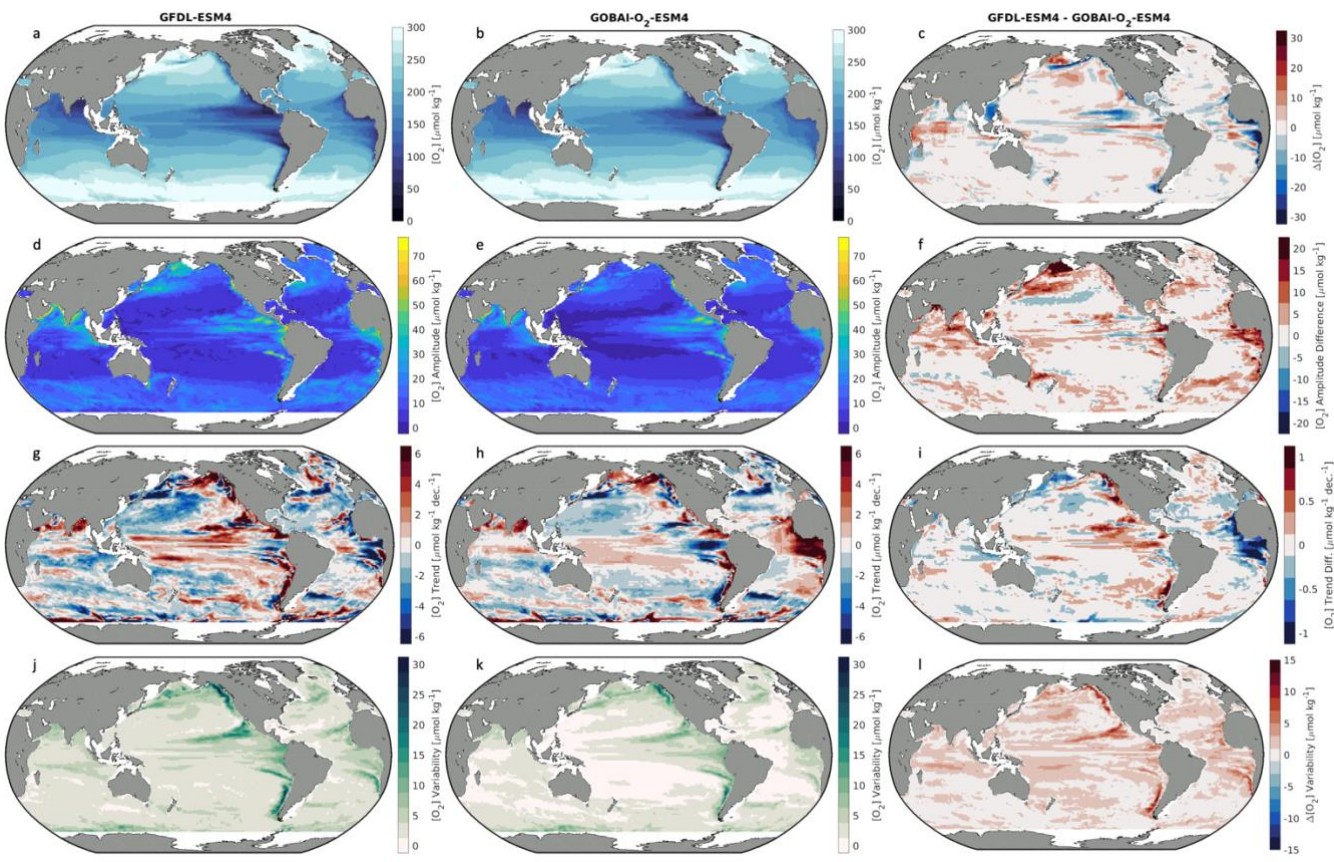

**Figure A8.** Integrated from 0 to 200 dbars: (a,b) long-term mean [O₂], (d,e) seasonal [O₂] amplitudes, (g,h) trends in [O₂], and (j,k) interannual variability in [O₂] for (a,d,g,j) GFDL-ESM4 and (b,e,h,k) GOBAI-O₂-ESM4, along with (c,f,i,l) the difference between the two.

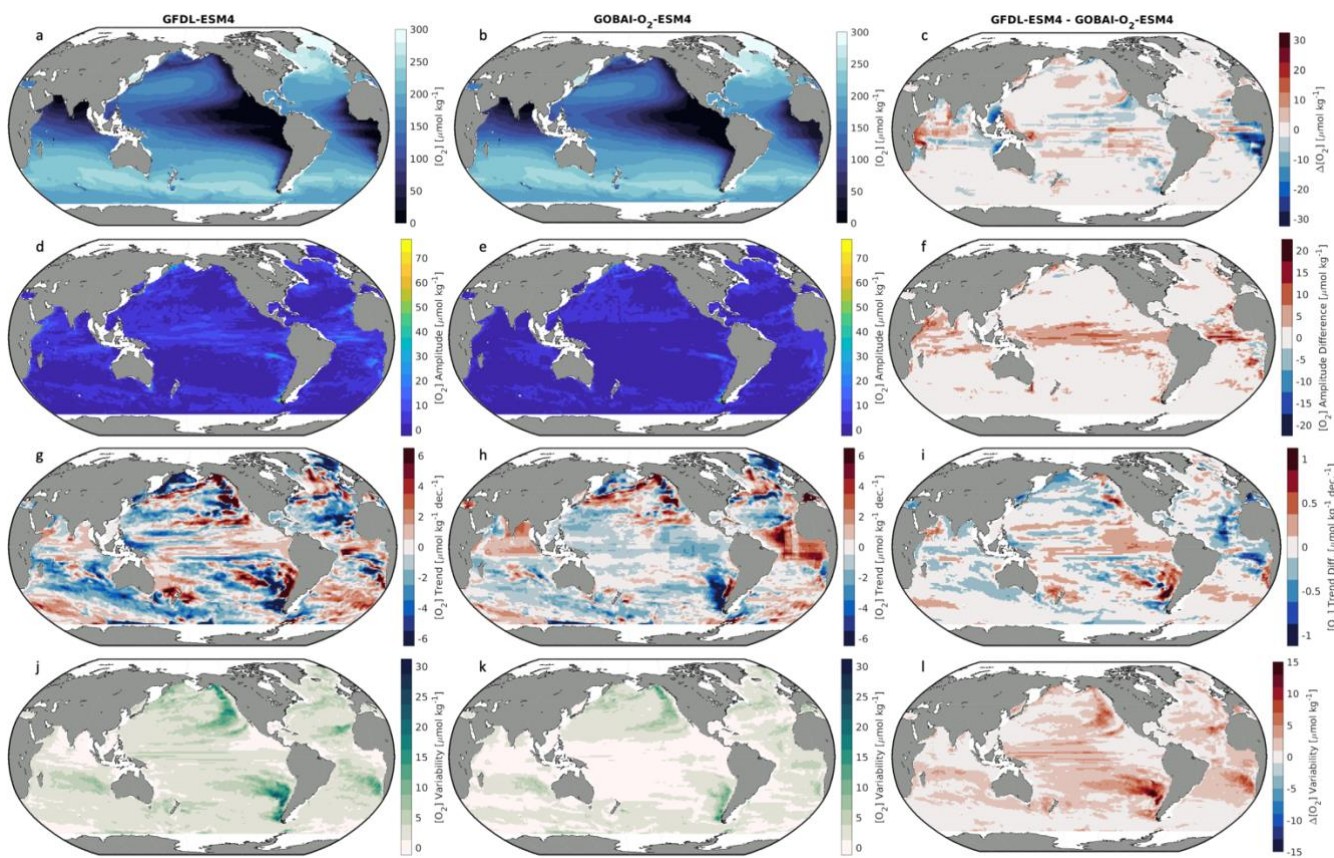

**Figure A9.** Integrated from 200 to 1000 dbars: (a,b) long-term mean [O₂], (d,e) seasonal [O₂] amplitudes, (g,h) trends in [O₂], and (j,k) interannual variability in [O₂] for (a,d,g,j) GFDL-ESM4 and (b,e,h,k) GOBAI-O₂-ESM4, along with (c,f,i,l) the difference between the two.

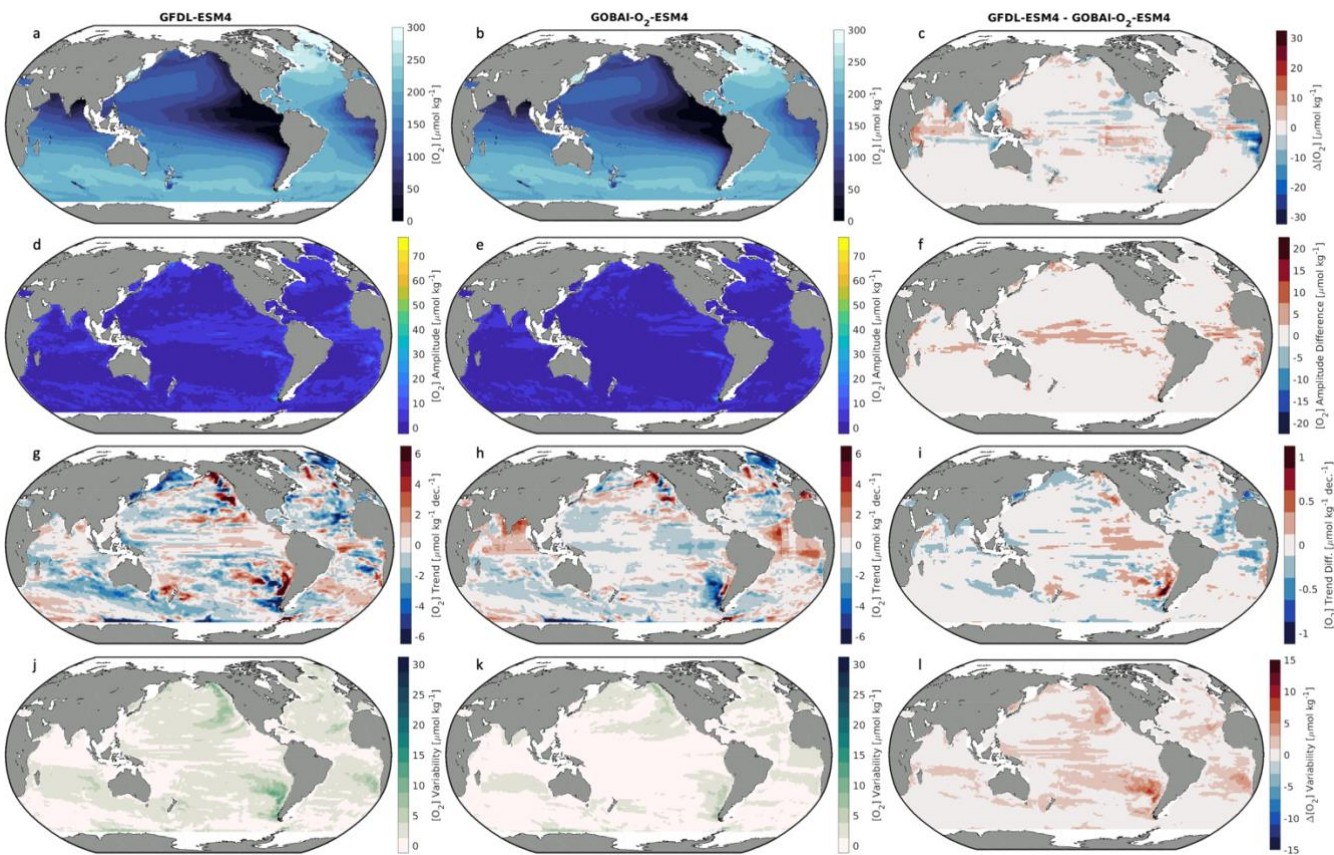

**Figure A10.** Integrated from 0 to 2000 dbars: (a,b) long-term mean [O₂], (d,e) seasonal [O₂] amplitudes, (g,h) trends in [O₂], and (j,k) interannual variability in [O₂] for (a,d,g,j) GFDL-ESM4 and (b,e,h,k) GOBAI-O₂-ESM4, along with (c,f,i,l) the difference between the two.

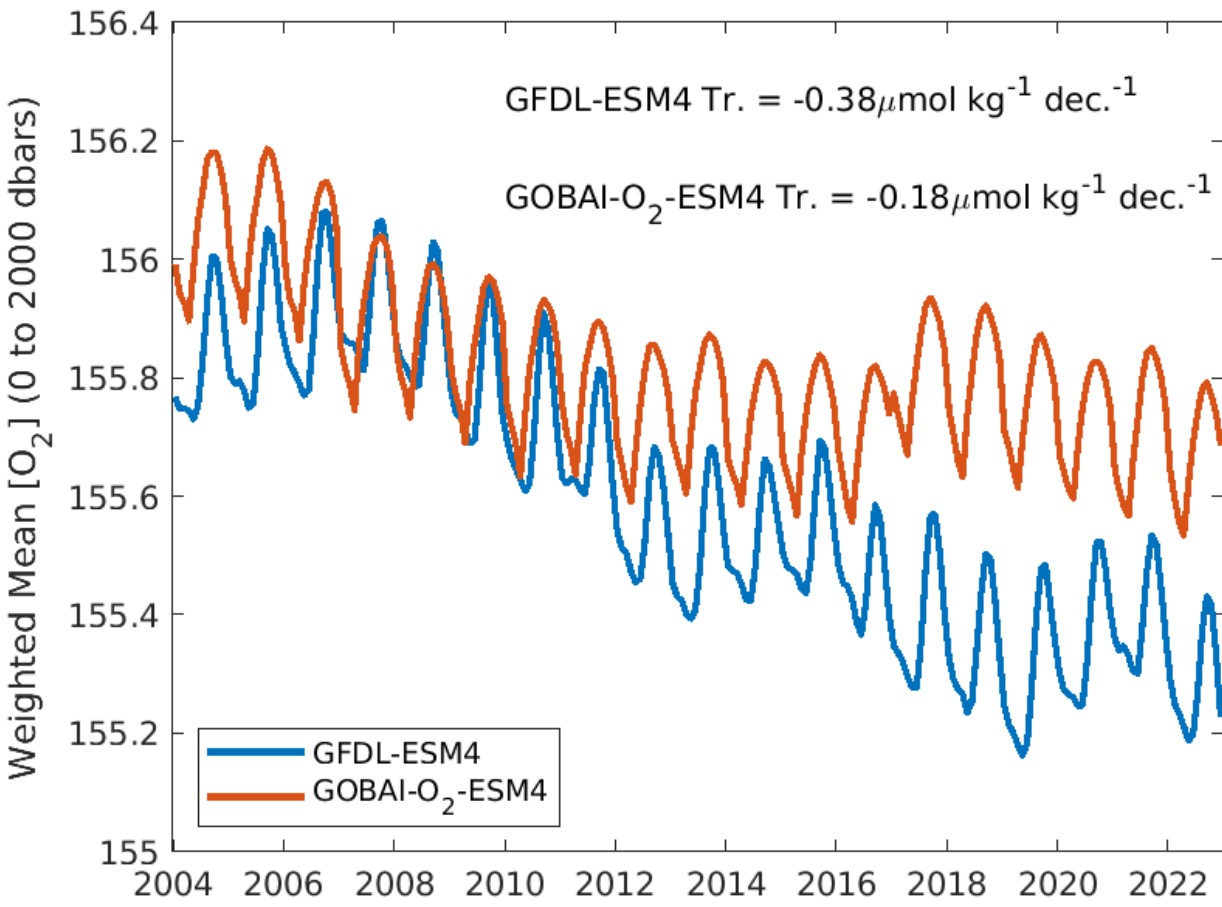

**Figure A11.** Monthly area-weighted mean [$O_2$] integrated globally from 0 to 2000 dbars from GFDL-ESM4 (blue) and GOBAI-$O_2$-ESM4 (orange).

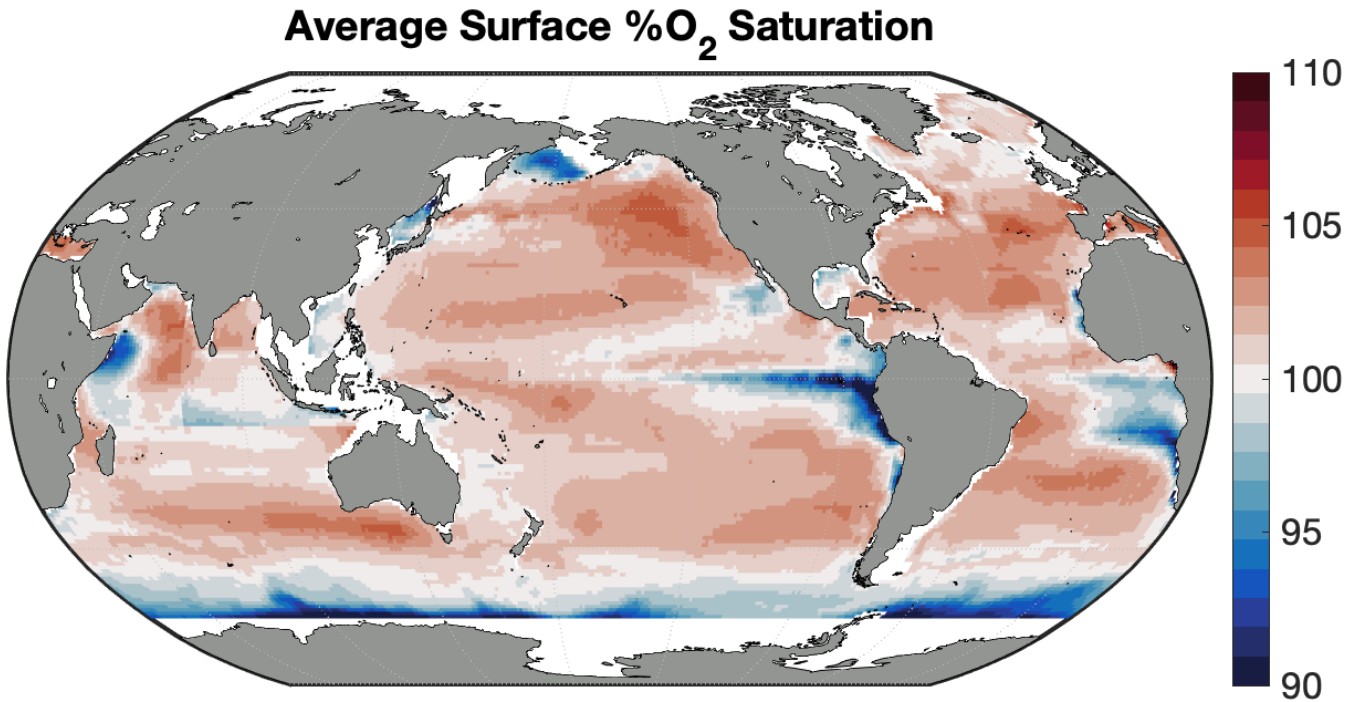

**Figure A12.** Long-term mean percent oxygen saturation on the uppermost pressure level in GOBAI-$O_2$.

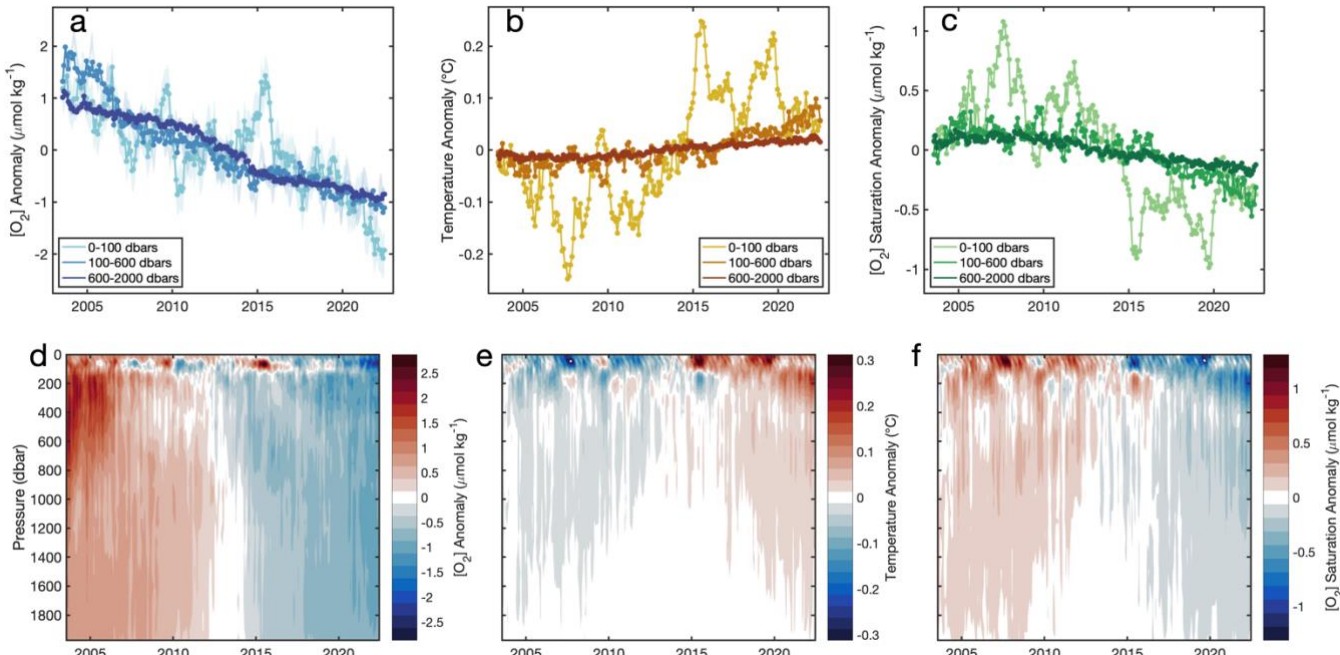

**Figure A13.** Monthly mean de-seasonalized (a) $[O_2]$ anomalies from GOBAI-$O_2$, (b) temperature anomalies from RG09, and (c) $[O_2]_{sat.}$ anomalies calculated from RG09 temperature and salinity fields, each integrated globally over three pressure layers: 0–100, 100–600, and 600–2000 dbars. (a) Shading represents uncertainty determined as the average difference between mean $[O_2]$ from GOBAI-$O_2$-ESM4 versus GFDL-ESM4 in each layer. Hovmöller diagrams showing monthly mean de-seasonalized (d) $[O_2]$ anomalies from GOBAI-$O_2$, (e) temperature anomalies from RG09, and (f) $[O_2]_{sat.}$ anomalies calculated from RG09 temperature and salinity fields, each over depth in decibars from 2004 to 2022. Anomalies in each parameter are calculated as monthly mean values with a seasonal cycle removed and minus the long-term mean either (a–c) integrated over a pressure layer or (d–f) on a given pressure level.

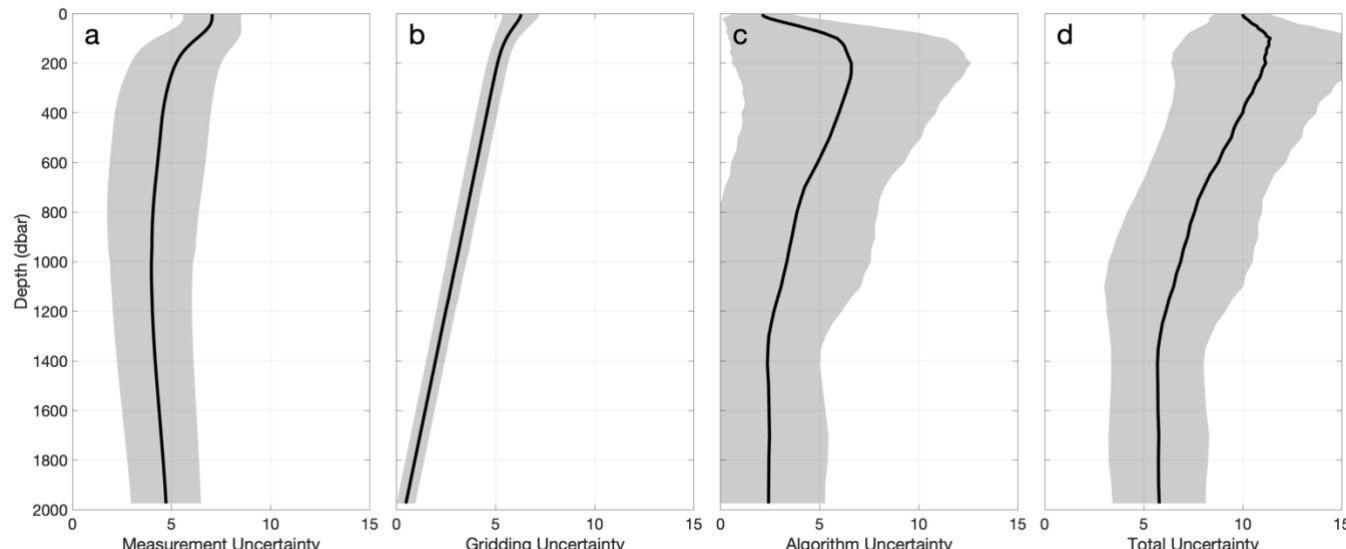

**Figure A14.** Global mean depth profiles of uncertainty contributors to GOBAI-O$_2$, including (a) measurement uncertainty, (b) gridding uncertainty, (c) algorithm uncertainty, and (d) total uncertainty. The shaded region represents variability in space, and is calculated as the standard deviation on each depth level of the mean uncertainties over time for each grid cell.

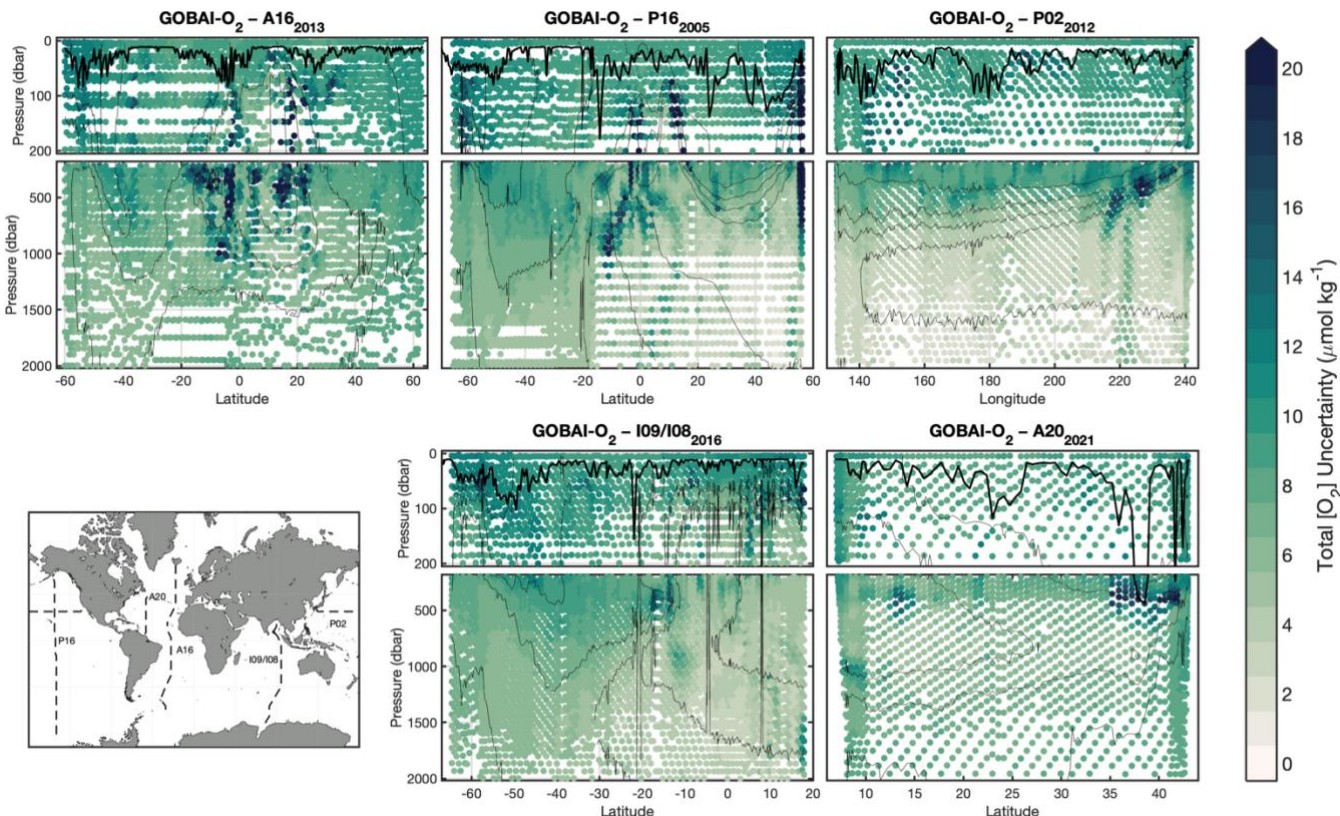

**Figure A15.** Section plots displaying total uncertainty estimates from GOBAI-O$_2$ that correspond to discrete measurements of [O$_2$] from repeat hydrography cruises, to be compared to Δ[O$_2$] values in Fig. 11.

## Appendix B. Supplemental Tables

**Table B1.** Boundaries for the seven large ocean regions used to fit machine learning algorithms.

| Basin | Polygon Vertices: [Longitude,Latitude; …] |
|---|---|
| Atl. | [−60,0; −79,9.4; −81,8.4,−100,22;−100,45;−6,45;−6,35;4,15;25,0;22,−35;−68,−35;−60,0] |
| Pac.* | [104,0;104,70;181,70;181,0;181,−35;145,−35;131,−30;131,0;104,0] |
| | [−180,0;−180,70;−150,70;−150,67;−120,67;−100,22; −81,8.4; −79,9.4;−60,0;−68,−35;−180,−35;−180,0] |
| Ind. | [22,−35;25,10;38,35;104,35;104,0;131,0;131,−30;116,−35;22,−35] |
| Arc. | [−180,64;−180,90;181,90;181,67;90,67;0,50;0,40;−6,40;−6,35;−90,35;−120,64;−180,64] |
| Med. | [−6.5,40;0,40;0,45;20,47;38,35;34,30;−5,30;−6.5,40] |
| N. Sou. | [−180,−60;−180,−25;181,−25;181,−60;−180,−60] |
| S. Sou. | [−180,−90;−180,−50;181,−50;181,−90;−180,−90] |

*Two sets of boundaries are given for the Pacific to accommodate crossing the international date line.

**Table B2.** Error statistics (mean $\Delta[O_2]$ and RMSD) for tests using RFR and FNN algorithms trained on a subset of Argo and GLODAP data and tested with a separate subset of withheld data. Also shown are error statistics corresponding to the ensemble average (ENS) of the estimates from both algorithms.

| Basin | | | Evaluation Exercise with Observational Data | | | | | |
|---|---|---|---|---|---|---|---|---|
| | Training Data Points | Assessment Data Points | RFR$_\text{Data-Eval}$ | | FNN$_\text{Data-Eval}$ | | ENS$_\text{Data-Eval}$ | |
| | | | Mean $\Delta[O_2]$ ($\mu$mol kg$^{-1}$) | RMSD ($\mu$mol kg$^{-1}$) | Mean $\Delta[O_2]$ ($\mu$mol kg$^{-1}$) | RMSD ($\mu$mol kg$^{-1}$) | Mean $\Delta[O_2]$ ($\mu$mol kg$^{-1}$) | RMSD ($\mu$mol kg$^{-1}$) |
| Atl. | 592,099 | 109,134 | -2.0 | 10.9 | -1.5 | 8.9 | -1.8 | 9.2 |
| Pac. | 1,816,367 | 466,788 | 0.5 | 10.0 | 0.8 | 10.9 | 0.6 | 9.9 |
| Ind. | 335,768 | 82,491 | 0.3 | 7.1 | 1.5 | 7.9 | 0.9 | 7.1 |
| Arc. | 800,328 | 263,873 | -1.1 | 9.1 | -1.7 | 9.6 | -1.4 | 9.1 |
| Med. | 214,540 | 33,899 | 3.8 | 10.8 | 2.6 | 12.0 | 3.2 | 11.0 |
| N. Sou. | 2,236,153 | 480,846 | 0.1 | 7.3 | 0.4 | 7.6 | 0.2 | 7.2 |
| S. Sou. | 1,430,492 | 364,133 | -0.6 | 8.2 | -0.8 | 8.6 | -0.7 | 8.2 |
| All | 7,425,747 | 1,801,164 | -0.2 | 9.0 | -0.2 | 9.5 | -0.2 | 8.8 |

**Table B3.** Error statistics (mean Δ[O$_2$] and RMSD) for tests using RFR and FNN algorithms trained on a subset of output from GFDL-ESM4 (corresponding to locations of available Argo and GLODAP data) and tested using a separate subset of withheld output from GDFL-ESM4. Also shown are error statistics corresponding to the ensemble average (ENS) of the estimates from both algorithms.

| Basin | Training Data Points | Assessment Data Points | RFR$_{ESM4-Eval}$ | | FNN$_{ESM4-Eval}$ | | ENS$_{ESM4-Eval}$ | |
|---|---|---|---|---|---|---|---|---|
| | | | Mean Δ[O$_2$] (µmol kg$^{-1}$) | RMSD (µmol kg$^{-1}$) | Mean Δ[O$_2$] (µmol kg$^{-1}$) | RMSD (µmol kg$^{-1}$) | Mean Δ[O$_2$] (µmol kg$^{-1}$) | RMSD (µmol kg$^{-1}$) |
| Atl. | 184,418 | 28,235,064 | -2.1 | 12.8 | -0.6 | 9.5 | -1.3 | 9.6 |
| Pac. | 533,208 | 69,369,456 | 0.1 | 7.5 | -0.1 | 8.5 | 0.0 | 7.4 |
| Ind. | 86,060 | 20,736,144 | 0.7 | 8.8 | -0.3 | 7.3 | 0.2 | 7.2 |
| Arc. | 293,540 | 11,547,744 | 0.1 | 4.1 | -0.1 | 4.6 | 0.0 | 4.1 |
| Med. | 32,110 | 1,096,680 | 0.8 | 4.8 | 1.3 | 7.5 | 1.0 | 5.5 |
| N. Sou. | 756,444 | 67,626,624 | -0.1 | 4.4 | -0.1 | 4.9 | -0.1 | 4.4 |
| S. Sou. | 519,610 | 31,412,472 | 0.1 | 3.4 | -0.1 | 3.6 | 0.0 | 3.3 |
| All | 2,405,390 | 230,024,184 | -0.2 | 7.7 | -0.2 | 7.3 | -0.2 | 6.7 |

**Table B4.** Error statistics (mean Δ[O$_2$] and RMSD) for tests using RFR and FNN algorithms trained on a subset of Argo and GLODAP data and tested with all available GLODAP data. Also shown are error statistics corresponding to the ensemble average (ENS) of the estimates from both algorithms and corresponding to the ESPER-Mixed model (Carter et al., 2021).

| Basin | Training Data Points | Assessment Data Points | RFR$_{Data-Eval}$ | | FNN$_{Data-Eval}$ | | ENS$_{Data-Eval}$ | | ESPER-Mixed | |
|---|---|---|---|---|---|---|---|---|---|---|
| | | | Mean Δ[O$_2$] (µmol kg$^{-1}$) | RMSD (µmol kg$^{-1}$) | Mean Δ[O$_2$] (µmol kg$^{-1}$) | RMSD (µmol kg$^{-1}$) | Mean Δ[O$_2$] (µmol kg$^{-1}$) | RMSD (µmol kg$^{-1}$) | Mean Δ[O$_2$] (µmol kg$^{-1}$) | RMSD (µmol kg$^{-1}$) |
| Atl. | 592,099 | 180,374 | 0.0 | 7.9 | 0.5 | 9.4 | 0.2 | 7.9 | 0.1 | 11.3 |
| Pac. | 1,816,367 | 495,035 | 0.6 | 6.1 | 1.2 | 9.3 | 0.9 | 7.2 | -0.3 | 11.0 |
| Ind. | 335,768 | 42,460 | 0.6 | 4.2 | 1.1 | 7.2 | 0.9 | 5.4 | -1.3 | 9.2 |
| Arc. | 800,328 | 227,905 | 0.0 | 5.5 | -0.1 | 8.7 | 0.0 | 6.7 | 1.2 | 11.0 |
| Med. | 214,540 | 60 | -3.6 | 8.0 | 0.0 | 4.3 | -1.8 | 5.3 | -5.5 | 7.7 |
| N. Sou. | 2,236,153 | 174,368 | 0.7 | 4.9 | 0.9 | 7.2 | 0.8 | 5.7 | -0.7 | 8.4 |
| S. Sou. | 1,430,492 | 141,065 | 0.4 | 5.3 | -0.2 | 7.9 | 0.1 | 6.2 | -0.3 | 9.5 |
| All | 7,425,747 | 1,261,267 | 0.4 | 6.1 | 0.7 | 8.9 | 0.6 | 6.9 | 0.0 | 10.7 |

**Table B5.** Estimated decadal trends and uncertainties in [$O_2$] ($\mu$mol kg$^{-1}$ decade$^{-1}$) and oxygen inventory (% decade$^{-1}$) in different pressure layers of GOBAI-$O_2$. Uncertainties are determined according to the procedure in Appendix E, both using the autocorrelation of residuals to the linear least squares model (Autocov.) and by incorporating estimated uncertainty in global mean GOBAI-$O_2$ fields ($u$(ESM4)). The value used to represent uncertainty on each trend (larger value) is in bold.

| Pressure Layer | [$O_2$] Trend ($\mu$mol kg$^{-1}$ dec.$^{-1}$) | Trend Uncertainty ($\mu$mol kg$^{-1}$ dec.$^{-1}$) | | $O_2$ Inventory Trend (% dec.$^{-1}$) | Trend Uncertainty (% dec.$^{-1}$) | |
| --- | --- | --- | --- | --- | --- | --- |
| | | Autocov. | $u$(ESM4) | | Autocov. | $u$(ESM4) |
| 0 – 100 dbar | **−1.10** | **0.60** | 0.53 | −0.49 | **0.26** | 0.23 |
| 100 – 600 dbar | **−1.38** | **0.42** | 0.36 | −0.86 | **0.26** | 0.22 |
| 600 – 2000 dbar | **−1.12** | 0.45 | **0.54** | −0.79 | 0.32 | **0.38** |
| 0 – 1000 dbar | **−1.28** | 0.14 | **0.18** | −0.82 | 0.09 | **0.11** |
| 0 – 2000 dbar | **−1.19** | 0.03 | **0.05** | −0.79 | 0.02 | **0.04** |

**Table B6.** Summary error statistics between direct observations from repeat hydrography cruises and GOBAI-$O_2$ and WOA18.

| Cruise | GOBAI-$O_2$ | | WOA18 | |
| --- | --- | --- | --- | --- |
| | Mean $\Delta$[$O_2$] | RMSD | Mean $\Delta$[$O_2$] | RMSD |
| A16 (2013) | −0.2 | 9.2 | 0.2 | 12.0 |
| P16 (2005) | 0.1 | 14.9 | 0.2 | 14.5 |
| P02 (2012) | −1.3 | 9.9 | −0.4 | 12.9 |
| I08/I09 (2016) | −2.3 | 10.9 | −1.1 | 13.0 |
| A20 (2021) | −7.8 | 23.4 | −2.2 | 21.4 |

**Appendix C. Supplemental Datasets**

1. The original and vertically interpolated observational datasets from the BGC Argo and GLODAP databases that are used to develop GOBAI-O$_2$ can be found at https://doi.org/10.5281/zenodo.7747237 (Sharp, 2023a).

2. The algorithms trained on vertically interpolated observational data that were applied to predictor variables to produce GOBAI-O$_2$ can be found at https://doi.org/10.5281/zenodo.7747308 (Sharp, 2023b).

**Appendix D. Float Data Adjustments**

A negative median bias ($-1.18$ µmol kg$^{-1}$) in float [O$_2$] measurements compared to co-located ship [O$_2$] measurements (below 300 dbars, to avoid the impact of high frequency variability near the surface) was adjusted by fitting the differences ($\Delta$[O$_2$]) to a linear least squares model as a function of float [O$_2$], and adding that [O$_2$]-dependent adjustment back on to the float [O$_2$] measurements. The $\Delta$[O$_2$] values as a function of float [O$_2$] before (a) and after (b) this adjustment are shown in the Figure D1. This resulted in a reduced median $\Delta$[O$_2$] of 0.33 µmol kg$^{-1}$.

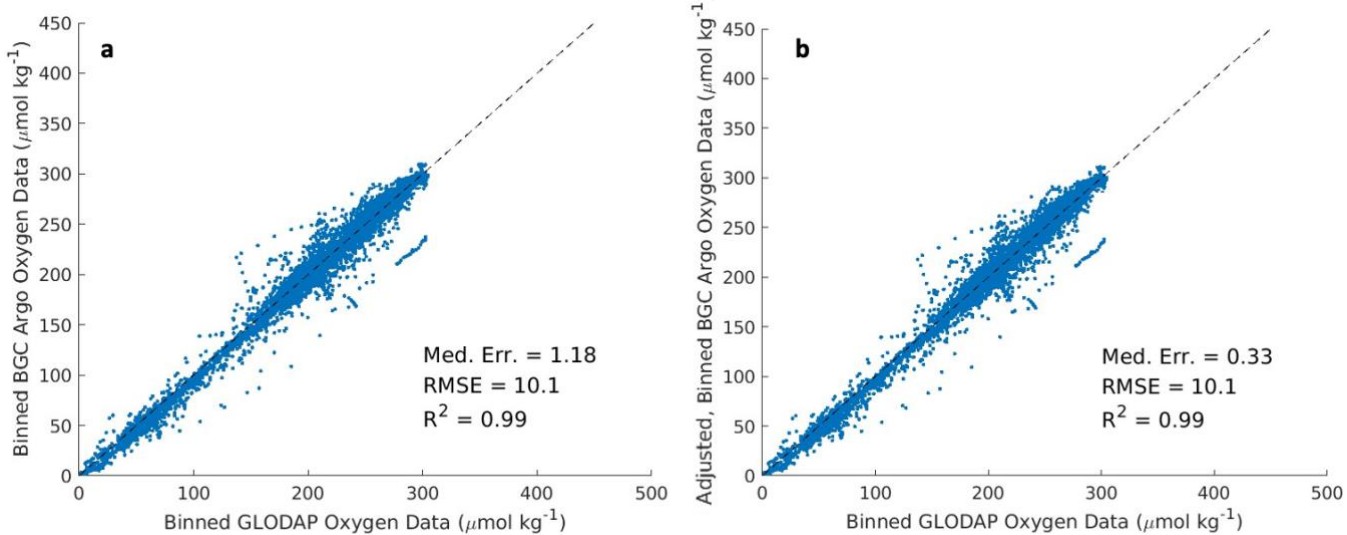

**Figure D1.** Unadjusted (a) and adjusted (b) matchups between BGC Argo [O$_2$] measurements (y-axis) versus GLODAP [O$_2$]
measurements (x-axis). The adjustment procedure doesn't mitigate the scatter between the matchups, but does reduce the median error.

**Appendix E. Determination of Trends**

Trends and associated uncertainties in GOBAI-$O_2$ were determined via the following procedure:

1.  Spatially weighted monthly mean [$O_2$] values for the entire GOBAI-$O_2$ domain or within specified pressure layers were calculated from gridded [$O_2$], using relative grid cell volumes as weights.

2.  A linear least squares model with a trend, intercept, and annual (12-month) and semi-annual (6-month) harmonics was fit to monthly mean [$O_2$] values. The monthly trend from the least squares model was multiplied by 120 to obtain a **decadal trend of weighted mean [$O_2$]**.

3.  Uncertainty on the decadal trends were assessed in two different ways, and the largest of the two uncertainty estimates taken for each analyzed pressure layer, indicating that either (a) uncertainty in the linear least squares model or (b) uncertainty in the GOBAI-$O_2$ fields was driving uncertainty in the trend. In all cases, the uncertainty estimate from the second method was chosen, indicating decadal trend uncertainty was controlled primarily by uncertainty in the GOBAI-$O_2$ fields. The two methods were as follows:

    a.  Using the autocovariance of residuals from the linear least squares model:

        i.   The standard error on the trend was calculated from the covariance matrix of the linear least squares model.

        ii.  The autocovariance of the residuals from the least squares model was examined to compute the e-folding timescale, and the effective degrees of freedom were obtained by dividing the number of monthly mean [$O_2$] values by the e-folding timescale and subtracting the number of least squares model parameters.

        iii. The standard error on the trend was scaled by the effective degrees of freedom, multiplied by 2 to obtain a 95% confidence interval, and multiplied by 120 to obtain an **uncertainty on the decadal trend of weighted mean [$O_2$]**.

    b.  By incorporating estimated uncertainty in global mean GOBAI-$O_2$ fields:

        i.   Uncertainties in monthly mean [$O_2$] values were determined as the standard deviations of monthly differences between GOBAI-$O_2$-ESM4 and GFDL-ESM4 (section 3.1.2).

        ii.  These uncertainties were used to compute a weight matrix for the linear least squares fit, and the effective degrees of freedom were obtained as previously described.

        iii. The standard error on the trend was scaled by the effective degrees of freedom, multiplied by 2 to obtain a 95% confidence interval, and multiplied by 120 to obtain an **uncertainty on the decadal trend of weighted mean [$O_2$]**.

4.  The process was repeated for oxygen inventories for the entire GOBAI-$O_2$ domain or within each specified pressure layer; inventories were determined from gridded [$O_2$], volumes of each grid cell, and densities of each grid cell.

## 6 Data availability

GOBAI-O$_2$ is available as a NetCDF file at https://doi.org/10.25921/z72m-yz67 (Sharp et al., 2022; last access: 07 Jul. 2023); additional information and animations can be found at https://www.pmel.noaa.gov/gobai/. GLODAPv2 is updated annually and is available at www.glodap.info. GFDL-ESM4 model output can be accessed via the World Climate Research Programme database (https://esgf-node.llnl.gov/projects/cmip6/). Data from the 2018 World Ocean Atlas can be accessed through NOAA NCEI (https://www.ncei.noaa.gov/products/world-ocean-atlas). The OneArgo-Mat toolbox used to download Argo float data is available at https://doi.org/10.5281/zenodo.6588041; the toolbox acquires data from two global data assembly centers: Coriolis (ftp://ftp.ifremer.fr/ifremer/argo) and US-GODAE (ftp://usgodae.org/pub/outgoing/argo). The Roemmich and Gilson (2009) Argo-based temperature and salinity product is available at https://sio-argo.ucsd.edu/RG_Climatology.html.

## 7 Author contributions

JDS, AJF, BRC, and GCJ conceptualized and planned the project. JDS produced the data product, conducted all analysis, created the data visualizations, and wrote the original draft of the manuscript. JDS, AJF, BRC, GCJ, JPD, and CS reviewed and edited the manuscript.

## 8 Competing interests

The authors declare no competing interests.

## 9 Acknowledgements

The authors thank the International Argo Program and the national programs that contribute to it (https://argo.ucsd.edu, https://www.ocean-ops.org); the Argo Program is part of the Global Ocean Observing System. The authors also thank the many scientists who have contributed to the GLODAPv2 database (www.glodap.info) by securing funding, dedicating their time to collect and share data, and assembling and quality-controlling those data. JDS and BRC were funded by the National Oceanic and Atmospheric Administration (NOAA) Climate Program Office's Climate Observations and Monitoring (COM) and Climate Variability and Predictability (CVP) programs, in partnership with NOAA's Global Ocean Monitoring and Observing (GOMO) program, through Award NA21OAR4310251. CS and GCJ were funded, and BRC and JDS were partially funded, by the Ocean Observing and Monitoring Division (now GOMO) for the establishment and use of a biogeochemical Argo array within the California Current System. JDS and BRC were funded through the Cooperative Institute for Climate, Ocean, & Ecosystem Studies (CIOCES) under NOAA Cooperative Agreement NA20OAR4320271. JDS, AJF, and GCJ were supported by NOAA's Pacific Marine Environmental Laboratory (PMEL). We are grateful to Hartmut Frenzel (CICOES) for

computational assistance, and to Hernan Garcia and two anonymous reviewers for providing comments that prompted significant improvements to this manuscript. This is CICOES contribution no. 2022-1226 and PMEL contribution no. 5416.

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
