# Peer review of "GOBAI-O2: temporally and spatially resolved fields of ocean interior dissolved oxygen over nearly two decades"

_Earth System Science Data, 2022_

## Referee Comment (RC1)

This is an interesting paper using a novel approach to quantify global ocean O2 content seasonal to decadal-scale (S2D) variability and trends. The authors use AI and ML in an effort to resolve global and regional ocean S2D O2 variability and trends.The authors combine/aggregate contemporary ship-based Winkler-based O2 data used in GLODAP and sensor-based O2 data from BCG-ARGO.

The authors indicate that the spatial and temporal heterogeneity coverage of the observational O2 data that they chose to use might not be representativeness of S2D variability and trends.They argue that because GOBAI-O2 has no data gaps in time and space (gridded fields), it is more representative of real O2 ocean S2D variability and trends than the observations themselves. The authors also suggest that GOBAI-O2 represents the global ocean O2 mean than other gridding mapping methods (i.e., WOA18-O2).

My concern is that the authors do not quantify (metrics) why GOBAI-O2 is more representative or has greater ability/skill to represent the real ocean O2 mean and S2D variability than the observations they chose to use. What if they had used much additional QC observed O2 profile data coverage from other sources? The paper would benefit from using an objective metric comparison approach. For example, comparing GOBAI-O2's to other mapping methods using the same starting baseline O2 data..

The authors compare GOBAI-O2 to WOA2018-O2 as well as to selected GLODAP sections. I would be surprised to not see differences between these data products. For example, WOA2018-O2 mean climatology is based on a much larger pool of QCed winkler O2 measurements collected over 50 years (1965-2017; about 0.9 million profiles) than the QCed Winkler+BCG-ARGO O2 used by GOBAI-O2 (2004-2021). The authors could also compare GOBAI-O2 to the GLODAPv2 gridded O2 fields. In the end, these comparisons do not resolve GOBAI-O2's ability (metrics) to represent variability and trends better than observations and/or other mapping methods.

Finally, it would be useful if scientists could independently reproduce the GOBAI-O2 results. Are the authors planning on openly sharing the exact data (obs and model) and algorithms used?

Specific line comments and suggestions for consideration

For simplicity, I sometimes use "model" to refer to GOBAI-O2

38. What is the quantifiable metric for indicating that GOBAI-O2 provides a better representation of the real global and/or regional deoxygenation variability and trends than could be estimated from the observations themselves? Please clarify.

63. "have substantially improved the accuracy and reproducibility of optode-based [O2] measurements on Argo floats. In the absence of a reference (i.e., a true known value, a community-adopted certified reference material, or science community consensus reference

data), it is difficult to assess the "accuracy" of O2 field measurements (winkler and sensor based data).  Suggestion: "have substantially reduced the uncertainty (or increased the precision?) and reproducibility of optode-based [O2] measurements on Argo floats"

82. GLODAP measurements were largely collected during summer and spaced several years apart. Is the model output biased towards the more abundant ARGO O2 data coverage (Fig 1)?

Combining O2 data measured by Winkler and sensor based is not as straightforward as merging them together. Did the authors conduct preliminary QC checks on the BCG-ARGO O2 for internal data consistency with co-located discrete GLODAP data?

226. What is the uncertainty in deoxygenation content variability as a function of time (assumed constant)?

270. Table 2 has no units. I assume O2 in umol/kg

Fig 2a,b. These figures suggest an envelope of Δ[O2] roughly +- 10-20 umol/kg for relatively higher freq. Is the GOBAI-O2 total uncertainty adequate to resolve decadal-scale deoxygenation trends? In section 3.2.3 Interannual oxygen variability, the authors indicate a relatively small global decadal trend of −1.15 ± 0.26 µmol/kg/decade. Global deoxygenation trends range between 0.6% for models to 2% for observations (Fig 2 in Grégoire et al. 2021; https://doi.org/10.3389/fmars.2021.724913).

Fig 2c, f. Coastal and other oceanic regions have high seasonal to interannual variability. Why are Δ[O2] so small near coasts when compared to the subtropics/tropics?

336. "demonstrates an ability"; ability is a subjective term. Is this ability quantifiable?

337-338: "This bodes well for the ability of GOBAI-O2, which is trained on actual observational data, to represent decadal scale and seasonal variability in global ocean oxygen in the real world"

What quantifiable metric is being used to indicate that GOBAI-O2 represents the decadal scale and seasonal variability in global ocean oxygen in the real world?

As stated earlier, a large fraction of the  ARGO O2 obs were collected in the S. Hemisphere (Fig 2c) and measurements in GLODAP were mostly collected in summer. Global and regional seasonal variability would arguably be difficult to quantify with certainty with a limited observational coverage as used in this case.

I note that in line 345, the authors write ""For example, large Δ[O2] values in the eastern tropical Pacific and Atlantic, coupled with negative correlations in annual mean [O2] and large differences in annual trends and seasonal amplitudes, suggest more observations will be required for GOBAI-O2 to capture variability in that region"

355. I note that in ice-covered regions, there is also little air-sea gas exchange and limited biologically-mediated O2 production adding to undersaturation; particularly in the S. Ocean.

380. "Oxygen concentrations are extremely low in the deep, high-density North Pacific Ocean and North Indian Ocean due to the ages of those water masses' Rather than age specifically, what matters is the net balance of sources and sinks (i.e., air-sea exchange, ventilation/mixing, O2 respiration, redox chemistry).

Fig 5. Is GOBAI-O2 trained using isobars (depth) and isopycnals independently?

Fig 7. Are the model O2 values de-seasoned before depth integration by layers (i.e., subtracting the climatological monthly mean O2 in addition to the long-term mean)? If not, why not?

412: Suggest changing "The spatially weighted rate of deoxygenation in.." to "The spatially weighted decadal rate of deoxygenation in.."

Fig 7d shows that the temperature anomalies below about 500 m are relatively smaller prior to about 2015 than in later years. On the other hand, fig 7e shows relatively high (absolute value) O2 anomalies before and after about 2015 and at all depths and reflected in fig 7a. Is the implication that this is due to mean changes in ventilation to deeper depths?

443. The deoxygenation trends (discussed in 3.2.3 Interannual oxygen variability) seem to be in the 0.5-0.9% range. These trends are in agreement with AR5 model trend estimates (about 0.6%, Bopp et al., 2013). Schmidtko et al. (2017) indicated a global ocean deoxygenation trend of about 2% (See Fig 2 in Grégoire et al. 2021; https://doi.org/10.3389/fmars.2021.724913). Please address this apparent discrepancy.

455. Why do the authors attribute all model (algorithm) variability to natural and/or anthropogenic variability? As shown in Fig 8, model uncertainty is not insignificant..

460. Averaged globally, total uncertainty is 6 umol/kg (line 466). Visual inspection of Fig 8 suggests oceanic regions with total uncertainty values approximately > 10-20 umol/kg.These appear to be due to regional differences in the skill of the algorithm (line 485). Given these regional uncertainties, what would the magnitude of error bars be in Fig 7 for O2 (net anomalies of < 3 umol/kg)?

Fig 8 has no units (umol/kg?). I am surprised to see relatively low uncertainty values along coasts and WBCs where O2 seasonal variability is nominally large and obscures interannual and longer time-scale variability. Why is the algorithm uncertainty largest near the eastern tropical Pacific and Atlantic?

Fig 9. Differences are not unexpected. GOBAI-O2 (2004-2021; Winkler+ARGO O2 sensor) uses a smaller spatial and temporal data coverage than WOA18-O2 (1960-2017; Winkler only). I

would argue that an objective comparison would be to compare GOBAI-O2 and other mapping methods including the gridded fields of GLODAP and WOA18-O2.

It is interesting to see that WOA18-O2 minus GOBAI-O2 largest differences seem to follow isopycnals in the N and S. Pacific (F9b) and in the S. Atlantic (F9e). Is this a real feature or an artifact? Comparing GLODAPv2 gridded fields minus GOBAI-O2 would be useful.

510. The authors compare GOBAI-O2 to WOA18-O2; with GOBAI-O2 being about 10 µmol/kg lower than WOA18-O2. GLODAP includes a gridded mean O2 climatology. The authors should also compare GOBAI-O2 to the GLODAP gridded fields. Are the authors indicating that GOBAI-O2 provides a more accurate representation of the global ocean long-term O2 mean than WOA18-O2 and/or other data products? Please elaborate. The GOBAI-O2 global mean total uncertainty as a function of depth is about 4-10 umol/kg (Fig A10). Suggest adding some form of error bars at each depth in Fig A10 (i.e., std, serror, other).

511. WOA18-O2 uses O2 data starting in 1965; not 1955.

513. "... the World Ocean Atlas has been demonstrated to overestimate [O2] in suboxic zones (Bianchi et al., 2012)". Bianchi et al. indicated deviations of about 6 umol/kg in suboxic areas when compared to discrete O2 data profiles in GLODAP (Key et al. 2004). It is not unexpected that a mean O2 climatology spanning 1955-2004 would not exactly represent selected discrete O2 values. Similarly, I would not expect that other mapping techniques such as GLODAP O2 gridded fields exactly match all the discrete O2 data/profiles at any given depth/grid location.The same reasoning applies to GOBAI-O2. For example, Fig 10 shows O2 > 15 umol/kg differences between GOBAI-O2 and O2 values from GLODAP transects in the top 1 km.

GOBAI-O2 uncertainties seem larger than open-ocean O2 observing systems. GOOS Panel-Biogeochemistry-01-EOV-Oxygen Essential Ocean Variables (EOV) version 2.0 (August, 2017) provides uncertainty estimates (ARGO O2: ±2 umol/kg; Bottle Winkler ±0.5 µmol/kg). The figures are improving over time.

https://www.goosocean.org/index.php?option=com_oe&task=viewDocumentRecord&docID=17473
https://oceanexpert.org/downloadFile/35904

---

## Referee Comment (RC2)

**GOBAI-O2 regionally-tuned predictors and more diverse ensembles of ML algorithms should lead to increased confidence in estimates of ocean interior [O2]**

*Authors: Jonathan D. Sharp, Andrea J. Fassbender, Brendan R. Carter, Gregory C. Johnson, Cristina Schultz, John P. Dunne*

**General comments:**

The objective of the manuscript is to present the GOBAI-O2 tool, a 4D gridded product of O2 concentrations in the global ocean. It is based on machine learning algorithms trained on observations from BGC-ARGO and GO-SHIP in 7 regions and applied to temperature and salinity fields constructed from the Argo network. This product allows a fairly fine prediction of O2 concentrations from 2004-2021 on 58 vertical levels with a spatial resolution of 1°x1° allowing an analysis of spatial variability, seasonal cycles and decadal trends in O2.

The article is well constructed and written. The authors clearly present the methodology, and the prediction uncertainties. The authors indicate that GOBAI-O2 provides homogeneous O2 coverage improving O2 observations where spatial and temporal gaps are present in some regions.

The authors mention at the end the limitations of the product but they do not specify the added value of GOBAI-O2 compared to the existing observation networks. For example, it would be interesting to compare the GOBAI-O2 contribution vs. the ARGO-O2 network (with and without GO-SHIP). What is the real contribution of GOBAI-O2 ?

In BGC-ARGO, few O2 data have been qualified properly and adjusted in delayed mode even if a strong global efforts is and will done by the different GDAC. In this context, the authors do not precise how many O2 profiles from Argo network exists and how many have been used for the training ? What is the ratio total vs. qualified ? Probably the efforts will lead to more usable ARGO O2 profiles and thus contribute significantly to the overall O2 content coverage. In this case it will be interesting to know the added value of GOBAI-O2 predictions (metric comparison of the two approaches)

Another use of GOBAI-O2 not mentioned by the authors would be the use of GOBAI-O2 predictions to generate quality time series in areas poorly covered by reference data (long time series) which would allow for a finer qualification of O2 measurements from different platforms and often sensitive to drift over time. This product would be much better than the fields from WOA2018.

Also GOBAI-O2 has been trained from the Winkler O2 data of GO-SHIP but it would have been interesting to start from the O2 profiles from the ship's CTD and adjusted via the Winkler data. The vertical resolution would then be significantly improved. What are the limitations? Access to adjusted O2 profiles? If so, the document should mention and alert to this crucial point. It is now becoming essential to follow the FAIR data principles for all platforms.

The authors also mention the lack of other platforms to improve predictions, but this concerns in particular fixed moorings, which would be a plus in certain regions to increase the temporal resolution of observations (from minutes to months) over the entire water column, but only if a mooring array is available, otherwise a fixed point will not be significant and will not bring much. Also, the contribution of gliders sections will be relevant if we are interested in coast-open sea exchanges because most of the gliders are deployed in these specific sub-regions and their integration in the learning methods will not necessarily bring much.

**Specific comments:**

- a diagram explaining the principle of FNN and RFR would help readers understand the different algorithms used in this paper
- Table 2: Units of O2 is missing
- Figure 7: The O2 anomaly over depth (panel D) is close to zero between 2010-2015. Why? This is because GOBAI-O2 is centered on the year 2012? In this case, explain why it is centered on 2012.
- Figure 8: Units of O2 is missing. O2 uncertainties are higher near the equator and subtropical zones. Explain why

---

## Author Comment (AC1)

**Response to Reviewer Comments for *GOBAI-O₂: temporally and spatially resolved fields of ocean interior dissolved oxygen over nearly two decades**

The authors thank the two reviewers for their insightful comments on this manuscript. Below we have included detailed responses (in bold) to each of the reviewers' comments, which no doubt have improved both the GOBAI-O$_2$ data product and its accompanying description in this submitted manuscript. A revised version of the manuscript and a version with tracked changes will accompany this document. References to line numbers in this document refer to the revised manuscript, rather than the original submitted version.

Sincerely,
Jonathan D. Sharp and coauthors

**Reviewer 1 – Hernan Garcia**

General Comments

This is an interesting paper using a novel approach to quantify global ocean O2 content seasonal to decadal-scale (S2D) variability and trends. The authors use AI and ML in an effort to resolve global and regional ocean S2D O2 variability and trends. The authors combine/aggregate contemporary ship-based Winkler-based O2 data used in GLODAP and sensor-based O2 data from BCG-ARGO.

The authors indicate that the spatial and temporal heterogeneity coverage of the observational O2 data that they chose to use might not be representativeness of S2D variability and trends. They argue that because GOBAI-O2 has no data gaps in time and space (gridded fields), it is more representative of real O2 ocean S2D variability and trends than the observations themselves. The authors also suggest that GOBAI-O2 represents the global ocean O2 mean than other gridding mapping methods (i.e., WOA18-O2).

My concern is that the authors do not quantify (metrics) why GOBAI-O2 is more representative or has greater ability/skill to represent the real ocean O2 mean and S2D variability than the observations they chose to use. What if they had used much additional QC observed O2 profile data coverage from other sources? The paper would benefit from using an objective metric comparison approach. For example, comparing GOBAI-O2's to other mapping methods using the same starting baseline O2 data.

**We thank the reviewer for this suggestion. We now emphasize objective metrics that compare model oxygen fields reconstructed via the GOBAI-O$_2$ procedure to (1) fully resolved model fields and (2) subsampled model grid cells that correspond to the real-world distribution of available observations. This addition to the manuscript is discussed in more detail below.**

**Still, we do not claim that the GOBAI-O$_2$ mapping strategy is superior to other methods of objective interpolation or regression-based gap-filling, either in terms of representing seasonal to decadal variability in [O$_2$] or global ocean mean [O$_2$]. That assessment would**

**require an extensive intercomparison exercise that is outside the scope of this manuscript. We very much support that kind of exercise. So to aid with any future intercomparison study, we now include the original and vertically interpolated data on which GOBAI-O₂ is based in our supplemental material (Appendix C; https://doi.org/10.5281/zenodo.7747237).**

The authors compare GOBAI-O2 to WOA2018-O2 as well as to selected GLODAP sections. I would be surprised to not see differences between these data products. For example, WOA2018-O2 mean climatology is based on a much larger pool of QCed winkler O2 measurements collected over 50 years (1965-2017; about 0.9 million profiles) than the QCed Winkler+BCG-ARGO O2 used by GOBAI-O2 (2004-2021). The authors could also compare GOBAI-O2 to the GLODAPv2 gridded O2 fields. In the end, these comparisons do not resolve GOBAI-O2's ability (metrics) to represent variability and trends better than observations and/or other mapping methods.

**We agree that differences are expected when comparing GOBAI-O₂ with gridded climatologies that are centered around different time periods or discrete hydrographic sections that represent point measurements in space and time. These expectations are now more comprehensively addressed in the text (lines 577–580, 597–600). Additionally, we have added a figure (Figure 10) to compare GOBAI-O₂ to the GLODAPv2 gridded fields.**

**These comparisons to gridded fields of [O₂] are merely intended to place GOBAI-O₂ in the context of other commonly used products, not to indicate anything about its representation of [O₂] variability, trends, or global means. Indeed, the annual climatological field provided by GLODAP cannot be used to assess seasonal variability in [O₂], and nether the monthly climatological fields provided by WOA18 nor the annual climatological field provided by GLODAP can be used to assess trends or interannual variability in [O₂]. Still, we have added to Figure 6 the hemispheric climatological cycles from WOA18, corresponding closely to the depth levels from GOBAI-O₂, to address a comparison of seasonal variability.**

Finally, it would be useful if scientists could independently reproduce the GOBAI-O2 results. Are the authors planning on openly sharing the exact data (obs and model) and algorithms used?

**We have now included in Appendix C of the supplemental material (1) the observational dataset, both at native resolution and vertically interpolated to standard depth levels (https://doi.org/10.5281/zenodo.7747237), and (1) the regional models used to construct GOBAI-O₂ from RG09 temperature and salinity fields (Roemmich and Gilson, 2009) as well as spatiotemporal information (https://doi.org/10.5281/zenodo.7747308).**

Specific line comments and suggestions for consideration

For simplicity, I sometimes use "model" to refer to GOBAI-O2

38. What is the quantifiable metric for indicating that GOBAI-O2 provides a better representation of the real global and/or regional deoxygenation variability and trends than could be estimated from the observations themselves? Please clarify.

In the revised manuscript, we more clearly highlight quantifiable metrics to indicate the ability of GOBAI-$O_2$ to represent seasonal to decadal oxygen variability. These metrics are derived from fully resolved oxygen fields from the GFDL-ESM4 model (Dunne et al., 2020), oxygen fields from GOBAI-$O_2$-ESM4 (a reconstruction of GFDL-ESM4 oxygen fields using the approach of GOBAI-$O_2$), and subsampled GFDL-ESM4 grid cells within which historical observations are available. We calculate global weighted means ($\mu$) of grid-cell level [$O_2$] means, seasonal cycle amplitudes, long-term trends, and interannual variabilities. We also calculate differences ($\Delta$) between the fully resolved GFDL-ESM4 means versus GOBAI-$O_2$-ESM4 and versus the subsampled GFDL-ESM4 grid cells where observations exist. These metrics are provided below and in Table 3 in the revised manuscript.

| | Depth Interval (dbar) | GFDL-ESM4 | GOBAI-$O_2$-ESM4 | | Subsampled GFDL-ESM4 | |
|---|---|---|---|---|---|---|
| | | $\mu$ | $\mu$ | $\Delta$ | $\mu$ | $\Delta$ |
| **Mean [$O_2$]** **($\mu$mol kg$^{-1}$)** | 0-200 | 214.02 | 214.18 | −0.17 | 230.21 | −16.19 |
| | 200-1000 | 154.83 | 155.18 | −0.35 | 173.62 | −18.79 |
| | 0-2000 | 155.59 | 155.75 | −0.16 | 169.58 | −13.99 |
| **Seasonal Cycle Amplitude ($\mu$mol kg$^{-1}$)** | 0-200 | 12.04 | 10.16 | 1.88 | 12.05 | −0.01 |
| | 200-1000 | 3.37 | 2.11 | 1.27 | 5.94 | −2.57 |
| | 0-2000 | 2.60 | 1.87 | 0.73 | 3.89 | −1.29 |
| **Long-term Trend ($\mu$mol kg$^{-1}$ dec.$^{-1}$)** | 0-200 | −0.30 | −0.26 | −0.04 | 6.58 | −6.88 |
| | 200-1000 | −0.48 | −0.23 | −0.25 | 3.97 | −4.46 |
| | 0-2000 | −0.38 | −0.18 | −0.20 | 6.05 | −6.43 |
| **Interannual Variability ($\mu$mol kg$^{-1}$)** | 0-200 | 0.22 | 0.22 | 0.00 | 9.05 | −8.83 |
| | 200-1000 | 0.29 | 0.18 | 0.11 | 10.59 | −10.30 |
| | 0-2000 | 0.22 | 0.12 | 0.10 | 10.43 | −10.21 |

The agreement between these metrics for GOBAI-$O_2$-ESM4 and GFDL-ESM4 indicate that the GOBAI-$O_2$ mapping procedure provides a good representation of the seasonal to decadal variability in [$O_2$], and the large differences between the subsampled GFDL-ESM4 grid cells and GFDL-ESM4 indicate that observations alone are not enough to quantify this variability, and that some mapping/interpolation is necessary.

However, as mentioned above, it is not our intention to contend that GOBAI-$O_2$ provides a better representation of the real global and/or regional deoxygenation variability than could be estimated from applying an alternative mapping technique to the available observations. This conclusion would require an extensive analysis across mapping techniques (as indicated by the reviewer). For example, the objective interpolation strategy employed by the producers of the World Ocean Atlas might capture variability with similar success, but a comparison to evaluate that possibility is outside the scope of this dataset description paper. We have now included our raw and vertically interpolated datasets (https://doi.org/10.5281/zenodo.7747237) in the supplemental material (Appendix C) so that interested data product producers can evaluate other mapping strategies with the same dataset used in the development of GOBAI-$O_2$.

63. "have substantially improved the accuracy and reproducibility of optode-based [O2] measurements on Argo floats. In the absence of a reference (i.e., a true known value, a community-adopted certified reference material, or science community consensus reference data), it is difficult to assess the "accuracy" of O2 field measurements (winkler and sensor based data). Suggestion: "have substantially reduced the uncertainty (or increased the precision?) and reproducibility of optode-based [O2] measurements on Argo floats"

**Agreed, the suggested change has been made (lines 67–68).**

82. GLODAP measurements were largely collected during summer and spaced several years apart. Is the model output biased towards the more abundant ARGO O2 data coverage (Fig 1)?

**The process of training algorithms to represent the relationships between dissolved oxygen and physical, temporal, and spatial information is intended to minimize biases toward seasons and regions with more abundant data coverage. BGC Argo data can provide valuable information about seasonal biogeochemical cycles to complement synoptic snapshots from hydrographic cruises that occur mostly during the summer. Conversely, the vast synoptic scale information from GLODAP hydrographic sections can help fill gaps in space between BGC Argo float profiles.**

**Still, to ensure each profile from a given dataset (ship and float) is assigned equal weight in model training, the algorithms used to produce GOBAI-O$_2$ are now based on vertically interpolated data, rather than data provided at their native vertical resolutions (lines 126–132).**

Combining O2 data measured by Winkler and sensor based is not as straightforward as merging them together. Did the authors conduct preliminary QC checks on the BCG-ARGO O2 for internal data consistency with co-located discrete GLODAP data?

**Preliminary checks of quality-controlled BGC Argo data vs. GLODAP data are now highlighted in a supplementary figure (Figure D1 and below) that displays a comparison between co-located measurements from the two datasets. This figure displays binned GLODAP data (x-axis) as it relates to binned float data (y-axis). Bin sizes were 1° latitude × 1° longitude × monthly × RG09 depth levels ($n = 58$). The global small global median bias ($-1.25$ µmol kg$^{-1}$) between the two datasets was mitigated (reduced to 0.34 µmol kg$^{-1}$) by fitting the differences ($\Delta[O_2]$) to a linear least squares model as a function of float [O$_2$], and adding that [O$_2$]-dependent correction back on to the float [O$_2$] measurements.**

**The root mean squared error in [O$_2$] ($\pm10.1$ µmol kg$^{-1}$) compares favorably to similar analyses: Johnson et al. (2017) report a standard deviation of $\pm8$ µmol kg$^{-1}$ for float [O$_2$] measurements compared to Winkler titrations at the time of float deployment and $\pm12$ µmol kg$^{-1}$ for float [O$_2$] measurements compared to matchups from the GLODAP dataset, and Maurer et al. (2021) report a standard deviation of $\pm6.3$ µmol kg$^{-1}$ for float [O$_2$] measurements compared to Winkler titrations at the time of float deployment.**

[Figure]

226. What is the uncertainty in deoxygenation content variability as a function of time (assumed constant)?

**Whereas this section (2.5) describes estimates for [O₂] uncertainty at the grid cell level, we utilize the comparison between GOBAI-O₂-ESM4 and GFDL-ESM4 to evaluate uncertainty in global average [O₂] and oxygen content within different depth intervals. This is now alluded to in lines 243–244. These global uncertainty estimates are used when calculating uncertainty in oxygen content trends over time (Appendix E), which are reported in section 3.2.3.**

270. Table 2 has no units. I assume O2 in umol/kg

**Yes, that's correct. Units have been added to this table.**

Fig 2a,b. These figures suggest an envelope of Δ[O2] roughly +- 10-20 umol/kg for relatively higher freq. Is the GOBAI-O2 total uncertainty adequate to resolve decadal-scale deoxygenation trends? In section 3.2.3 Interannual oxygen variability, the authors indicate a relatively small global decadal trend of −1.15 ± 0.26 μmol/kg/decade. Global deoxygenation trends range between 0.6% for models to 2% for observations (Fig 2 in Grégoire et al. 2021; https://doi.org/10.3389/fmars.2021.724913).

**At the regional or 1° × 1° grid cell level, care should certainly be taken when interpreting trends, due to the level of uncertainty demonstrated in Figure 2. At the global scale, the metrics in Table 3 and comparison between GFDL-ESM4 and GOBAI-O₂-ESM4 in Figure A11 indicate that GOBAI-O₂ can resolve decadal-scale deoxygenation trends on the global scale with a good degree of confidence. Estimated uncertainty in global deoxygenation trends now takes into account uncertainty estimates in global average [O2] and oxygen inventory (Appendix E).**

Fig 2c, f. Coastal and other oceanic regions have high seasonal to interannual variability. Why are Δ[O2] so small near coasts when compared to the subtropics/tropics?

**Though some coastal regions have relatively low Δ[O₂], others are quite high (e.g., southeast Pacific in 2c, eastern Atlantic in 2f, and western Indian in 2f). These high-Δ[O₂] regions will often coincide with regions of high underlying interannual variability (panel j in Figures A8–A10). Nevertheless, some coastal areas do show relatively low Δ[O₂]. Here are two potential explanations for the apparently low Δ[O₂] values along some coastlines:**

**(1) Observational density is often relatively high along coasts, for example in the northwest Pacific, northeast Pacific, and northwest Atlantic (see Figure 1). In coastal areas where observational density is low (western equatorial Indian, eastern equatorial Atlantic), Δ[O₂] values (Figure 2) and total uncertainty values (Figure 8d) are very high.**

**(2) The GFDL-ESM4 model on which algorithm uncertainty (Figure 8c) is based may not be sufficiently capturing the true variability in dissolved oxygen along coasts. In this case, the GOBAI-O₂ algorithms will have an easier time trying to reconstruct the ESM4 variability than real-world variability. This is a potential deficiency of our uncertainty estimation procedure.**

336. "demonstrates an ability"; ability is a subjective term. Is this ability quantifiable?

**The metrics reported in Table 3 quantify the ability of GOBAI-O₂-ESM4 to capture seasonal to decadal scale variability in [O₂].**

337-338: "This bodes well for the ability of GOBAI-O2, which is trained on actual observational data, to represent decadal scale and seasonal variability in global ocean oxygen in the real world" What quantifiable metric is being used to indicate that GOBAI-O2 represents the decadal scale and seasonal variability in global ocean oxygen in the real world?

**There really is no way to directly quantify the ability of GOBAI-O₂ to represent [O₂] variability on a global scale in the real world. We display statistics in Tables 2 and B2 to demonstrate the ability of GOBAI-O₂ algorithms to predict [O₂] observations not included in model training, and to indicate their improved performance over previously developed seawater property estimation algorithms with the same predictor data (Carter et al., 2021; Table 2 and B4). We also display statistics in Tables 2 and B3 to demonstrate the ability of GOBAI-O₂ algorithms to predict simulated [O₂]. And, as discussed earlier, the metrics that are now highlighted in Table 3 indicate the ability of GOBAI-O₂ to represent decadal and seasonal [O₂] variability on a global scale in a simulated world. These exercises collectively provide our best approximation for how GOBAI-O₂ performs in the real world.**

As stated earlier, a large fraction of the ARGO O2 obs were collected in the S. Hemisphere (Fig 2c) and measurements in GLODAP were mostly collected in summer. Global and regional seasonal variability would arguably be difficult to quantify with certainty with a limited observational coverage as used in this case.

I note that in line 345, the authors write ""For example, large Δ[O2] values in the eastern tropical Pacific and Atlantic, coupled with negative correlations in annual mean [O2] and large

differences in annual trends and seasonal amplitudes, suggest more observations will be required for GOBAI-O2 to capture variability in that region"

**We agree with the reviewer that regions and time periods with limited data coverage are the most difficult to reconstruct with the current distribution of observations. However, Figure 3 indicates that most basin-scale surface and subsurface variability is represented well by the GOBAI-O$_2$ algorithms. Further, Table 3 and Figure A11 indicate that global variability can be reconstructed well, and far more effectively than with observations alone.**

355. I note that in ice-covered regions, there is also little air-sea gas exchange and limited biologically-mediated O2 production adding to undersaturation; particularly in the S. Ocean.

**We thank the reviewer for this note; a sentence has been added to acknowledge the effect of sea ice on air-sea gas exchange in ice-covered regions (lines 406–407).**

380. "Oxygen concentrations are extremely low in the deep, high-density North Pacific Ocean and North Indian Ocean due to the ages of those water masses' Rather than age specifically, what matters is the net balance of sources and sinks (i.e., air-sea exchange, ventilation/mixing, O2 respiration, redox chemistry).

**Yes, good point. That clarification has been added (lines 435–438).**

Fig 5. Is GOBAI-O2 trained using isobars (depth) and isopycnals independently?

**GOBAI-O$_2$ is trained using both depth and potential density as predictor variables. Sensitivity testing indicated that the best error statistics were obtained when both were included as predictors.**

Fig 7. Are the model O2 values de-seasoned before depth integration by layers (i.e., subtracting the climatological monthly mean O2 in addition to the long-term mean)? If not, why not?

**Values displayed in Figure 7 are anomalies of annual means from the long-term mean. Taking the annual means leads to a cleaner presentation and obviates the need to explicitly de-seasonalize the monthly values before calculating anomalies from the long-term mean. The same figure using de-seasonalized monthly anomalies rather than annual anomalies is shown below, and now as Figure A13 in the revised manuscript. De-seasonalized monthly values are indeed used for the calculations of trends and interannual variabilities given in Section 3.2.3, according to Appendix E.**

[Figure]

412: Suggest changing "The spatially weighted rate of deoxygenation in.." to "The spatially weighted decadal rate of deoxygenation in.."

**We have opted to keep this sentence as is because dec.$^{-1}$ is given in the units of the rate.**

Fig 7d shows that the temperature anomalies below about 500 m are relatively smaller prior to about 2015 than in later years. On the other hand, fig 7e shows relatively high (absolute value) O2 anomalies before and after about 2015 and at all depths and reflected in fig 7a. Is the implication that this is due to mean changes in ventilation to deeper depths?

**Oxygen anomalies at depth that are relatively larger than temperature (and therefore O$_2$ saturation) anomalies may reflect the importance on non-thermal drivers to deoxygenation, such as circulation/ventilation changes as the reviewer suggests, or changes in subsurface oxygen demand. This implication is discussed in lines 481–484.**

443. The deoxygenation trends (discussed in 3.2.3 Interannual oxygen variability) seem to be in the 0.5-0.9% range. These trends are in agreement with AR5 model trend estimates (about 0.6%, Bopp et al., 2013). Schmidtko et al. (2017) indicated a global ocean deoxygenation trend of about 2% (See Fig 2 in Grégoire et al. 2021; https://doi.org/10.3389/fmars.2021.724913). Please address this apparent discrepancy.

**We report our trends in this section as µmol kg$^{-1}$ or % per decade., whereas the values shown in Figure 2 of Grégoire et al. (2021) (0.6% and 2%) represent a 50-year period (1960–2010), so the Bopp et al. (2013) trend is about 0.12% per decade and Schmidtko et al. (2017) trend is about 0.4% per decade.**

**Therefore, the GOBAI-O$_2$ trend (~0.7% per decade) is actually larger in magnitude than both of those results. This may reflect an expected acceleration in deoxygenation over the**

**more recent period (2004–2022), interannual variability aliasing into the trend over a relatively short period of time, or a fundamental difference in the way GOBAI-O₂ represents global [O₂] relative to previous observational studies.**

**We compare GOBAI-O₂ trends with Schmidtko et al. (2017) and others — Helm et al. (2011) and Ito et al. (2017), nicely compiled by Bindoff et al. (2019) — in lines 501–511. In addition, we have added a sentence to address relatively lower deoxygenation trends from Earth system models (lines 511–513).**

455. Why do the authors attribute all model (algorithm) variability to natural and/or anthropogenic variability? As shown in Fig 8, model uncertainty is not insignificant.

**We now indicate that uncertainties from algorithm predictions can also contribute variability to the gridded fields (line 519).**

460. Averaged globally, total uncertainty is 6 umol/kg (line 466). Visual inspection of Fig 8 suggests oceanic regions with total uncertainty values approximately > 10-20 umol/kg. These appear to be due to regional differences in the skill of the algorithm (line 485). Given these regional uncertainties, what would the magnitude of error bars be in Fig 7 for O2 (net anomalies of < 3 umol/kg)?

**Uncertainty shading has been added to Figure 7. This uncertainty represents the standard deviation among differences between monthly mean [O₂] from GFDL-ESM4 versus GOBAI-O₂-ESM4 (section 3.1.2) in the relevant depth level. These uncertainties have also been incorporated into the analysis of trend uncertainties that are reported in section 3.2.3 (Appendix E).**

Fig 8 has no units (umol/kg?). I am surprised to see relatively low uncertainty values along coasts and WBCs where O2 seasonal variability is nominally large and obscures interannual and longer time-scale variability. Why is the algorithm uncertainty largest near the eastern tropical Pacific and Atlantic?

**Thanks for pointing this out. Units of $\mu$mol kg$^{-1}$ are now included on this figure. Panels are also now labelled a–d. Two potential explanations for the apparently low uncertainties along some coasts are provided in a previous response. Also, keep in mind that Figure 8 displays uncertainties on the 150 dbar pressure level, and so it is not representative of the integrated uncertainty over the entire depth range.**

Fig 9. Differences are not unexpected. GOBAI-O2 (2004-2021; Winkler+ARGO O2 sensor) uses a smaller spatial and temporal data coverage than WOA18-O2 (1960-2017; Winkler only). I would argue that an objective comparison would be to compare GOBAI-O2 and other mapping methods including the gridded fields of GLODAP and WOA18-O2.

**Agreed. However, as we've indicated above, a comparison between various mapping methods is outside the scope of this manuscript. We have compared GOBAI_O2 to the gridded fields of GLODAP and WOA18, but it should be noted that the monthly, time-**

**varying fields of GOBAI-O₂ are fundamentally different than the climatological monthly (WOA18-O2) and annual (GLODAP) averages of the other two products.**

It is interesting to see that WOA18-O2 minus GOBAI-O2 largest differences seem to follow isopycnals in the N and S. Pacific (F9b) and in the S. Atlantic (F9e). Is this a real feature or an artifact? Comparing GLODAPv2 gridded fields minus GOBAI-O2 would be useful.

**These anomalies are largely consistent in the GOBAI-O₂ to GLODAP gridded dataset comparison, now shown in Figure 10. As the reviewer has mentioned, it is difficult to determine whether these features are functions of data availability (ship data for WOA18 and GLODAP versus ship and float data for GOBAI-O₂), time period (1960–2017 for WOA18, centered on 2002 for GLODAP, and 2004–2022 for GOBAI-O₂), or mapping method (objective interpolation for WOA18 and GLODAP versus machine learning algorithms for GOBAI-O₂). This challenge has now been emphasized in lines 582–586. We wholeheartedly agree with the reviewer that a comprehensive comparison between mapping methods with consistent datasets will be an important future step to diagnose the origins of differences between resulting gridded fields.**

510. The authors compare GOBAI-O2 to WOA18-O2; with GOBAI-O2 being about 10 μmol/kg lower than WOA18-O2. GLODAP includes a gridded mean O2 climatology. The authors should also compare GOBAI-O2 to the GLODAP gridded fields. Are the authors indicating that GOBAI-O2 provides a more accurate representation of the global ocean long-term O2 mean than WOA18-O2 and/or other data products? Please elaborate. The GOBAI-O2 global mean total uncertainty as a function of depth is about 4-10 umol/kg (Fig A10). Suggest adding some form of error bars at each depth in Fig A10 (i.e., std, serror, other).

**We have added a new figure (Figure 10) comparing the long-term mean of GOBAI-O₂ to the GLODAP gridded product. We are not suggesting that GOBAI-O₂ provides a better or worse representation of global ocean long-term mean oxygen than GLODAP or any other commonly used data product. GOBAI-O₂ adds value in that it is unique compared to other available products in terms of its temporal resolution and coverage; nevertheless, we feel it is important to compare what can be compared between the available products.**

**Error shading representing spatial variability in uncertainty estimates has been added to what is now Figure A14, calculated as the standard deviation on each depth level of the mean uncertainties over time for each grid cell.**

511. WOA18-O2 uses O2 data starting in 1965; not 1955.

**Thanks for pointing this out. The change has been made (line 579).**

513. "... the World Ocean Atlas has been demonstrated to overestimate [O2] in suboxic zones (Bianchi et al., 2012)". Bianchi et al. indicated deviations of about 6 umol/kg in suboxic areas when compared to discrete O2 data profiles in GLODAP (Key et al. 2004). It is not unexpected that a mean O2 climatology spanning 1955-2004 would not exactly represent selected discrete O2 values. Similarly, I would not expect that other mapping techniques such as GLODAP O2

gridded fields exactly match all the discrete O2 data/profiles at any given depth/grid location.The same reasoning applies to GOBAI-O2. For example, Fig 10 shows O2 > 15 umol/kg differences between GOBAI-O2 and O2 values from GLODAP transects in the top 1 km.

**Since mapped products like GOBAI-O₂ are not expected to exactly represent discrete profiles, as indicated by the reviewer and shown in now Figure 11, we have removed the comment regarding overestimation of suboxic zone [O₂] by WOA18 as a potential explanation for disagreement between GOBAI-O₂ and WOA18.**

GOBAI-O2 uncertainties seem larger than open-ocean O2 observing systems. GOOS Panel-Biogeochemistry-01-EOV-Oxygen Essential Ocean Variables (EOV) version 2.0 (August, 2017) provides uncertainty estimates (ARGO O2: ±2 umol/kg; Bottle Winkler ±0.5 μmol/kg). The figures are improving over time.

**Indeed, measurement uncertainty is just a part of the uncertainty estimate for GOBAI-O₂. Therefore, our estimated uncertainties — including those from spatiotemporal gridding and algorithm-based estimates — are larger than the estimates from BGC Argo floats or Winkler titrations.**

https://www.goosocean.org/index.php?option=com_oe&task=viewDocumentRecord&docID=17473

https://oceanexpert.org/downloadFile/35904

**Reviewer 2 – Anonymous**

General comments:

The objective of the manuscript is to present the GOBAI-O2 tool, a 4D gridded product of O2 concentrations in the global ocean. It is based on machine learning algorithms trained on observations from BGC-ARGO and GO-SHIP in 7 regions and applied to temperature and salinity fields constructed from the Argo network. This product allows a fairly fine prediction of O2 concentrations from 2004-2021 on 58 vertical levels with a spatial resolution of 1°x1° allowing an analysis of spatial variability, seasonal cycles and decadal trends in O2.

The article is well constructed and written. The authors clearly present the methodology, and the prediction uncertainties. The authors indicate that GOBAI-O2 provides homogeneous O2 coverage improving O2 observations where spatial and temporal gaps are present in some regions.

The authors mention at the end the limitations of the product but they do not specify the added value of GOBAI-O2 compared to the existing observation networks. For example, it would be interesting to compare the GOBAI-O2 contribution vs. the ARGO-O2 network (with and without GO-SHIP). What is the real contribution of GOBAI-O2 ?

**Compared to the network of Argo floats with oxygen sensors or the plethora of ship-based oxygen measurements contained in the GLODAP database (and others), the contribution of GOBAI-O₂ is that it leverages those two datasets along with the Core Argo network to fill spatiotemporal gaps in the available observations. So GOBAI-O₂ is fundamentally different from and a value-added extension of the observational networks alone.**

**The contribution of the BGC Argo network compared to just GO-SHIP (GLODAP) measurements can be observed in Table B4: the [O₂]-estimation algorithms used to create GOBAI-O₂ outperform ESPER algorithms (which are trained on GLODAP data alone) at estimating ship-based oxygen observations. This highlights the added value of seasonally-resolved [O₂] data from Argo floats. Repeating the GFDL-ESM4 subsampling exercise outlined in this manuscript with simulated observations from BGC-Argo-only or GLODAP-only could further emphasize the impact of using both networks to create GOBAI-O₂, rather than one or the other.**

In BGC-ARGO, few O2 data have been qualified properly and adjusted in delayed mode even if a strong global efforts is and will done by the different GDAC. In this context, the authors do not precise how many O2 profiles from Argo network exists and how many have been used for the training ? What is the ratio total vs. qualified ? Probably the efforts will lead to more usable ARGO O2 profiles and thus contribute significantly to the overall O2 content coverage. In this case it will be interesting to know the added value of GOBAI-O2 predictions (metric comparison of the two approaches)

**For the development of GOBAI-O₂, we only use Argo profiles that have undergone delayed mode quality control (DMQC) and have quality flags of 1 (good), 2 (probably good), or 8 (interpolated/extrapolated) for pressure, temperature, salinity, and [O₂] (lines 119–121). Of the over 265,000 [O₂] profiles from 1,780 floats that were in the BGC Argo database at the time data were recovered (03 Mar. 2023), 133,488 profiles from 972 floats had undergone some degree of DMQC, and 128,562 profiles from 907 floats had some data points that met the required quality flags. This discrepancy between total Argo O₂ profiles and those that have been quality controlled emphasizes the potential for GOBAI-O₂ to be improved in a future iteration; even if no new observations are collected, the Argo-based training dataset can significantly increase in size with more resources directed toward quality control.**

Another use of GOBAI-O2 not mentioned by the authors would be the use of GOBAI-O2 predictions to generate quality time series in areas poorly covered by reference data (long time series) which would allow for a finer qualification of O2 measurements from different platforms and often sensitive to drift over time. This product would be much better than the fields from WOA2018.

**We thank the reviewer for this suggestion. A new sentence describing this use case has been added to the conclusions section (lines 620–622): "GOBAI-O₂ can also be useful as a dynamic reference check in data-sparse regions for new, sensor-based [O₂] measurements that would otherwise be compared to a static monthly climatology like WOA18."**

Also GOBAI-O2 has been trained from the Winkler O2 data of GO-SHIP but it would have been interesting to start from the O2 profiles from the ship's CTD and adjusted via the Winkler data. The vertical resolution would then be significantly improved. What are the limitations? Access to adjusted O2 profiles? If so, the document should mention and alert to this crucial point. It is now becoming essential to follow the FAIR data principles for all platforms.

**We mainly chose to use discrete Winkler [O₂] data from GLODAP for two reasons:**

**(1) this dataset is extensively quality-controlled, ensuring a reliable set of measurements is going into the GOBAI-O₂ algorithm training, and**

**(2) the vertical resolution of GOBAI-O₂ is on the order of tens to hundreds of meters, so the very fine vertical resolution offered by ship CTD data would make algorithm training prohibitively computationally expensive without adding much information to the final product.**

**These points are detailed in lines 105–109 of the revised manuscript.**

The authors also mention the lack of other platforms to improve predictions, but this concerns in particular fixed moorings, which would be a plus in certain regions to increase the temporal resolution of observations (from minutes to months) over the entire water column, but only if a mooring array is available, otherwise a fixed point will not be significant and will not bring much. Also, the contribution of gliders sections will be relevant if we are interested in coast-open sea exchanges because most of the gliders are deployed in these specific sub-regions and their integration in the learning methods will not necessarily bring much.

**We agree that information from fixed moorings and/or gliders could bring substantial benefits to GOBAI-O₂. However, data from these sources are not as well curated and quality controlled as data from discrete ship measurements and Argo floats. Also, the high temporal (moorings) and spatial (glider) resolutions of the raw datasets would a computational burden in the training of GOBAI-O₂ algorithms. The institution of a database like GO₂DAT (Grégoire et al., 2021) would be extremely helpful in bringing these new data sources into a product like GOBAI-O₂.**

Specific comments:

a diagram explaining the principle of FNN and RFR would help readers understand the different algorithms used in this paper

**This diagram has been added as Figure A4 and is referenced in line 191.**

Table 2: Units of O2 is missing

**Thank you, the units have been added to this table.**

Figure 7: The O2 anomaly over depth (panel D) is close to zero between 2010-2015. Why? This is because GOBAI-O2 is centered on the year 2012? In this case, explain why it is centered on 2012.

**The caption for Figure 7 has been modified to more clearly describe how anomalies shown in this figure are calculated: "Anomalies in each parameter are calculated as annual mean values minus the long-term mean either (a–c) integrated over a depth interval or (d–f) on a given depth level".**

Figure 8: Units of O2 is missing. O2 uncertainties are higher near the equator and subtropical zones. Explain why

**Units have been added to the colorbar label and text has been added to explain the high algorithm uncertainties in certain regions (lines 537–541).**

**References**

Bindoff, N. L., et al. Changing ocean, marine ecosystems, and dependent communities. In IPCC Special Report on the Ocean and Cryosphere in a Changing Climate, ed. H.-O. Pörtner, et al., 447–588, https://doi.org/10.1017/9781009157964, 2019.

Bopp, L., et al. Multiple stressors of ocean ecosystems in the 21st century: projections with CMIP5 models. Biogeosciences, 10, 6225–6245, https://doi.org/10.5194/bg-10-6225-2013, 2013.

Carter, B. R., et al. New and updated global empirical seawater property estimation routines. Limnol. Oceanogr.: Methods 19, 785–809, https://doi.org/10.1002/lom3.10461, 2021.

Dunne, J. P., et al. The GFDL Earth System Model version 4.1 (GFDL-ESM4.1): Model description and simulation characteristics. J. Adv. Model. Earth Syst. 12, e2019MS002015, https://doi.org/10.1029/2019MS002015, 2020.

Grégoire, M., et al. A Global Ocean Oxygen Database and Atlas for Assessing and Predicting Deoxygenation and Ocean Health in the Open and Coastal Ocean. Front. Mar. Sci. 8, 724913. https://doi.org/10.3389/fmars.2021.724913, 2021.

Helm, K. P., et al. Observed decreases in oxygen content of the global ocean. Geophys. Res. Lett. 38, 1–6, https://doi.org/10.1029/2011GL049513, 2011.

Ito, T., et al. Upper ocean O2 trends: 1958–2015. Geophysical Research Letters 44, 4214–4223, https://doi.org/10.1002/2017GL073613, 2017.

Johnson, K. S., et al. Biogeochemical sensor performance in the SOCCOM profiling float array. JGR: Oceans, 122, 6416–6436. https://doi.org/10.1002/2017JC012838, 2017.

Maurer, T. L., et al. Delayed-Mode Quality Control of Oxygen, Nitrate, and pH Data on SOCCOM Biogeochemical Profiling Floats. Front. Mar. Sci., 8, https://doi.org/10.3389/fmars.2021.683207, 2021.

Schmidtko, S., et al. Decline in global oceanic oxygen content during the past five decades. Nature, 542, 335–339, https://doi.org/10.1038/nature21399, 2017.

---

## Referee Report (RR1)

The authors have addressed my major concerns with the previous version. This is an interesting approach with potential for other applications. Thank you.

I would like the encourage the authors to include a statement of the new use of AI (GOBAI-O2) to map O2 in addition to other gap filling mapping techniques in the abstract or summary.  This is the main take home message of the paper.

Sampling, integrating data of known quality, and understanding ocean O2 variability is difficult. I am curious about the potential use of AI to recognize similar S2D meso-scale (or larger scale) O2 distribution patterns? Are there repetitive temporal/spatial patterns and/or higher frequency in the observations and/or model output?

Hernan Garcia

---

## Author Response (AR2)

**Response to Reviewer Comments for *GOBAI-O2: temporally and spatially resolved fields of ocean interior dissolved oxygen over nearly two decades**

Thanks again to the reviewers for carefully looking over our manuscript and providing constructive comments and suggestions. Below we have included detailed responses (in bold) to the editor's and reviewers' comments, which have led to further improvement of the GOBAI-O2 product and associated manuscript. A revised version of the manuscript and a version with tracked changes will accompany this document. References to line numbers in this document refer to the revised manuscript, rather than the original submitted version.

Sincerely,
Jonathan D. Sharp and coauthors

**Topical Editor – Anton Velo**

Public justification (visible to the public if the article is accepted and published):

In this article, the authors provide a way relevant work and dataset for the field, which promises significant future use by a wide number of different stakeholders. That also implies an important responsibility to ensure the precision and accuracy.

Getting to the content, I believe that reviewer#1's comments (after 1st iteration) about uncertainties of the ARGO-O2s should be addressed before publication, I copy them below:

[see Reviewer 1 comments]

And just as a minor comment, I'd have preferred the usage of ML term instead of AI in the work, as the latter tends to be used for smart systems with take autonomous decisions, but as the term definition is very ambiguous and broad (and includes ML), I've no objections.

**We thank the editor for taking care to ensure that uncertainty in GOBAI-O2 is carefully quantified and properly communicated. We indeed hope to see widespread use of the product in the future by a variety of stakeholders, so we recognize this as an important and necessary responsibility. In response to the reviewer comments, we have updated the data product with new estimates for oxygen measurement uncertainty, as well as a set of newly quality-controlled float oxygen profiles.**

**The inclusion of "AI" in the data product title was partly to form a coherent acronym, with the recognition that ML is a specific subfield under the AI umbrella. We now address this choice in the manuscript (lines 95–98).**

**Reviewer 1 – Anonymous**

My comment is on the revised manuscript by Sharp and co-authors, developing a global map of dissolved oxygen concentrations since 2004 using sensor O2 measurements from autonomous floats and the QCed shipboard measurements from GLODAP. I have read the original manuscript,

review comments, and the revised manuscript, and this report is based on the latest version. Starting from the conclusion, I would like to enthusiastically encourage the publication of this manuscript and the dataset with one further (minor) revision.

This work is an important milestone in the biogeochemical oceanography. The methodology of the machine-learning based oxygen maps was initially developed by Giglio et al (2018), and the authors did an excellent job of extending it to the global map. One notable step is that the authors merged the float and shipboard O2 with a small offset determined from the overlapping profiles. The float O2 is given a small and uniform offset based on the apparent, underestimation of a few micro mol/kg. This is important because we are concerned about the long-term trend O(1%) per decade or less, such that small offset like this can change the conclusion significantly. Another noteworthy effort in this paper is the creative use of the ESM output (GFDL-ESM4) to assess the skill of the gap-fill and mapping, adding confidence to the effectiveness of this ML-based approach.

I read that there were many excellent comments and questions from the first two reviewers, and I appreciate that the authors took the time to address these comments. I believe this process improved the manuscript significantly. I do not feel the need to repeat any of the points raised by these review comments at this time.

Below are my comments about the uncertainties of the GOBAI-O2 product that I would like the authors to consider & discuss before finalizing this paper. I do not wish to further delay the publication of this paper, but I think the authors are responsible for raising the awareness for the data users about the limitations and potential deficiencies of the ARGO-O2 dataset upon which GOBAI-O2 dataset is built.

**We thank the review for their thorough evaluation of our work and enthusiasm about its publication. We've dedicated a significant amount of effort to evaluate the quality of the datasets, quantify the skill of the machine learning model predictions, and assess confidence in the GOBAI-O$_2$ fields through model-based simulations, so we appreciate the reviewer's recognition of these important steps. Finally, we acknowledge the importance of providing the highest quality data product possible at this time and fully alerting potential users to its strengths and weaknesses, so we will do our best to respond to the reviewer's concerns.**

1. The ARGO-O2 is an evolving technology with variable accuracy and uncertainties even in the delayed mode dataset. Can we confirm whether or not GOBAI-O2 uses only optode sensors calibrated by the known methods (either climatology or in-air O2 measurements)? If other types of sensors are used, it should be stated.

**More than 90% of the BGC Argo floats used in the creation of GOBAI-O$_2$ are equipped with optode oxygen sensors. The rest are equipped with electrochemical oxygen sensors. Quality control measures for the oxygen sensors are primarily based on climatology comparisons and in-air measurements (see answer to next question).**

2. The uncertainty range should be re-considered. Of the O(900) floats that passed the QC step, there is a diversity of sensors and calibration methods. Only relatively new ARGO-O2 profiles are

calibrated with in-air O2 measurement, and the older data (essentially all floats deployed 2015) had to be calibrated using climatology. Because GOBAI-O2 blends all kinds of O2 sensors, and uncertainties should be re-assessed in section 2.5. My suggestion would be no less than 3% for measurement uncertainty considering the climatological calibration from older O2 sensor profiles.

**The reviewer raises an important question regarding the reliability of BGC Argo oxygen measurements, as methods of quality control for these measurements are being actively developed and refined. Common methods include calibration via in-air measurements, comparisons to surface climatologies, comparisons to subsurface measurements, and in situ optode calibrations. A statement has been added (lines 128–132) detailing the proportion of float profiles used in the development of GOBAI-O$_2$ that fall into each of these categories of quality control. These proportions are based on a thorough analysis of the calibration comments provided with the float data. We revise our measurement uncertainty estimate based on this analysis (see answer to next question).**

3. In section 2.5, the combined measurement uncertainty is stated as 1.5%. I believe this is an average of 1% in GLODAPv2 and 2% in ARGO-O2. As stated above, ARGO-O2 should have 3% uncertainty at least, and the combined uncertainty should consider the number of profiles from each dataset. My reading is that the authors used O(21k) profiles from GLODAPv2 and O(133k) profiles from ARGO-O2. Therefore, measurement uncertainty should be dominated by the ARGO-O2, and due to the blend of unaccounted sensor/calibration type, it should be 3% in my opinion.

**Measurement uncertainty has been amended to 3%, as suggested by the reviewer. We now discuss this choice as a consequence of multiple factors: GLODAP uncertainty, BGC Argo uncertainty, the lack of response time corrections to BGC Argo data, and the relative proportion of float profiles compared to ship profiles (lines 271–284). We have added a statement to section 3.2.4 indicating the implications of measurement uncertainty in GOBAI-O$_2$ for continued progress in sensor development and quality-control (lines 581–583).**

4. The issue of sensor response time is briefly discussed in section 2.5. The community has not yet implemented the response time correction (RTC) of optode sensors. In section 2.5 the authors stated that 2% uncertainty of ARGO-O2 is still optimistic, but I don't think this statement adequately describe the problem. The lack of RTC in ARGO-O2 data not only increases the measurement uncertainty but also causes systemic bias in the vicinity of oxycline. This is a larger problem beyond the scope of this paper, however, it should be discussed and the data users must be warned. Much stronger statement of caution should be provided, perhaps in the discussion/conclusion. Current ARGO-O2 dataset can include significant bias and uncertainty in the oxycline region beyond the level that is characterized in the uncertainty analysis. Simply adding a constant offset or blending GLODAPv2 data using machine learning will NOT correct this issue. Progress in this area will be much needed for future research.

**We thank the reviewer for highlighting this opportunity to emphasize the importance of considering optode sensor response time. We have expanded the discussion of sensor response time in section 2.5 and now highlight the potential for systematic biases (lines 280–284). We have also added a statement in the Conclusions to caution potential users about the**

**impact of the lack of response-time-corrections to float [$O_2$] data, especially when floats cross steep gradients (lines 676–677).**

**Reviewer 2 – Hernan Garcia**

The authors have addressed my major concerns with the previous version. This is an interesting approach with potential for other applications. Thank you.

I would like the encourage the authors to include a statement of the new use of AI (GOBAI-O2) to map O2 in addition to other gap filling mapping techniques in the abstract or summary. This is the main take home message of the paper.

Sampling, integrating data of known quality, and understanding ocean O2 variability is difficult. I am curious about the potential use of AI to recognize similar S2D meso-scale (or larger scale) O2 distribution patterns? Are there repetitive temporal/spatial patterns and/or higher frequency in the observations and/or model output?

**The authors thank the reviewer for evaluating the manuscript again. We have added a statement in the abstract indicating the novelty of machine learning for gap-filling ocean interior biogeochemical observations, and advocating for continued development alongside other mapping techniques (lines 22–23).**